# Power-free knee rehabilitation robot for home-based isokinetic training

Yanggang Feng[1,8] ✉, Haoyang Wu[1,8], Jiaxin Ren[1], Wuxiang Zhang[1] ✉, Xiu Jia[2], Xiuhua Liu[3], Xingyu Hu[1], Haoxiang Jing[1], Yuebing Li[1], Yuhang Zhao[1], Ziyan Wang[1], Xuzhou Lang[1], Junjia Xu[1], Yixin Shao[1], Qi Su[4], Yuanmingfei Zhang[5], Mouwang Zhou[5], Ke Liu[6], Yong Nie[7], Jian Wang[5], Fuzhen Yuan[5], Liu Wang[2] ✉ & Xilun Ding[1] ✉

Robot-assisted isokinetic training has been widely adopted for knee rehabilitation. However, existing rehabilitation facilities are often heavy, bulky, and extremely energy-consuming, which limits the rehabilitation opportunities only at designated hospitals. In this study, we introduce a highly integrated and lightweight (52 kg) knee rehabilitation robot that can provide home-based isokinetic training without external power. By integrating a motor, torque/angle sensors, control circuit, and energy regeneration circuit into a single driver module, our robot can provide power-free isokinetic training by recycling mechanical work from the trainee. Ten postsurgical subjects were involved in an interventional randomized trial (ChiCTR2300076715, Part I) and the cross-sectional area of trained legs (experimental group) was significantly higher than that of untrained legs (control group). The primary outcomes, muscle growth (quadriceps: 5.93%, hamstrings: 10.27%) and strength improvements (quadriceps: 70%, hamstrings: 84%), achieved with our robots surpass those of existing commercial rehabilitation devices. These findings indicate that our robot presents a viable option for home-based knee rehabilitation, significantly enhancing the accessibility of effective treatment.

Knee injuries, such as tears of ligament or meniscus and dislocations, impact the mobility of billions worldwide, leading to chronic pain and reduced quality of life[1-4]. The current therapeutic approaches for managing knee injuries usually involve plyometric rehabilitation training that helps build strength, power, and balance of the associated muscles[4-7]. Categorized by the muscle contraction types, plyometric rehabilitation training includes isometric, isotonic, and isokinetic training[8-10]. Particularly, isokinetic training denotes dynamic muscular contractions wherein movement speed is constant, ensuring that muscles exert maximal tension throughout the entire range of joint motion[8-12]. This constant speed characteristic naturally integrates a safety mechanism that reduces resistance if the patient experiences pain or discomfort[13], while also enabling continuous monitoring of rehabilitation progress through objective assessments of muscle strength[14,15]. Recent studies have demonstrated the effectiveness of isokinetic training for knee rehabilitation[16-18], such as enhancing muscle strength, improving stability, increasing postoperative walking speed, and reducing knee pain, in various knee conditions, including

[1]School of Mechanical Engineering and Automation, Beihang University, Beijing, China. [2]CAS Key Laboratory of Mechanical Behavior and Design of Materials, Department of Modern Mechanics, University of Science and Technology of China, Hefei, China. [3]Intelligent Science & Technology Academy Limited of CASIC, Beijing, China. [4]Department of Automation, Shanghai Jiao Tong University, Shanghai, China. [5]Peking University Third Hospital, Beijing, China. [6]Department of Advanced Manufacturing and Robotics, Peking University, Beijing, China. [7]West China Hospital, Sichuan University, Chengdu, China. [8]These authors contributed equally: Yanggang Feng, Haoyang Wu. ✉e-mail: yanggangfeng@buaa.edu.cn; zhangwuxiang@buaa.edu.cn; wangliu05@ustc.edu.cn; xlding@buaa.edu.cn

anterior cruciate ligament (ACL, *16*), posterior cruciate ligament (PCL, *17*) and osteoarthritis (OA, *18*).

Isokinetic training is usually administered by a specialized rehabilitation facility that applies resistive forces for controlled movement of the injured knee. Existing commercial facilities, such as Biodex[19], Kineo[20], CONTEX MJ[21], IsoMed2000[22], and research prototypes represent the state-of-the-art in isokinetic training technology[23–26] (Supplementary Table S1). These systems employ various operating principles in their driver modules, including high-power servo motors[19–23], magnetorheological (MR) fluids[24], electrorheological (ER) fluids[25], and electromagnetic powder brake[26] (Fig. 1a). By modulating the electrical current, the driver produces an adjustable damping force that precisely counteracts knee motion. However, the employment of a high-power driver module and external electricity usually involves several external hardware (e.g., energy supply module, power transformer, and control cabin), making these facilities bulky, heavy, and extremely energy-consuming. For instance, the Biodex system works under 230 V power, consumes 3450 W, and weighs over 600 kg. As a result, access to isokinetic rehabilitation is largely limited to well-equipped hospitals or rehabilitation centers. We were unable to find any studies that utilized isokinetic training devices in home-based settings, and there has been a growing yet unfulfilled need for compact isokinetic training systems that can be deployed for home-based rehabilitation.

Generating controlled damping without external electricity is the key to downsizing the rehabilitation device[19–22,27], e.g., isokinetic facility. A feasible solution is to convert the excessive mechanical work (referred to as negative mechanical work) by the patient to generate damping during training[27]. Among others, the motor-winding short (MWS) stands as a promising energy-efficient damping strategy[28,29]. By modulating the time of short motor windings, it can regulate the induced electrical current, thereby controlling the resultant damping effect from the motor while harvesting negative mechanical work.

Notably, this strategy can provide electromagnetic damping torque of tens of N·m with energy consumption lower than several watts[28,29]. For instance, a transtibial prosthesis utilizing the MWS method achieved precise control with a small energy consumption of 1.5 W.[28] Moreover, while providing the damping, the energy can be optimized[30–32]. Another study integrated energy regeneration with the MWS technique, successfully harvesting 1/3 of energy during ambulation, thus further minimizing energy usage[30]. Despite the low energy consumption, existing MWS-based rehabilitation facilities still require additional energy input. In addition, the MWS strategy has not been applied to the isokinetic facilities, not to mention a highly integrated prototype with clinical trials.

In this work, we develop a highly integrated isokinetic knee rehabilitation robot for home-based isokinetic training (Fig. 1b). Our robot features a single driver module (Fig. 1c) that highly integrates all key components, including the angle sensor, motor, control circuit, and torque sensor. Utilizing the MWS strategy, the robot can regenerate enough energy that is higher than that for operating the robot, eliminating the requirement for external energy and achieving a fully power-free system (Fig. 1d). The isokinetic training is realized by closed-loop control of the damping provided by the robot in real-time. Compared with existing heavy and energy-consuming products, our robot has a significantly reduced weight and power (Fig. 1e), holding great potential for home-based knee rehabilitation. The power-free results of the robot were empirically tested with 20 subjects: 10 post-surgical subjects and 10 healthy individuals. We also carried out 6-week interventional clinical trials with 10 patients after knee surgeries and validated recovered knee functions after training using our robot (Fig. 1f).

## Results
### Power-free isokinetic training robot and working principle
The power-free isokinetic training robot and its working principle are shown in Fig. 2. The driver module, shown in Fig. 2a, includes a main

arm, two limiters, a brushless direct current (BLDC) motor, a torque and an angle sensor, an adapter plate, and a control circuit (see Supplementary Movie S1). The main arm is used to connect the leg by securing the ankle of the trainee. By altering the ankle fixture, both the left leg and right leg can be trained, as illustrated by the inset in Fig. 2a. The predefined maximum lifting and retracting angle range of the main arm is constrained by two limiters placed at the front and back, which ensures the safety during the training. The main arm is linked to the BLDC motor through a series connection with the torque sensor and the adapter plate. The torque sensor measures the training torque, while the adapter plate facilitates the connection between the torque sensor and the BLDC motor. The angle sensor measures the angle of the knee joint (denoted as $\theta$) during the training. The control circuit commands the BLDC motor to offer electromagnetic braking while collecting and transferring the negative mechanical energy.

Our robot enables isokinetic training while regenerating energy from human dynamics. The working principle is shown in Fig. 2b–h. To perform the knee rehabilitation training, a trainee sits on the chair with the leg fastened to the ankle fixture (Fig. 2b). The training objectives such as angular velocity (denoted as $\omega = d\theta/dt$) are obtained from a professional therapist via a smartphone (Fig. 2c) and transmitted to the microcontroller unit (MCU) (Fig. 2h) via a Bluetooth module (Fig. 2f). The subject actuates the BLDC motor (Fig. 2d) by lifting and retracting the main arm. The real-time training velocity and torque of the motor (Fig. 2d) are detected by the angle and torque sensors, respectively (Fig. 2e). The measured data are displayed on the smartphone via a Bluetooth module (Fig. 2f). The constant training velocity is realized by a proportional-integrated (PI) control algorithm. By comparing the actual data with training objectives, the MCU generates a precise and appropriate pulse width modulation (PWM) signal. This signal is conveyed to the driver, where it undergoes amplification through the Smart Gate Driver and MOSFETs (Fig. 2g). The amplified signal is then transmitted to the BLDC motor, which modulates the damping of the motor. The modulated damping provides accommodative resistance to realize the isokinetic training. The detailed mechanical assembly of the driver module is shown in Supplementary Fig. S1.

### Demonstration of isokinetic training
To evaluate the performance of our isokinetic training robot, we conducted a study with 10 post-surgical subjects and 10 healthy subjects (see Supplementary Tables S2 and S3). Preparation for the isokinetic training involves four steps as outlined in Fig. 3a (detailed description see "Methods" section). The training protocol consists of eight repetitions of lifting and retracting movements in one cycle. Experimental pictures are shown in Fig. 3b. Figure 3c presents representative isokinetic training data from a post-surgical subject. A constant training velocity of $\omega = 60° \cdot s^{-1}$ is maintained during both the lifting and retracting processes[33]. During the training, subjects are required to exert maximum muscle strength to lift and retract the calf. Notably, the training device is designed not to assist leg motion but to provide accommodative resistance if the velocity exceeds $60° \cdot s^{-1}$, ensuring the training intensity. Within each repetition, the training velocity exhibits a pattern of quick acceleration, stability at target velocity, and rapid deceleration, in which the constant velocity time occupies about 70% of training (Fig. 3c, Velocity). The torque displays a similar pattern with a gradual decline over the training course due to muscle fatigue (Fig. 3c, Torque). As the training continues, the regenerated energy accumulates in both the lifting and retracting processes (Fig. 3c, Regenerated energy). A demonstration video is shown in Supplementary Movie S2.

### Validation of power-free isokinetic training
Figure 4 presents the validation of power-free isokinetic training conducted with 10 post-surgical participants and 10 healthy controls (Supplementary Tables S2 and S3). By recording the regenerated

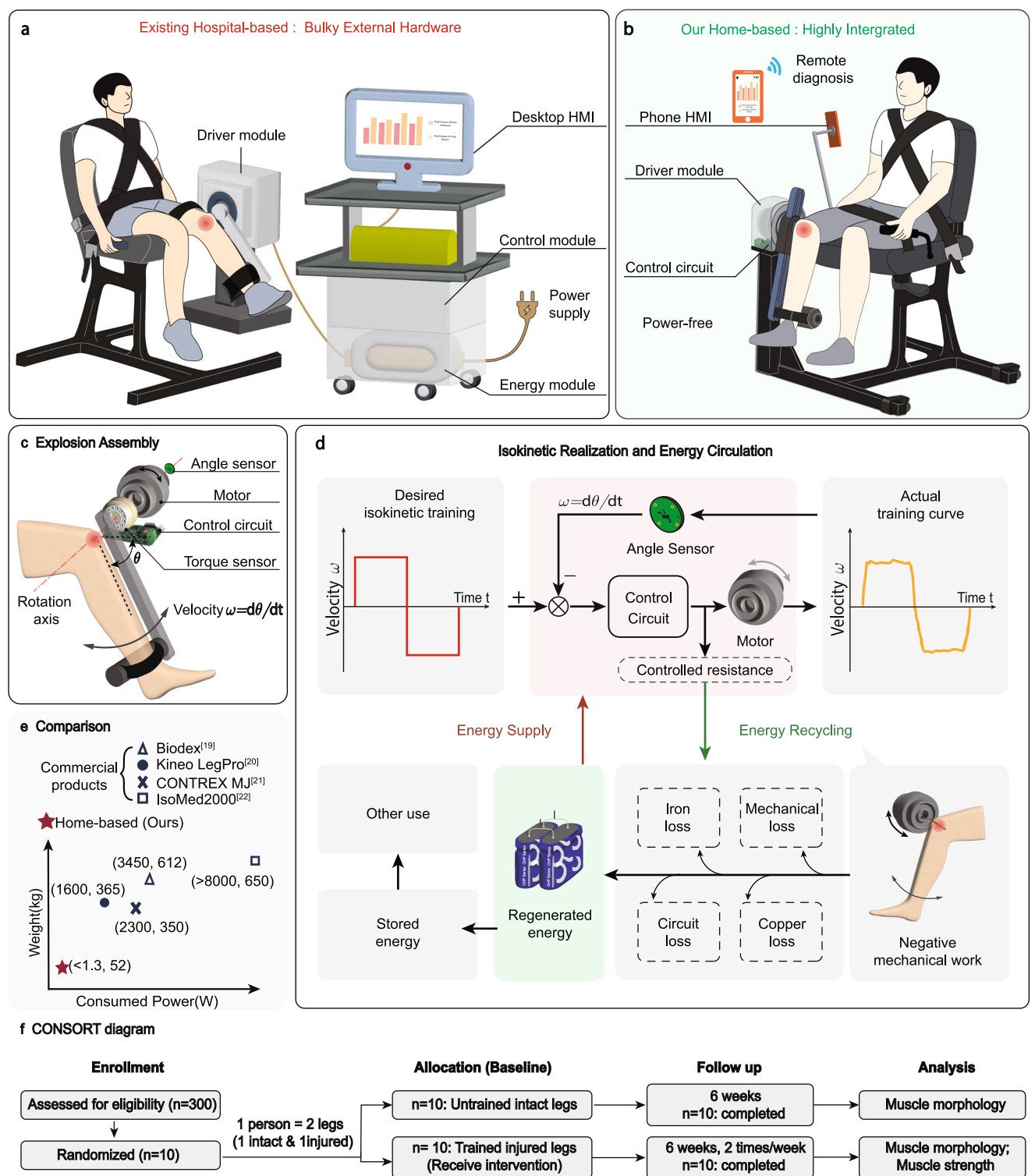

**Fig. 1 | Proposed power-free knee rehabilitation robot, realization and comparison. a** Existing hospital-based rehabilitation devices. These devices are bulky and need additional power supplies. **b** Proposed home-based isokinetic training robot. The proposed isokinetic training robot is power-free and highly integrated. **c** Explosion Assembly. During the training, the rotation center of the knee joint and the motor are concentric. **d** Isokinetic realization and energy regeneration. The isokinetic training robot can generate energy while training. A desired constant velocity is predetermined. The velocity is monitored via angle sensors and the motor output resistance is modulated to ensure the actual velocity closely approximates the predetermined desired velocity. The controlled resistance is adjusted and the negative mechanical work during training is harvested. The harvested energy is subsequently channeled to power the control circuit, sensors, and other usage. **e** Comparison. A comparison between commercial isokinetic training robots and our proposed isokinetic training robot. The consumed power and weight of our robot are both dramatically reduced. **f** CONSORT diagrams of subjects.

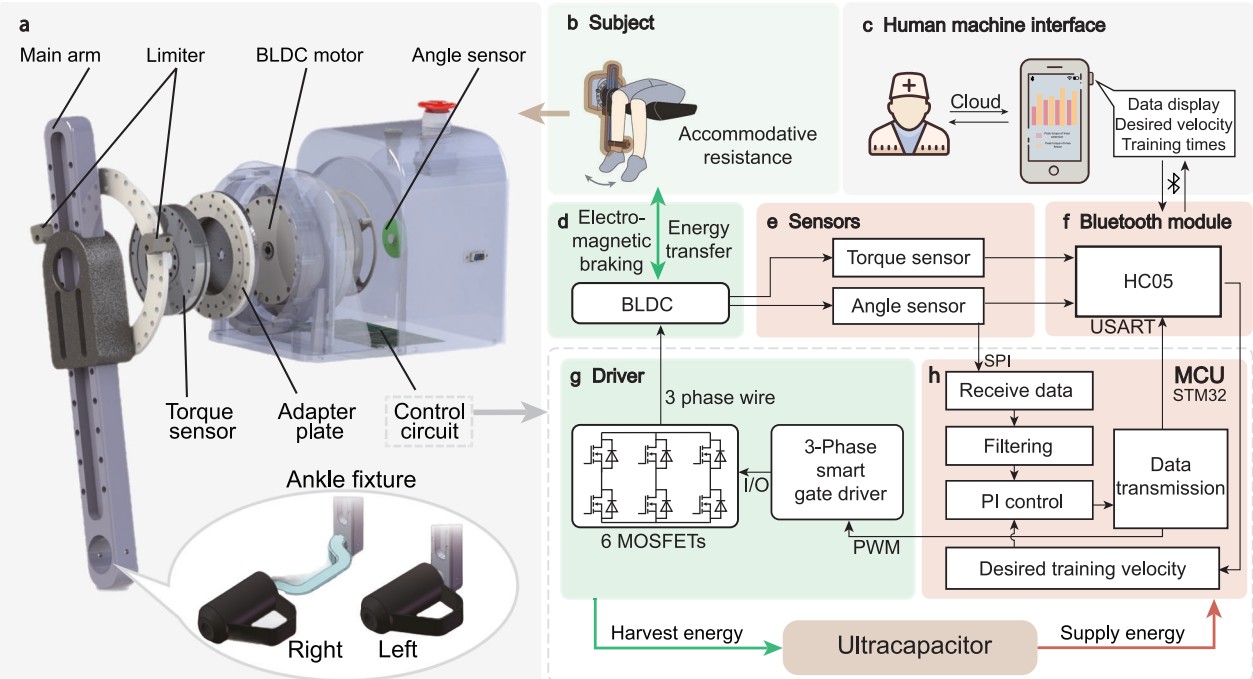

**Fig. 2 | Highly integrated isokinetic training robot and working principle.**
**a** Exploded view of the driver module with ankle fixture for both legs. **b** Subject training: A subject sits on the chair during the training, and the main arm offers the subject accommodative resistance to maintain the isokinetic training.
**c** Human−machine interface: A smartphone displays training data in real-time and communicates with training assistants remotely. **d** Brushless direct current (BLDC) motor: BLDC motor provides electromagnetic braking and transfer energy.
**e** Sensors: The angle sensor and torque sensor communicate with a microcontroller

unit (MCU) through a serial peripheral interface (SPI) and analog-to-digital (A/D) module, respectively. **f** Bluetooth module: A Bluetooth module receives the data from the MCU through a universal synchronous asynchronous receiver transmitter (USART) and sends it to the smartphone. **g** Driver: The 6 MOSFETs and gate driver amplify the control signal received from MCU and connect with BLDC through 3-phase wires. **h** MCU: The STM32F405 MCU communicates with the sensors, drivers, and a Bluetooth module.

energy and consumed energy over eight repetitions of exercises, we can calculate the average regenerated power (denoted as $P_{reg}$) and consumed power (denoted as $P_{con}$). A power-free isokinetic training is realized when $P_{reg} > P_{con}$ or the efficiency ratio $\eta = P_{reg}/P_{con}$ is >1. Initial testing involved 10 healthy subjects and results are presented in Fig. 4a. With the highest recorded value being 373% (Subject 10) (Fig. 4b), the average efficiency ratio reaches $\eta = 212\%$. Notably, the efficiency ratio exceeds 1 across all subjects, indicating a net positive energy regeneration (See Supplementary Table S4). This performance is consistent despite variations in individual muscle strength, as shown by the large difference between maximum and minimal torque in Fig. 4c. Further validation with 10 post-surgical subjects is presented in Fig. 4d and the average efficiency ratio of 186% is achieved (Fig. 4e). Despite the expected decrease in torque due to knee impairments (Fig. 4f), the efficiency ratio remains above 1, underscoring the robot's ability to provide power-free training across varying levels of physical condition in diverse rehabilitation scenarios.

**Primary outcomes-muscle morphology**
To assess rehabilitation outcomes, 10 post-surgical subjects participated in a 6-week rehabilitation program (status of the trial: Part I completed). The program comprises 12 sessions (2 sessions/week), each consisting of four exercises with eight repetitions of a lifting-retracting movement. The primary outcomes comprise the cross-sectional areas (CSAs) measurements and the isokinetic muscle strength assessment. Magnetic resonance imaging (MRI) was employed to capture the CSAs of the quadriceps and hamstrings in both the trained and untrained legs, before and after the program

(Fig. 5a). The CSAs are widely adopted as the principal metric for assessing muscle growth and recovery[34–36]. Quantification of CSAs was performed using specialized medical imaging software (see the "Methods" section, Supplementary Table S5). Comparative analyses of the quadriceps and hamstrings in CSA in both the trained leg and untrained leg are shown in Fig. 5f and g, respectively. For subjects with different physical conditions, there is a noticeable increase in quadriceps' CSA with an average of $2.92 \pm 1.46$ cm$^2$ (Fig. 5f), leading to a growth rate of $5.93 \pm 3.3\%$ (Fig. 5g). Similarly, the hamstrings in the trained leg also exhibit a notable increase in CSA with an average of $2.42 \pm 1.25$ cm$^2$ (Fig. 5f). This increase translates to a significant growth rate of $10.27 \pm 5.5\%$ (Fig. 5g), underscoring the effectiveness of the rehabilitation program in enhancing muscle volume across different muscle groups.

In contrast to the trained leg, the quadriceps muscle in the untrained leg demonstrated negligible growth, with an average increase in CSA of $0.17 \pm 0.80$ cm$^2$ (Fig. 5f) which equals a growth rate of $0.4 \pm 1.1\%$ (Fig. 5g). Similarly, the hamstring muscle of the untrained leg showed a slight increase in CSA, averaging $0.32 \pm 1.50$ cm$^2$ (Fig. 5f). This growth, representing a rate of $1.9 \pm 5.0\%$ (Fig. 5g), while slightly higher than that of the quadriceps, remains substantially lower than the improvements observed in the trained leg. The comparative analysis between the trained and untrained legs underscores the efficacy of our rehabilitation robot in eliciting targeted muscle growth. The minimal changes in the untrained leg serve as a control, demonstrating that the significant muscle growth in the trained leg is directly attributed to the rehabilitation program. Detailed MRIs of all subjects are presented in Supplementary Figs. S2 and S3.

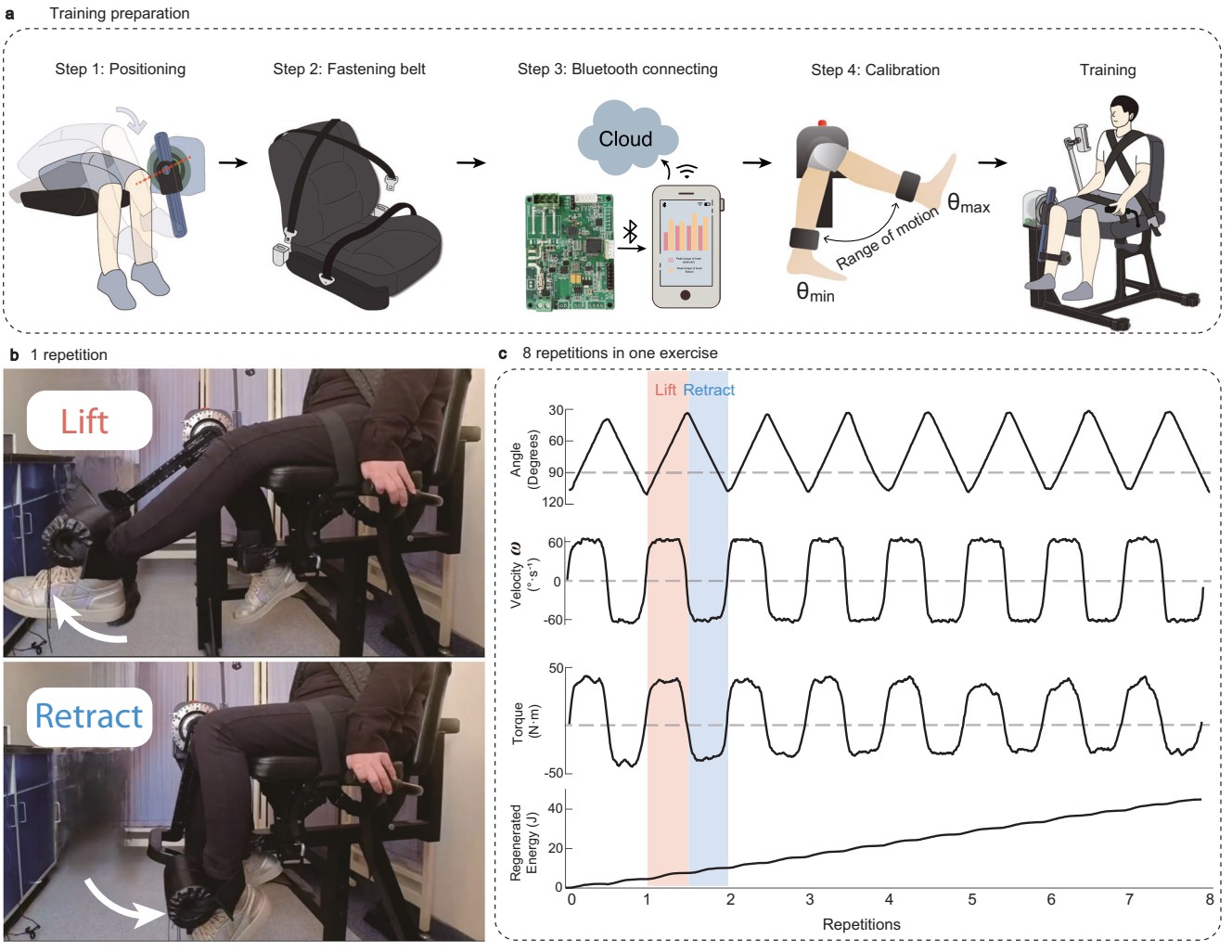

**Fig. 3 | Demonstration of isokinetic training. a** Preparation steps—Step 1: Position the leg and align the knee joint with the rotation axis. Step 2: Fasten the belt to stabilize the body. Step 3: Connect Bluetooth with the smartphone. Step 4: Calibration by testing the maximum and minimum angle of the subject. **b** Experimental pictures showing that a post-surgical subject lifts and retracts the calf during the training. **c** The training data of knee angle, velocity, torque, and regenerated energy over 8 repetitions of lifting-retracting movement. Source data are provided as a Source Data file.

## Primary outcomes-muscle strength

Following a 6-week rehabilitation program, there was not only a noticeable morphological change in the subjects but also a substantial enhancement in the muscle strength of both the quadriceps and hamstrings (see Supplementary Tables S6 and S7). This improvement was quantitatively demonstrated by the significant increase in the maximum torque values associated with lifting, and retracting movements observed in 10 post-surgical subjects, as presented in Fig. 6a. Specifically, subject 8 and subject 10 showed the most remarkable improvement, with their maximum lifting torque increasing by 140% and maximum retracting torque increasing by 223%, respectively. Subject 1 and Subject 2 exhibited the smallest gains in maximum torque, with increments of 22% and 23%, respectively. At the beginning of the rehabilitation program, during session 1, the average maximum torque of lifting and retracting among 10 subjects was merely $30.6 \pm 9.6$ and $24.6 \pm 10.4$ N m, respectively (Fig. 6b). Remarkably, by the conclusion of the rehabilitation program at session 12, these values had risen to $51.9 \pm 16.4$ and $45.1 \pm 8.6$ N m, respectively (Fig. 6c), which corresponds to an increment of 70% for lifting and 84% for retracting (Fig. 6d). During the entire experiment, no important harms or unintended effects (severe knee pain, swelling, inflammation, or inability to walk) were observed.

## Discussion

Conventional knee isokinetic training facilities are often limited by their substantial size, weight, and high-energy consumption. At the core of this limitation is the energy-intensive driver module, which is required to generate the controllable resistive torque. Existing driver modules rely on high-power servo motors[19–23], magnetorheological (MR) fluids[24], electrorheological (ER) fluids[25], and electromagnetic powder brakes[26]. All of these require continuous and precise electrical control—whether it involves supplying large currents, maintaining high voltages, or generating strong magnetic fields. These characteristics limit the rehabilitation training exclusively to clinical settings such as hospitals and rehabilitation centers.

The proposed power-free isokinetic robot can provide home-based knee rehabilitation without an external power supply. The power-free working mechanism is realized by the implementation of a controllable motor-winding short strategy. This strategy allows the device to harvest excessive mechanical energy generated by the user during exercise sessions, converting it into electrical energy that runs the system. Meanwhile, the controllable short motor-winding produces a controllable accommodative resistance torque. Using the velocity as closed-loop feedback, isokinetic training is realized.

This self-sustaining mechanism ensures that the robot can operate independently of external power sources, making it particularly

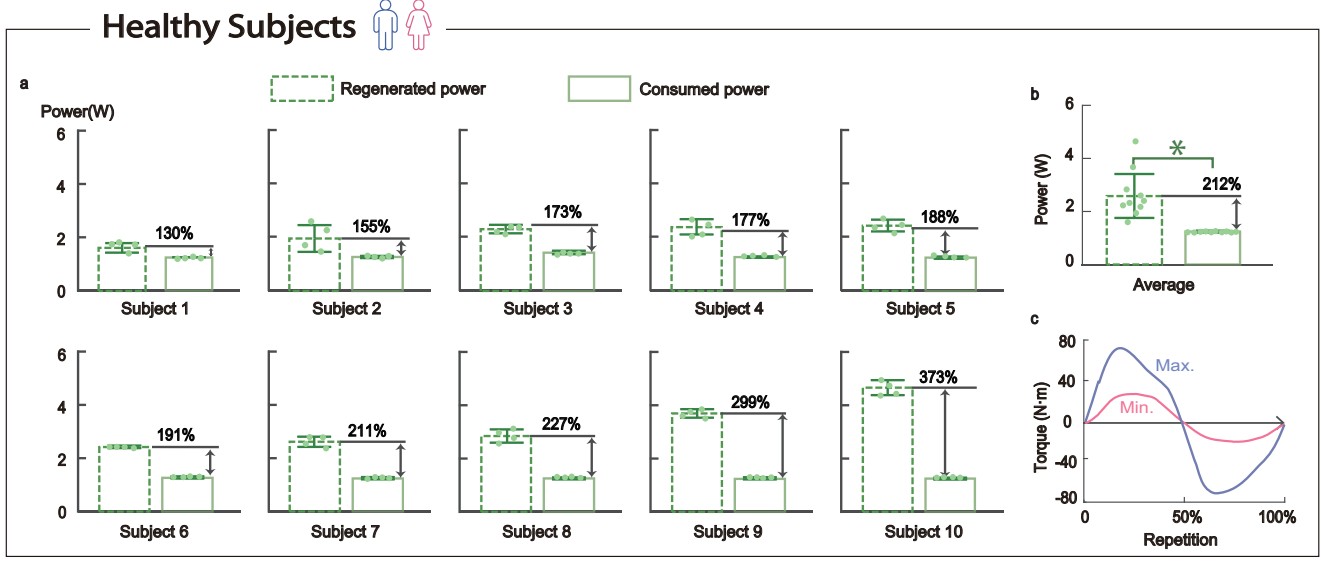

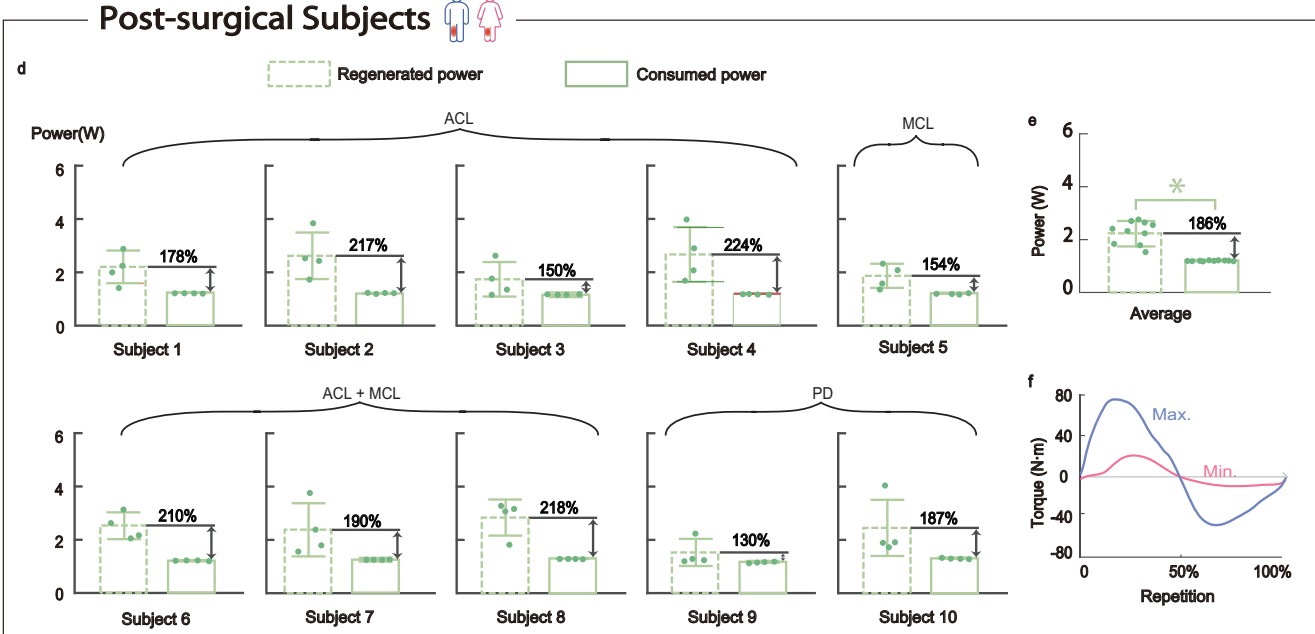

**Fig. 4 | Validation of power-free isokinetic training. a** Comparison between regenerated power and consumed power of 10 healthy subjects. **b** Comparison of the average power of healthy subjects ($n = 10$). $p = 4.84*10^{-5}$. **c** The maximum torque generated from subject 10 and minimum torque generated from subject 1 within one repetition of lifting-retracting movement. **d** Comparison between regenerated power and consumed power of 10 post-surgical subjects. ACL anterior cruciate ligament, MCL medial collateral ligament, PD patellar dislocation.

**e** Comparison of the average power of 10 post-surgical subjects ($n = 10$). $p = 2.01*10^{-7}$. **f** The maximum torque generated from subject 4 and minimum torque generated from subject 5 within one repetition of lifting-retracting movement. The dashed border represents the regenerated power, while the solid border represents the consumed power. Data are presented as mean ± SD. *Refers to *p*-value < 0.05 by two-tail *t*-test. Source data are provided as a Source Data file.

suitable for home-based use. The efficiency of this power-free isokinetic robot has been rigorously tested and confirmed through exercises involving both healthy individuals and post-surgical patients presenting a variety of knee conditions. These tests have not only validated the robot's operational viability but have also demonstrated its potential for more accessible and convenient rehabilitation at home.

The 6-week rehabilitation program demonstrates the capability of our robot not just for aiding in muscle growth but for substantially enhancing muscle strength. The variability in response among subjects also highlights the personalized nature of rehabilitation, suggesting that individual outcomes can significantly differ based on various factors, including initial muscle condition, commitment to the program, and biological response to rehabilitation exercises. The observed increases in both muscle morphology and muscle strength are a testament to the potential of targeted rehabilitation programs, especially those incorporating advanced robotic assistance, in achieving notable improvements after surgeries. This evidence advocates for the broader application and further development of such rehabilitation technologies, aiming for optimized recovery protocols tailored to individual needs.

### Efficacy comparison between our study and commercial isokinetic robots

To assess the rehabilitation outcomes, the quadriceps CSA growth rate (Fig. 7a) and lifting torque increase (Fig. 7b) of the proposed

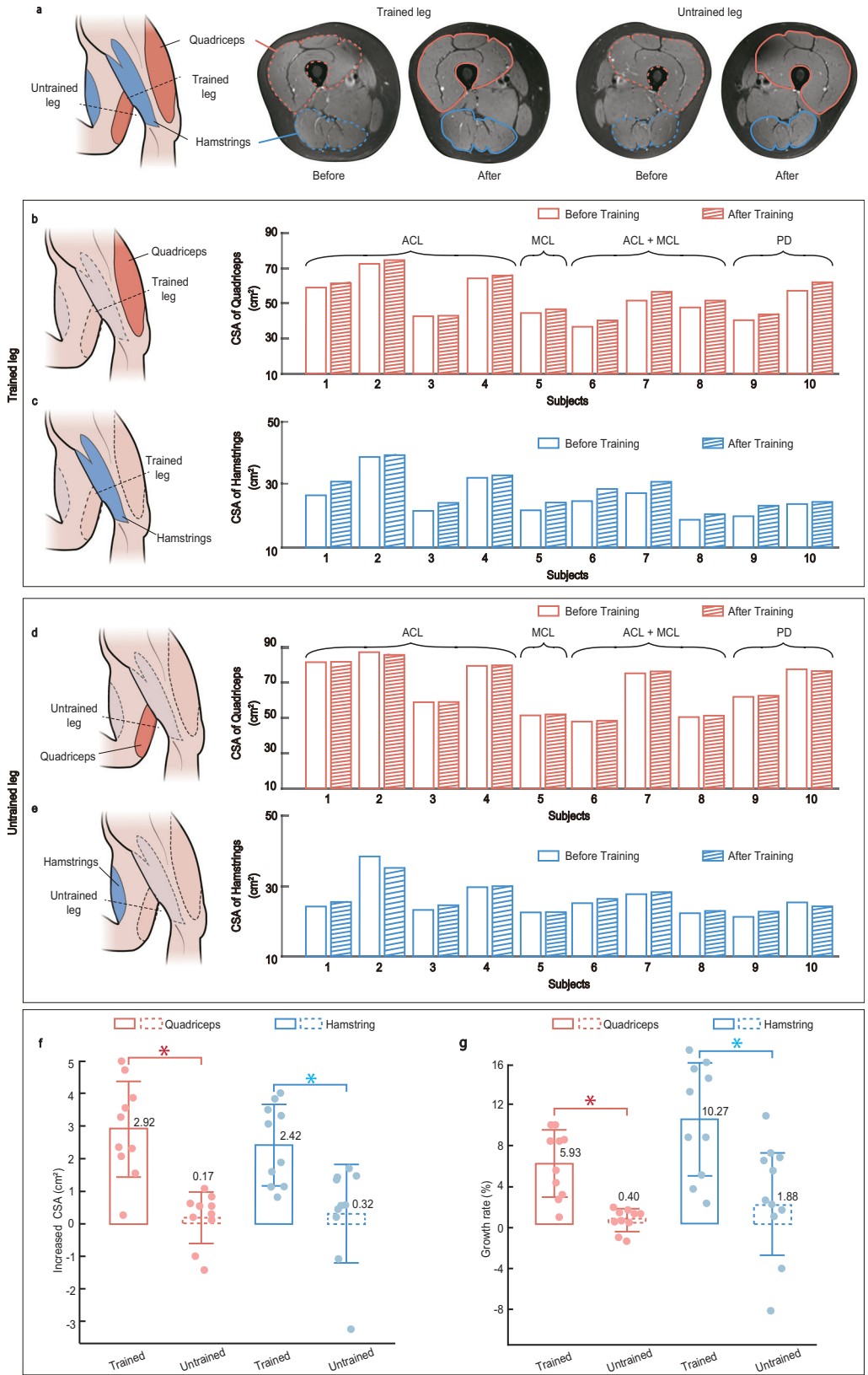

robots and references utilizing the commercial isokinetic training robots were compared. Notably, the CSA growth rate of the proposed power-free isokinetic training robot 5.9% (Fig. 7a) is much higher than the documented 3.7% value by a commercial rehabilitation device (CONTREX MJ, PHYSIOMED, Germany) where the same training program was carried out (33, see Fig. 7a). Quadriceps

muscle CSA in the untrained leg demonstrated a negligible difference in both the proposed power-free robots and commercial devices compared with the trained leg (Fig. 7a). This result may indicate that the impact of unilateral isokinetic training on the contralateral untrained leg is insignificant. Additionally, routine exercises (e.g., walking, jogging, climbing stairs) do not provide as

**Fig. 5 | Evaluation of muscle morphology after a 6-week rehabilitation program. a** MRI images of quadriceps and hamstrings in both trained and untrained legs before and after training. **b–e** Quantitative cross-sectional area (CSA) before and after training. The red bar represents the quadriceps, while the blue bar represents the hamstring. The empty bars represent the CSA before training, while the diagonally hatched bars represent the CSA after training. **b** Quadriceps of trained leg. **c** Hamstrings of trained leg. **d** Quadriceps of untrained leg. **e** Hamstrings of untrained leg. **f** Increased CSA of quadriceps/hamstrings in the trained/untrained leg. Quadriceps: $p = 3.03 \times 10^{-4}$. Hamstrings: $p = 6.88 \times 10^{-5}$.

(Subjects: $n = 10$) The solid border represents the trained leg, while the dashed border represents the untrained leg. **g** Growth rate of quadriceps/hamstrings in the trained/untrained leg. The solid border represents the trained leg, while the dashed border represents the untrained leg. Quadriceps: $p = 2.33 \times 10^{-4}$. Hamstrings: $p = 4.40 \times 10^{-6}$. (Subjects: $n = 10$) ACL anterior cruciate ligament, MCL medial collateral ligament, PD patellar dislocation. Data are presented as mean ± SD. *Refers to $p$-value < 0.05 by two-tail $t$-test. Source data are provided as a Source Data file.

pronounced stimulatory growth as isokinetic training. Similarly, the significant torque increases utilizing the power-free robots 70% is higher than the documented results of 35.9% utilizing the commercial isokinetic training device (*33*, see Fig. 7b). Other experiments conducted using commercial isokinetic devices with the same isokinetic training speed ($60° \cdot s^{-1}$) but varying training durations and cycles also did not demonstrate torque increases surpassing the results of our study[33,37–39], see Fig. 7b).

## Limitations

Firstly, our proposed power-free isokinetic training robot is primarily suitable for patients in the middle to late stages of rehabilitation. In other words, the device is not intended for patients in the early postoperative, bedridden stages, and they need to possess basic walking abilities. Secondly, our current design is specifically targeted at post-knee surgery rehabilitation training. Knee rehabilitation primarily focuses on training the quadriceps muscle and hamstrings muscle, which are major muscle groups in the thigh. As for isokinetic training of other joints, such as the ankle, elbow, and shoulder joints, whether the proposed isokinetic training robot can be approved power-free has not yet been experimentally validated. Third, the present system supports only concentric isokinetic contractions. Future work will aim to incorporate additional training modes, such as eccentric contractions, to broaden its rehabilitative capabilities. Fourth, the study employed a uniform training protocol for all post-surgical subjects. Customizing training loads based on individual clinical conditions could potentially enhance rehabilitation outcomes and will be explored in future research. Fifth, future studies will explore the use of varying angular velocities for antagonist muscle groups during the same exercise, such as applying angular velocities of $60°\cdot s^{-1}$ for the knee extensors and $90°\cdot s^{-1}$ for the knee flexors. Finally, as we are currently in the experimental validation phase, we can only assess the manufacturing costs (see Supplementary Table S8). Due to uncertainties in marketing costs, including factors such as marketization, it is challenging to compare the costs of our device with the sale prices of commercial isokinetic training devices.

## Methods
### Clinical trial

This intervention study (https://trialsearch.who.int/Trial2.aspx?TrialID=ChiCTR2300076715, ChiCTR2300076715), approved by the Biomedical Ethics Review Committee of West China Hospital, Sichuan University (No. 20231559), Peking University Third Hospital (No. 202349204) and Beihang University (No. BM20230120) is a non-blinded trial designed to evaluate muscle morphology and strength improvements in post-surgical subjects following 6 weeks of isokinetic training using the proposed isokinetic training robot. This is a randomized trial, collaboratively designed by Beihang University, Peking University Third Hospital, and West China Hospital of Sichuan University. Note that the experiment procedure of our study is the same as ref. 33, i.e., duration, evaluation methods (CSA and isokinetic muscle strength), and training intensity. The data reported in this study concerns "Part I" of the aforementioned clinical trial. All subjects underwent the same training protocol, wherein only the injured leg was trained (experimental group), while the healthy leg remained

untrained (control group). There were no incidents of adverse events or subjects withdrawals during the experiment, therefore the method employed from the start of the clinical trial remained unchanged. Subject selection was performed using a computer-generated random number method, from which 10 out of 300 post-surgical subjects were randomly selected. Once the subject was chosen, the experiment group (trained leg) and the control group (untrained leg) were pre-determined. All subjects underwent the same experimental procedure. Interim analyses were conducted after half training sessions.

### Isokinetic program

To evaluate the effectiveness of the proposed power-free isokinetic training robot, we made a comparison between the proposed isokinetic training robot and a commercial isokinetic training device, performing identical experiments on patients.[33] The post-surgical subjects participated in sessions twice a week for six weeks, with each session conducted at a velocity of $60° \cdot s^{-1}$. Each session comprised four exercises, and each exercise consisted of eight repetitions. Before and after the experiment, all subjects underwent bilateral thigh MRI scans using a 3 T Magnetom Trio, A Tim System (Siemens Medical Solutions, Erlangen, Germany). The CSAs of the quadriceps and hamstrings were quantified utilizing medical imaging software (RadiAnt DICOM Viewer). The isokinetic muscle strength was assessed using the proposed power-free isokinetic training robot, which features an integrated torque sensor (JH-NJLF, China). The primary outcomes include CSAs of both the trained leg and untrained leg, as well as the isokinetic muscle strength of the trained leg. The secondary outcomes encompass self-assessment of recovery, walking test, knee pain, and self-assessment of knee stability. All secondary outcomes are subjective evaluations, intended to assist patients in assessing whether they experience any abnormal or uncomfortable conditions. During the experiment, no such abnormal or uncomfortable situations were reported. Therefore, no separate discussion of the secondary outcomes is provided. Different from the post-surgical subjects, the healthy subjects only encountered one session.

Prior to the training of each session, several training preparation steps were undertaken (Fig. 4a): Firstly, the knee joint was aligned with the rotational center to ensure the synchronized lifting and retracting of the calf and the main arm. Secondly, the belts were secured to immobilize the upper body of the subject to the greatest extent feasible. Thirdly, the control circuit and the human-machine interface were activated. The control circuit communicates with a human-machine interface via a Bluetooth module, enabling the real-time transmission of collected training data to a secure cloud service for analysis by a skilled medical practitioner. Lastly, subjects were instructed to lift and retract their calf to the maximum position, with the angles of the position being recorded.

### Subjects

The 10 post-surgical subjects are 6 women and 4 men with an average age of $31.7 \pm 9.7$ years, height of $167.4 \pm 4.8$ cm, and weight of $67.4 \pm 12.7$ kg (see Supplementary Movie S3). All subjects were fully informed about the experiment prior to participation, and written informed consent was obtained from all subjects. The inclusion criteria were: age 18–75 years; have done knee surgery before; self-reported

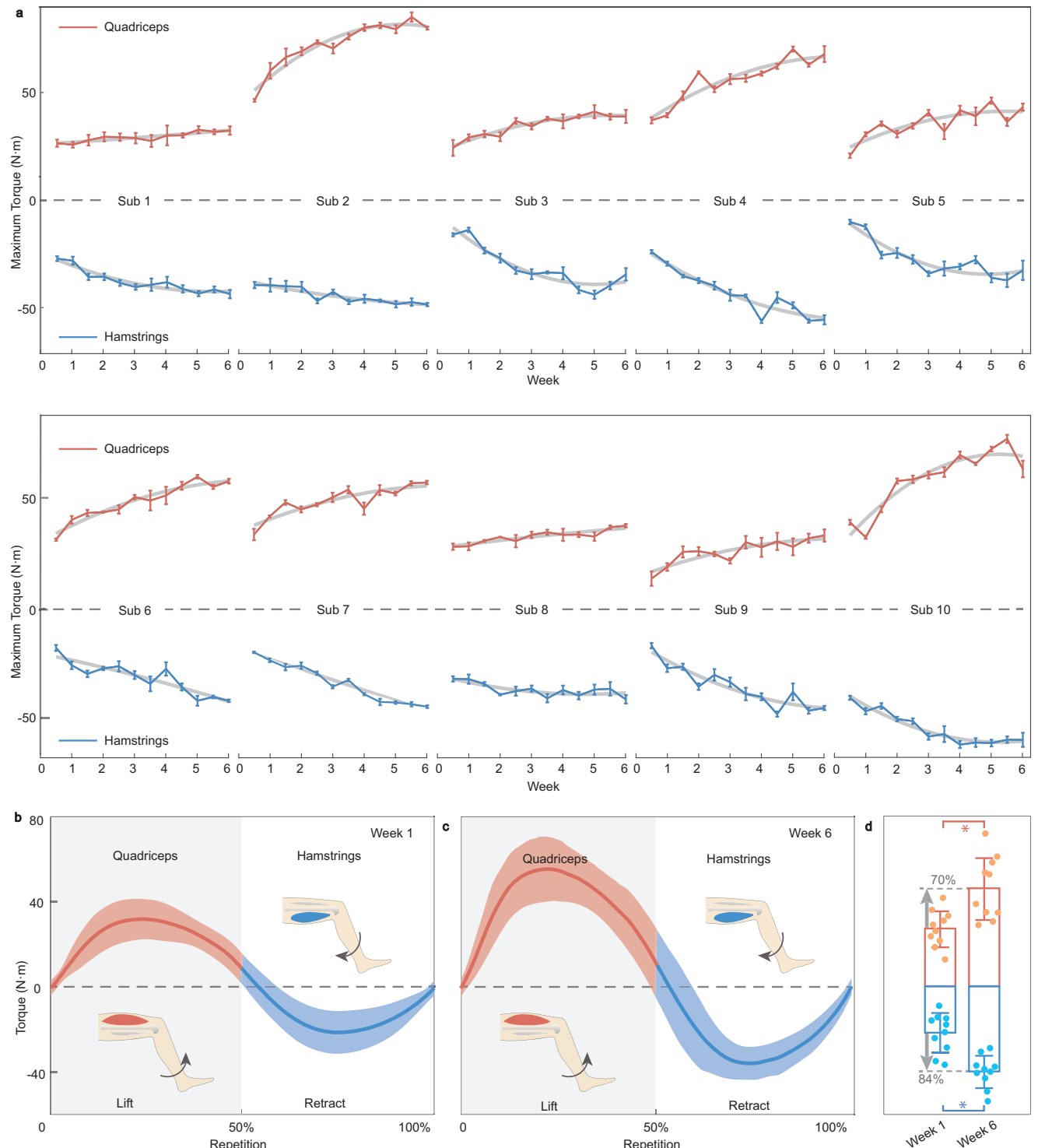

**Fig. 6 | Evaluation of muscle strength after a 6-week rehabilitation program.**
**a** The maximum torque of 10 subjects within 6 weeks. Positive values represent lifting torque by quadriceps and negative values denote retracting torque by hamstrings. **b** The maximum torque of 10 post-surgical subjects in one repetition in the first session. **c** The maximum torque of 10 post-surgical subjects in the last session. **d** Comparison of averaged maximum torque in the first and last session ($n = 10$). The left and right bars indicate torques in the first and last training sessions, respectively, with red bars representing lifting and blue bars for retracting. The percentage values indicate the increase from the first to the last session. The red dots, lines, and bars represent quadriceps muscle strength, while the blue dots, lines, and bars represent hamstring muscle strength. Data are presented as mean ± SD. *Refers to $p$-value < 0.05 by two-tail $t$-test. Source data are provided as a Source Data file.

reduction in thigh muscle strength; BMI < 35. The exclusion criteria were: patients with diseases that cause pain or dysfunction of the lower limbs (lumbar disc herniation, Parkinson's, Alzheimer's, etc.); unable to walk short distances; severe cognitive impairment and communication

disorders. Within this group, four subjects have impaired anterior cruciate ligament (ACL), two have patellar dislocation (PD), one has medial collateral ligament (MCL) injury, and the remaining three have combined ACL and MCL injuries. Exclusion, loss to follow-up,

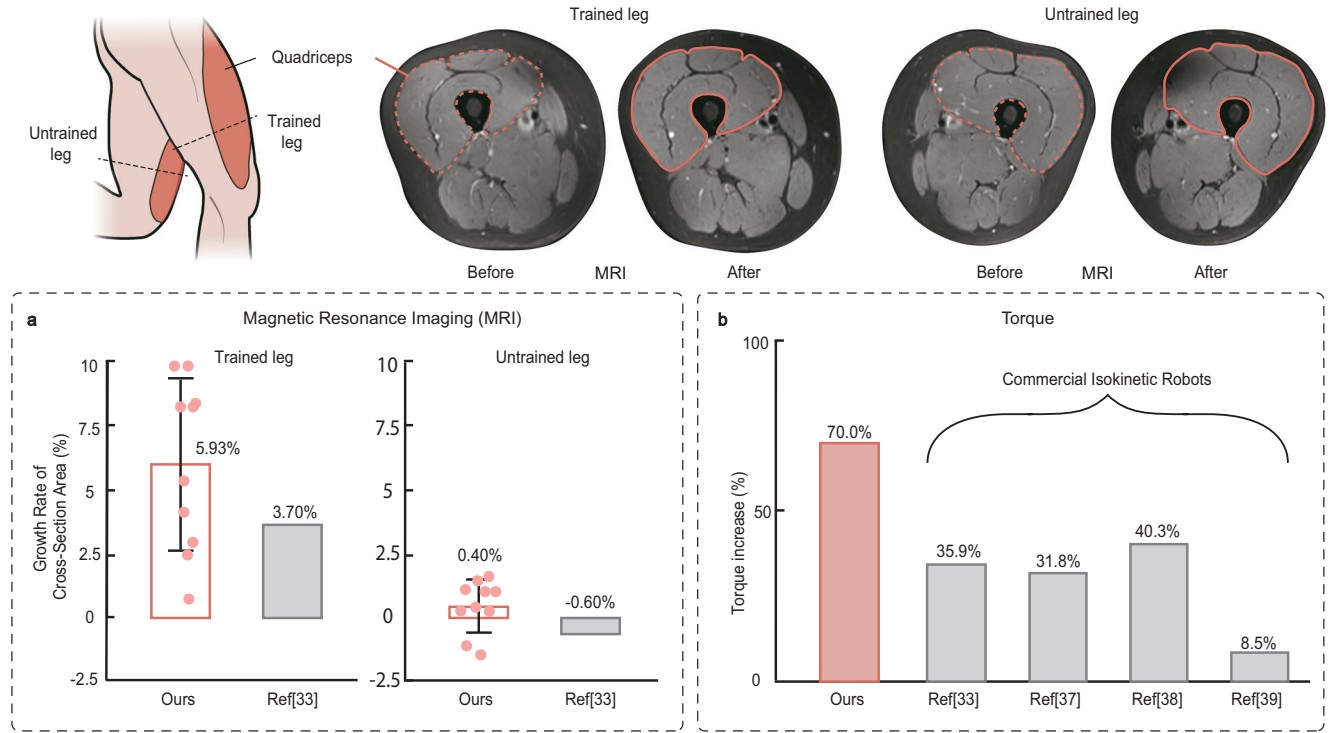

**Fig. 7 | Efficacy comparison between our study and commercial isokinetic robots. a** The CSA growth rate. Ours: Subjects, $n = 10$. **b** The torque increases. The red bar represents data from our study, while the gray bars represent data from the references. Ours: Subjects, $n = 10$. Source data are provided as a Source Data file.

confounding, termination, and suspension did not occur in this study. All subjects were recruited between November 2023 and February 2024. Besides clinical trials involving post-surgical subjects, the study also included 10 healthy subjects, including 5 women and 5 men with an average age of $22.8 \pm 2.7$ years, height of $170.2 \pm 9.8$ cm, and weight of $65.8 \pm 15.9$ kg.

## Procedure of isokinetic training

All demonstration experiments involved four steps as demonstrated in Fig. 3a. The first step is positioning the leg to ensure that the main arm of the robot is parallel to the calf, which allows for the precise measurement of the knee joint angle $\theta$[40]. The second step is fastening the belt to stabilize the body. This stabilization minimizes the movement of the hip joint, isolating the knee for target training only. The third step is connecting to the smartphone via Bluetooth, establishing the human-machine interface. The fourth step is calibration. The subject lifts and retracts his calf to the maximum and minimum angles, establishing a range of motion benchmark for the training protocol. In all trials, a training speed of $60° \cdot \text{s}^{-1}$ was used. Previous studies show that $60° \cdot \text{s}^{-1}$ is effective in various rehabilitation aspects, contributing to improvements in muscle strength and functional recovery[41,42]. Additionally, the proposed device is capable of accommodating different speeds for isokinetic training, such as $90° \cdot \text{s}^{-1}$ and $120° \cdot \text{s}^{-1}$.

## Energy regeneration during isokinetic training

The proposed dynamic energy regeneration method enables power-free functionality and isokinetic training, simultaneously. The knee joint is regarded as a bio-mechanical system comprising two skeleton bones, the femur, and the tibia, as well as two antagonistic muscles, quadriceps femoris (dominating knee lifting) and hamstring muscle (dominating knee retracting), as shown in Supplementary Fig. S4a. In this system, after the gravitational compensation, the torque $T$ from human dynamics is approximately

modeled as

$$T = F_{\text{s}} \cdot l_{\text{s}} \approx T_{\text{m}} \cdot \beta \cdot \widetilde{W_{\text{l}}}(\theta) \cdot \widetilde{W_{\text{v}}}(\dot{\theta}) \tag{1}$$

where $F_{\text{s}}$ is a force exerted by the trainee, $l_{\text{s}}$ is the corresponding moment arm. In detail, $F_{\text{Q}}$ and $l_{\text{Q}}$ represent quadriceps femoris contraction, $F_{\text{H}}$ and $l_{\text{H}}$ represent hamstring muscle contraction, $T_{\text{m}}$ is the maximum torque that muscle can exert, $\beta$ ($0 \le \beta \le 1$) is a normalized muscle activation level, $\widetilde{W_{\text{l}}}(\theta)$ is a torque-angle factor and $\widetilde{W_{\text{v}}}(\dot{\theta})$ is a torque-velocity factor. The detailed principle is shown in Supplementary Fig. S5 and Supplementary Information Text.

If the motor windings were fully shorted, indicating without any control and all low-side MOSFETS are on (Fig. S4b), the maximum current $I_{\text{max}}$, due to the torque $T$,

$$I_{\text{max}} = \frac{T_{\text{m}} \cdot \beta \cdot \widetilde{W_{\text{l}}}(\theta) \cdot \widetilde{W_{\text{v}}}(\dot{\theta})}{k} \tag{2}$$

where $k$ is the motor torque constant.

If the duration of the short motor windings was controlled dynamically (see Supplementary Fig. S4b), as duty cycle $D$, the accommodative resistance torque $\tau_{res}$ is

$$\tau_{res} = kI_{\text{a}} = kf(D) \le kI_{max} \tag{3}$$

where $I_{\text{a}}$ is the phase current, which is associated with $D$, the maximum $I_{\text{a}}$ is equal to $I_{\text{max}}$.

To realize isokinetic training, a proportional-integral (PI) controller is to adjust $D$ based on errors,

$$D = K_{\text{p}}(\dot{\theta} - \bar{\dot{\theta}}) + K_{\text{I}}(\dot{\theta} - \bar{\dot{\theta}}) \tag{4}$$

where $K_{\text{p}}$ is a proportional coefficient, $K_{\text{I}}$ is an integral coefficient and $\bar{\dot{\theta}}$ is the desired velocity.

Accompanying isokinetic training, dynamic energy regeneration can be obtained, and the regenerative power is

$$P_{\text{reg}} = \frac{\left(\int_0^{t_2} I_{\text{charge}} dt\right)^2 - \left(\int_0^{t_1} I_{\text{charge}} dt\right)^2}{2C(t_2 - t_1)} \qquad (5)$$

where $I_{\text{charge}}$ is the charging current. When low-side MOSFETs are off, $I_{\text{charge}}$ is equal to $I_a$, when low-side MOSFETs are on, $I_{\text{charge}}$ is 0 (Fig. S5b), $C$ is the capacitance and $t_1$ and $t_2$ are different timings. The detailed charging circuit principle is shown in Supplementary Figs. S6 and S7.

## Statistical methods

All statistical analyses were conducted on data collected from all subjects and performed with the SPSS 25.0 statistical package (IBM, Armonk, NY, USA), and Microsoft Excel. The significant difference of the data is analyzed using a two-tailed $t$-test. We consider $p$-values of less than 0.05 to be statistically significant. All data were analyzed independently, and shown as mean ± SD.

## Human–machine interface

For the convenience of rehabilitation, we developed a home-based isokinetic training robot to offer a real-time human–machine interface. Its operational framework includes several stages. Initially, a custom application was developed to collect training data in real-time via a Bluetooth module. Subsequently, this amassed training data is instantaneously transmitted to a secure cloud service, facilitating seamless and immediate analysis. Following this, a proficient medical practitioner evaluates the patient's adherence to the training regimen, assessing their training performance and providing insightful recommendations. This feedback loop completes as the doctor's assessments and recommendations are relayed back to the patient, fostering a continuous cycle of communication. Noteworthy is the dynamic nature of this engagement, as the received feedback prompts the fine-tuning of the training schedule, adapting it precisely based on the doctor's assessments. This holistic approach underscores the patient-centric design of the robot, harmonizing technological innovation with medical expertise to optimize patient well-being.

## Mechanical design

As shown in Fig. 2a, in the motor assembly, the main component is the LSG-32-142 brushless direct current (BLDC) motor, which is fixed on the motor base at the side of the seat via screws. The central axis, which is inserted into the central hole of the motor, is stabilized with the joint adapter. An adapter plate bridges a torque sensor to the joint adapter. For safety concerns, a limit plate is mounted on the motor base to ensure that movement is within an allowable range. Additionally, an angle sensor, secured on the mounting plate at the tail of the motor, registers movement information (angular velocity) of the central axis (motor rotor). The detailed mechanical assembly is shown in Supplementary Fig. S1.

As shown in Fig. 2b–h, the control circuit comprises a power supply, a microcontroller unit (STM32F4), and an energy regeneration circuit. The microcontroller unit, communication circuit (Bluetooth, HC05), energy regeneration circuit, and current/voltage sensor components are integrated into the same board. In this study, the BLDC motor operates as a generator, transforming kinetic energy into electrical energy. The regenerated energy is then stored in an ultracapacitor (40 F, 5.5 V), ready to be utilized for isokinetic training control, sensing, and communication.

A fixture assembly consists of an adapter sleeve, a main arm and an ankle fixture (Fig. 2a, Supplementary Fig. S1). The adapter sleeve merges the main arm with the output end of the motor assembly. Uniquely, the main arm can slide relative to the adapter sleeve,

adjusting its effective length for different calf lengths. The ankle fixture secures the ankle to the main arm, ensuring that calf movement drives the main arm's movement, with torque transmitted to the motor assembly. Notably, the ankle fixture is engineered to be interchangeable for either the left or right leg.

The base serves as the foundational support for the entire structure of our isokinetic training robot, engineered with pivotal features to enhance user convenience and adaptability. It enables the seat angle to be adjusted, ensuring the seat is precisely aligned to accommodate the user's body posture, thereby providing a tailored training experience. Furthermore, the structure is designed to be disassembly-friendly and incorporates universal wheels, enabling the robot with notable portability. This design ensures that the isokinetic training robot is not only easy to reposition but also straightforward to transport, allowing for versatile usage across various locations.

## Charging principle

The circuit, consisting of three phases a–c, includes equivalent resistors $R_a$, $R_b$, and $R_c$, equivalent inductances $L_a$, $L_b$, and $L_c$ and the back electromotive forces (EMFs) $E_a$, $E_b$, and $E_c$.[43–45] When the MOSFETs S2, S4, and S6 are switched ON, the back EMFs and currents in phases a and b are expressed as $E_{a_{on}}$, $E_{b_{on}}$, $I_{a_{on}}$, and $I_{b_{on}}$ respectively. For more details, refer to Supplementary Figs. S7 and S8 and Eqs. (S14)–(S22). The average charging current $\overline{I_{\text{charge}}}$, can be expressed as follows:

$$\overline{I_{\text{charge}}} = \begin{cases} 0 & \text{Case1.1} \\ \frac{(E_b - U_a)e^{-\frac{t}{RC}}}{R_a + R_b} & \text{Case1.2} \\ \frac{E_b - E_a - (1-D)((\overline{U_{uc}} + 2U_{FD})))}{2(R_a + R_s D)} & \text{Case2} \\ \left(0, \frac{E_b - E_a - ((1-D)((\overline{U_{uc}} + 2U_{FD})))}{2(R_a + R_s D)}\right) & \text{Case3} \\ \frac{E_b - E_a - (1-D)((\overline{U_{uc}} + 2U_{FD})))}{2(R_a + R_s D)} & \text{Case4} \\ 0 & \text{Case5} \end{cases} \qquad (6)$$

where $E_b - E_a = k_{ge}\dot{\theta}$, $k_{ge}$ is the generator constant of the motor (V/rpm).

## Speed control

The motor damping is controlled by adjusting the duty cycle of PWM. A large duty cycle gives a large damping, resulting in a low motor speed, and conversely, a short duty cycle results in small damping and a high motor speed. Consequently, by adjusting the duty cycle of PWM, the isokinetic (constant speed) training was realized. To adjust the duty cycle of PWM, a proportional-integral (PI) closed-loop control was used.

## Reporting summary

Further information on research design is available in the Nature Portfolio Reporting Summary linked to this article.

# Data availability

All data supporting this study, including primary outcome data (CSAs, muscle strength) and data used for figures, are provided in the Supplementary Source Data files. Additional data can be found in Supplementary Tables S1–10. Data from subjects, which are deidentified and thus non-identifiable, are presented as ranges rather than specific values in Supplementary Tables S2 and S3 to ensure anonymity. The study protocol and statistical analysis plan are available in the Supplementary files. All data from this study have been publicly available since the publication date and can be downloaded. For additional data requests, readers are encouraged to contact the corresponding author, Yanggang Feng (email: yanggangfeng@buaa.edu.cn). Source data are provided with this paper.

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

## Acknowledgements

We extend our gratitude to Prof. Qining Wang, Prof. Li Wen, Prof. Steven Hartley Collins, and Prof. Yan Huang for their invaluable suggestions, which greatly enhanced the quality of this study. This work was supported by the National Key Research and Development Program of China [2023YFB4705100; 2022YFB4701200], the National Natural Science Foundation of China [T2121003;12388101; 12272369], the Fundamental Research Funds for the Central Universities [YWF-23-L-1205], and the "Scientific Innovation Yongjiang 2035" Key Technology Breakthrough Plan Projects in Ningbo City [2024Z199].

## Author contributions

Conceptualization: Y.G.F; Methodology: Y.G.F., H.Y.W; Investigation: Y.G.F., H.Y.W., Y.M.F.Z., M.W.Z., and K.L.; Clinical methodology: Y.G.F., F.Z.Y., J.W., and Y.N.; Visualization: Y.G.F., J.X.R., X.J., X.H.L., X.Y.H., H.X.J., Y.B.L., Y.H.Z., Z.Y.W., X.Z.L., J.J.X., and Y.X.S.; Funding acquisition: Y.G.F., F.Z.Y., and X.L.D.; Project administration: Y.G.F., W.X.Z., L.W., and X.L.D.; Supervision: Y.G.F., W.X.Z., L.W., and X.L.D.; Writing—original draft: Y.G.F., H.Y.W., and J.X.R.; Writing—review & editing: Y.G.F., H.Y.W., J.X.R., Q.S., and L.W.

## Competing interests

The authors declare no competing interests.
