## [Transparent Peer Review file · Nature Communications]

Power-Free Knee Rehabilitation Robot for Home-Based Isokinetic Training

Corresponding Author: Professor Liu Wang

Version 1:

Reviewer comments:

Reviewer #1

(Remarks to the Author)

MANUSCRIPT TITLE: Highly Integrated and Power-Free Knee Rehabilitation Robot for Home-Based Isokinetic Training
GENERAL EVALUATION

The objectives of this study were (1) to present an integrated isokinetic knee rehabilitation robot for home-based isokinetic training, (2) test the device with 10 post-surgical subjects and 10 healthy individuals, and (3) carry out a 6-week interventional clinical trial with the 10 patients after knee surgeries and validate recovered knee functions after training using the robot. The manuscript's theme is adequate for publication in Nature Communications, is clinically relevant to the area of knee home-based rehabilitation and has novelty for the specific area of assistive technologies for knee rehabilitation.

According to the authors, the main findings of the study were that the muscle growth and strength improvements achieved with the robot outperformed existing commercial rehabilitation devices, thereby indicating that the robot presents a viable option for home-based knee rehabilitation, significantly enhancing the accessibility of effective treatment. The reviewed literature is partially adequate and updated. The authors correctly conducted the reader through the possible limitations of existent isokinetic devices and a possible way to resolve these problems through a new device. However, they did not present in the introduction recent results from previous studies regarding the use of isokinetic devices in knee rehabilitation, what is missing in the area and how this new device will achieve similar/better results when used clinically than the exercises that can be done without the need of any equipment. Ethical aspects regarding the clinical trial were correctly presented in the manuscript. The methods section is well described, but some aspects are not completely clear (see Specific Comments). The results need to be reorganized, as several sections are more related to the Methods Section, not to the Results. In the discussion section, the authors clearly described the device features, described several results that were obtained with the robot, but did not adequately discuss their results while contrasting them with the existent literature. Finally, I would like to congratulate the authors on the excellent new technology that they presented very well and that may indeed help in the rehabilitation of patients with knee problems, and I hope that my constructive criticism helps to improve the manuscript and the rationale for the use of this new technology in clinical practice.

SPECIFIC COMMENTS

TITLE

The words "highly integrated" in the title do not allow the reader to understand with what exactly the rehabilitation robot is integrated with.

ABSTRACT

It would be interesting to see some results in the abstract so that the reader does not have to just believe in what you are saying that happened, but actually can see what happened.

INTRODUCTION

Line 51. I suggest that you use more recent studies to support what you are stating here, as reference 8 is among the earliest studies in the isokinetic area.

Line 59. Please avoid the use of "etc" and list the most important external hardware.

Why would an isokinetic device be better than closed kinetic chain exercises involving body weight (e.g., squatting, single-leg squat) or than exercises with free weights for the home-based knee rehabilitation program? There is no or little cost and we may get similar structural and functional gains post-training. In other words, you need to convince the reader of why your device is better.

In addition, it would be important for the reader to get an idea at the introduction about previous literature on the use of isokinetic devices for knee rehabilitation and the limitation of these devices for home-based rehabilitation. How many studies have been conducted to rehabilitate knee patients using isokinetic devices? Is there any study that used isokinetic

devices in home-based knee rehabilitation? What is the state of the art regarding isokinetic knee rehabilitation, and which are the main possible advances your new device may bring?

RESULTS

Line 95. Please better define what you mean by the words “Highly integrated” in the subtitle. Are the other isokinetic devices not highly integrated?

Lines 125-155. This section on “Energy regeneration during isokinetic training” appears to me out of place, as it does not show any result, but explains how the system regenerates energy during the isokinetic training, and therefore should be moved to the Methods Section.

Lines 157-178. I would separate this section into two different sections: (1) In lines 158-172 you are describing the methodology used to position the subject in the device and the procedures for executing the isokinetic knee extension-flexion exercises. Therefore, this belongs to the Methods section. (2) In lines 172-178 you show results that illustrate the functioning of the device and how the training is evaluated and how energy is regenerated. Therefore, this part belongs to the Results section.

Lines 168-169. Why did you choose the single angular velocity of 60°/s? In addition, what do you mean by “optimizing the training outcomes”? It would be interesting if your device had the possibility of establishing different angular velocities allowing setting these different angular velocities for each antagonist muscle group. According to the torque-velocity relationship of concentric contractions, the lower the velocity the higher is the muscle force generating capacity. In this way you would be able to establish different external loads and different mechanical work for each muscle group, thereby individualizing your training outcomes.

Lines 169-172. How do you know if the patient is exerting maximal knee extensor torque during the exercise if the device only provides the resistance to control the angular velocity of the patient’s limb? I imagine that the patient would be able to exert a knee extension faster than 60°/s, but not necessarily at maximal effort. In this case the system will limit the angular velocity but will not be able to determine if maximal effort was performed. In addition, determining the angular velocity of the knee extensor exercise into a specific velocity does not give you training intensity. To get the training intensity at a constant angular velocity you will need to determine the work performed over time or the area under each torque-velocity curve of the whole protocol, and changing the protocol time will allow you to change the training intensity.

Lines 197-198. Please provide some references giving support for this sentence regarding CSAs being widely adopted as the principal metric for assessing muscle growth and recovery.

Lines 217-232. As the knee extensors and the hamstrings have different muscle architecture and different (opposite) functions regarding the anterior/posterior tibial translation, and as the patients had different knee ligament tears, how do you define the best load for each muscle group from each patient? For example, for an ACL rupture patient, you may want to strengthen more the hamstrings to avoid tibial anterior translation, whereas for a PCL rupture the hamstrings may not be the best choice. And you have patients in your sample with different clinical problems regarding the knee ligaments that probably need different loads for each knee muscle group. Therefore, you need to be careful with the interpretation of your knee flexor and extensor increments, which should be evaluated critically in your discussion.

Lines 233-248. This part of the results seems to me more appropriate for the Discussion Section (not the Results Section), as here you are comparing and discussing your results with the results from previous literature on the theme.

DISCUSSION

Usually in the Discussion Section, in addition to reminding the reader about the goals of the study and showing a summary of the main findings, authors should contrast the obtained results with those from previous studies from the existent literature. Your discussion does not have a single article from previous literature in the area of existent devices and there is no contrast of your results with the results obtained with these isokinetic devices in knee rehabilitation. Similarly, it does not show any mention to previous home-based isokinetic training in clinical rehabilitation of patients with knee joint problems and/or ligament reconstruction. Part of this discussion was observed in different sections of the manuscript, and therefore I suggest that you do restructure the manuscript so that methodological aspects are presented in the Methods Section, results are presented in the Results Section, and discussion and contrasts with the literature are presented in the Discussion Section. Lines 276-288. I suggest that you add, in your limitations paragraph, all the other limitations, which I presented/discussed above and below, here in this final paragraph of the discussion.

METHODS

The technical aspects of the robot are well described. However, it is not clear in the methods section why a single angular velocity was determined for the isokinetic training robot to be used in the rehabilitation program. Is there a technical limitation, for example, in the energy regeneration system which needs that a single angular velocity is established for the system to work properly?

A second important aspect that needs to be indicated is that the robot only allows knee flexor-extensor concentric isokinetic contractions, which is a limitation of the device when compared to the commercially available isokinetic machines that were presented in the manuscript, as eccentric and isometric contractions can be performed in these isokinetic machines and are important contraction modalities that are clinically important for rehabilitation programs.

Why 10 patients? Have you performed a sample size calculation? If not, this needs to be added as a limitation in the discussion.

REFERENCES

Ok.

TABLES AND FIGURES

Fig. 3a. Please correct the word “angle” in the figure.

(Remarks on code availability)

(Remarks to the Author)

The experiment design and data analysis are generally appropriate. I have the following comments:

1. Please summarize the experimental data more systematically, with mean and standard deviation for trained and untrained legs, before and after the training.
2. Since there is a control group (untrained leg), the hypothesis to be tested should be the improvement/change in the trained leg is larger than that of the untrained leg.
3. Comparing Figs 6b and 6f, the CSA of quadriceps of the trained leg appears to be consistently smaller than the untrained leg. Please explain.
4. On line 345, it should be 'We consider p-values of less than 0.05 to be statistically significant'.

(Remarks on code availability)

Version 2:

Reviewer comments:

Reviewer #1

(Remarks to the Author)

The objectives of this study were (1) to present an integrated isokinetic knee rehabilitation robot for home-based isokinetic training, (2) test the device with 10 post-surgical subjects and 10 healthy individuals, and (3) carry out a 6-week interventional clinical trial with the 10 patients after knee surgeries and validate recovered knee functions after training using the robot. The manuscript's theme is adequate for publication in Nature Communications, is clinically relevant to the area of knee home-based rehabilitation and has novelty for the specific area of assistive technologies for knee rehabilitation. According to the authors, the main findings of the study were that the muscle growth and strength improvements achieved with the robot outperformed existing commercial rehabilitation devices, thereby indicating that the robot presents a viable option for home-based knee rehabilitation, significantly enhancing the accessibility of effective treatment. The reviewed literature is now adequate and updated. The authors correctly conducted the reader through the possible limitations of existent isokinetic devices and a possible way to resolve these problems through a new device. The authors now present at the introduction recent results from previous studies regarding the use of isokinetic devices in knee rehabilitation, what is missing in the area and how this new device will achieve similar/better results when used clinically than the exercises that can be done without the need of any equipment. Ethical aspects regarding the clinical trial were correctly presented in the manuscript. The methods section is well described, and now all aspects that were raised by the reviewer are completely clear. The results are now well organized, and several sections related to the methods were correctly placed in the Methods Section. In the discussion section, the authors clearly described the device features, described several results that were obtained with the robot, and now adequately discuss their results while contrasting them with the existent literature. Congratulations to the authors on the excellent work done in the revised manuscript and on the new technology that they presented very well and that may indeed help in the rehabilitation of patients with knee problems. The manuscript now demonstrates the rationale for the use of this new technology in clinical practice and can therefore be accepted for publication.

MINOR COMMENTS

I would add some of the information that you provided in the Rebuttal Letter for the Reviewers in the manuscript. For example, I would add this sentence at the introduction or at the discussion sections.

- We were unable to find any studies that utilized isokinetic training devices in home-based settings.

I would use the same units for the angular velocities throughout the text, and perhaps $60^{\circ} \cdot s^{-1}$ (with the -1 superscript) may be better.

Also, please add the range of angular velocities that the isokinetic training device can work in the Methods section and add information regarding if different angular velocities can be determined at the same exercise for the two antagonist muscle groups: knee extensors and flexors (e.g., can we set up the angular velocities of $60^{\circ} \cdot s^{-1}$ for the knee extensors and $90^{\circ} \cdot s^{-1}$ for the knee flexors?). You showed in your rebuttal letter results from three angular velocities, and how you set up the angular velocities for the two movements (knee extension vs flexion) needs to be better described.

(Remarks on code availability)

Reviewer #3

(Remarks to the Author)

Thanks for the thorough revision. I don't have further comments.

(Remarks on code availability)

**Responses to Comments on "Nature Communications-
#NCOMMS-24-18580A-Z"**

Dear Editor and Reviewers,

We would like to express our sincere gratitude for the opportunity to revise our manuscript entitled “*Highly Integrated and Power-Free Knee Rehabilitation Robot for Home-Based Isokinetic Training*”. We deeply appreciate the constructive and insightful feedback provided by the reviewers. In response, we have carefully addressed all of the comments point-to-point and made the necessary revisions. All comments from the reviewers are highlighted in purple, and the corresponding modifications in the revised manuscript are marked in blue. We believe these revisions have significantly strengthened the manuscript, and we hope that it now meets the high publication standards of *Nature Communications*.

Prof. Jian Wang from Peking University Third Hospital provided invaluable guidance in addressing the clinical aspects of the reviewers' comments. All authors have agreed to include him as a co-author in recognition of his contributions.

Thank you again for your valuable input and for considering our revised submission.

Best,
Liu Wang, Yanggang Feng

Reviewer #1:

General Comment:

The objectives of this study were

- (1) to present an integrated isokinetic knee rehabilitation robot for home-based isokinetic training,
- (2) test the device with 10 post-surgical subjects and 10 healthy individuals, and
- (3) carry out a 6-week interventional clinical trial with the 10 patients after knee surgeries and validate recovered knee functions after training using the robot.

The manuscript's theme is adequate for publication in *Nature Communications*, is clinically relevant to the area of knee home-based rehabilitation and has novelty for the specific area of assistive technologies for knee rehabilitation.

According to the authors, the main findings of the study were that the muscle growth and strength improvements achieved with the robot outperformed existing commercial rehabilitation devices, thereby indicating that the robot presents a viable option for home-based knee rehabilitation, significantly enhancing the accessibility of effective treatment. The reviewed literature is partially adequate and updated. The authors correctly conducted the reader through the possible limitations of existent isokinetic devices and a possible way to resolve these problems through a new device.

However, they did not present in the introduction recent results from previous studies regarding the use of isokinetic devices in knee rehabilitation, what is missing in the area and how this new device will achieve similar/better results when used clinically than the exercises that can be done without the need of any equipment. Ethical aspects regarding the clinical trial were correctly presented in the manuscript. The methods section is well described, but some aspects are not completely clear (see Specific Comments). The results need to be reorganized, as several sections are more related to the Methods Section, not to the Results. In the discussion section, the authors clearly described the device features, described several results that were obtained with the robot, but did not adequately discuss their results while contrasting them with the existent literature. Finally, I would like to congratulate the authors on the excellent new technology that they presented very well and that may indeed help in the rehabilitation of patients with knee problems, and I hope that my constructive criticism helps to improve the manuscript and the rationale for the use of this new technology in clinical practice.

Our response

We sincerely appreciate your high regard for the scientific contribution of our work. Your insightful and constructive comments are invaluable, and we have made our best effort to address the comments. The additional content included in this revision offers

a more detailed and comprehensive description of the home-based, power-free isokinetic training robot. We hope that the revised content addresses your concerns and provides a more thorough overview of this study.

(1) Response to “Recent results from previous studies regarding the use of isokinetic devices in knee rehabilitation”.

Thank you for the professional comment. To the best of our knowledge, there are only a few studies on the design of isokinetic training robots. Most studies on isokinetic training focus on utilizing the bulky, heavy, energy consuming, and expensive commercial isokinetic devices to study their clinical applications, demonstrating the effectiveness of isokinetic devices in the diagnosis, assessment, and rehabilitation of various knee injuries (16-18). To the best of our knowledge, we have outlined the **most recent results for achieving isokinetic training** in the manuscript.

In the second paragraph of the INTRODUCTION, we added:

“Isokinetic training is usually administrated by a specialized rehabilitation facility that applies resistive forces for controlled movement of the injured knee. Existing commercial facilities, such as Biodex (19), Kineo (20), CONTEX MJ (21), IsoMed2000 (22), and research prototypes represent the state-of-the-art in isokinetic training technology (23-26) (**Supplementary Table 1**). These systems employ various operating principles in their driver modules, including high-power servo motors (19-23), magnetorheological (MR) fluids (24), electrorheological (ER) fluids (25), and electromagnetic powder brake (26) (**Fig. 1a**). ”

Table S1 | Comparison between existing hospital-based and our home-based rehabilitation robot

	Research	Example/Picture	Power	Weight	Size	Human-machine Interface	Price
Commercial Product (Hospital-based)	Biodex ^[19]		Immovable (230V, 3450W)	612 kg	2.92 (m ²)	22" flat panel touchscreen	\$57,765
	Kineo ^[20]		Immovable (230V, 1600W)	345 kg	2.79 (m ²)	2×15.6" touch-screen displays	\$49,500
	CONTREX MJ ^[21]		Immovable (230V, 2300W)	350 kg	1.48 (m ²)	Display	Est. \$50000
	IsoMed2000 ^[22]		Immovable (380V, 8000W)	650 kg	2.6 (m ²)	Display	Est. \$50000
This Study (Home-based)			Movable (Power-free)	52 kg	0.82 (m ²)	6.67" Smart Phone	<\$1,000

Fig. 1 | Proposed power-free knee rehabilitation robot, realization and comparison. (a) Existing hospital-based rehabilitation devices. These devices are bulky and need additional power supplies. (b) Proposed home-based isokinetic training robot. The proposed isokinetic training robot is power-free and highly integrated. (c) Explosion Assembly. During the training, the rotation center of the knee joint and the motor are concentric. (d) Isokinetic realization and energy regeneration. The isokinetic training robot can generate energy while training. A desired constant velocity is predetermined. The velocity is monitored via angle sensors and the motor output resistance is modulated to ensure the actual velocity closely approximates the predetermined desired velocity. The controlled resistance is adjusted and the negative mechanical work during training is harvested. The harvested energy is subsequently channeled to power the control circuit, sensors, and other usage. (e) Comparison. A comparison between commercial isokinetic training robots and our proposed isokinetic training robot. The consumed power and weight of our robot are both dramatically reduced.

Reference:

19. Biodex System 4 Pro™, Biodex Medical System; <https://www.biodex.com/physical-medicine/products/dynamometers/system-4-pro>.
20. Kineo LegPro, Kineo System Intelligent Load; <https://www.kineosystem.com/range/kineo-leg-pro/>.
21. CON-TREX MJ, PHYSIOMED, <https://physiomed.de/en/produkt/con-trex-mj/>.
22. IsoMed2000, D&R, http://www.isomed2000.de/comp.gen.intro.php?lc=en_cn.
23. Hu B. et.al, Disturbance Rejection Speed Control Based on Linear Extended State Observer for Isokinetic Muscle Strength Training System, *IEEE Trans. Autom. Sci. Eng.* **20**, 1962-1971 (2023).
24. Dong S., Lu K., Sun J., Rudolph K., Adaptive force regulation of muscle strengthening rehabilitation device with magnetorheological fluids. *IEEE Trans. Neural Syst. Rehabil. Eng.* **14**, 55-63 (2006).
25. Nikitzuk, J., Weinberg, B. Canavan, P. K., and Mavroidis C., Active Knee Rehabilitation Orthotic Device With Variable Damping Characteristics Implemented via an Electrorheological Fluid, *IEEE Trans. Mechatron.* **15**, 952-960 (2010).
26. Peñaloza-González, J. A., González-Mejía, S., García-Melo, J. I., Development of a Control Strategy in an Isokinetic Device for Physical Rehabilitation. *Sensors*, **23**, 5827-5846 (2023).

(2) Response to: “What is missing in the area.”

In the introduction, we have included the missing content in this area.

In the second paragraph of the INTRODUCTION, we listed the unaddressed challenges in this field:

“By modulating the electrical current, the driver produces an adjustable damping force that precisely counteracts knee motion. However, the employment of a high-power driver module and external electricity usually involves several external hardware (e.g., energy supply module, power transformer, and control cabin), making these facilities **bulky, heavy, energy-consuming, and expensive**. For instance, Biodex system works under 230 V power, consumes 3450 W, weighs over 600 kg, and costs over \$50,000 (19). As a result, access to isokinetic rehabilitation is largely limited to well-equipped hospitals or rehabilitation centers. **There has been a growing yet unfulfilled need for compact isokinetic training systems that can be deployed for home-based rehabilitation at low cost, especially in resource-limited areas.**”

The implementation of home-based isokinetic training devices is hindered by their

bulky, heavy, energy-consuming, and expensive nature. A key contributing factor is that all existing isokinetic devices require a high-power driver module and external hardware for electricity supply. As a result, all existing isokinetic training devices are hospital-based (14-18). To better support our study, we have conducted a thorough search on the *Web of Science*, and to the best of our efforts, **we were unable to find any studies that utilize isokinetic training devices in home-based settings.**

Fig. R1. Screenshot of the Web of Science search results for the keywords “isokinetic knee home”.

14. Goetz, J., et.al, Isokinetic knee muscle strength comparison after enhanced recovery after surgery (ERAS) versus conventional setup in total knee arthroplasty (TKA): a single blinded prospective randomized study. *J. Exp. Orthop.* **10**, 44-56 (2023).

15. Wang, S. et.al, Analysis of Isokinetic Strength Test in Arthroscopic Meniscus Suture to Improve Knee Joint Strength and Function, *Altern. Ther. Health Med.* **29**, 416-424, (2023).

16. Liu, H. et.al, Effect of isokinetic training of thigh muscle group on graft remodeling after anterior cruciate ligament reconstruction. *Chin. J. Reparative Reconstr. Surg.* **33**, 1088-1094 (2019).

17. Kim, H., KOOK, K., Lee, J., Lee, S., The Effects of a Rehabilitation Exercise Program After Posterior Cruciate Ligament (PCL) Reconstruction Between Genders. *Asian J. Kinesiol.* **24**, 39-45 (2022).

18. Rosa, U. et.al, Comparison of the Effectiveness of Isokinetic Exercise Vs Isometric Exercise Performed at Different Angles in Patients with Knee *Osteoarthritis*. *Reumatol Clin.* **8**, 10-4 (2012).

19. Biodex System 4 Pro™, Biodex Medical System; <https://www.biodex.com>.

(3) Response to: “how this new device will achieve similar/better results when used clinically than the exercises that can be done without the need of any equipment”

For patients with knee injuries, certain exercises, such as squats, can be performed without the assistance of an isokinetic training device. However, robot-assisted isokinetic training achieves better results when used clinically than exercises without the need for any equipment.

a) Isokinetic training provides safer training

During isokinetic training, the trainee sits on the isokinetic training device and actively exerts force, with the machine providing accommodating resistance throughout the process. This setup offers an **inherent safety mechanism**, as the resistance disengages when the patient experiences pain or discomfort (13).

In contrast, bodyweight squats present **potential safety risks**, particularly for patients with knee injuries. Improper posture during squatting can lead to secondary knee injuries, or excessive strain on other joints, thereby increasing the likelihood of further damage. Additionally, squatting requires a high degree of body stability and control, which patients with knee injuries often lack, making them more susceptible to improper movements and elevating the risk of additional injury.

In the first paragraph of INTRODUCTION, we updated the inherent safety nature of isokinetic training:

“Particularly, isokinetic training denotes dynamic muscular contractions wherein movement speed is a constant, ensuring that muscles exert maximal tension throughout the entire range of joint motion (8-12). This constant speed characteristic naturally integrates a safety mechanism that reduces resistance if the patient experiences pain or discomfort (13), while also enabling continuous monitoring of rehabilitation progress through objective assessments of muscle strength (14-15).”

b) Isokinetic training achieves better effectiveness

Many clinical studies demonstrate that isokinetic training yields superior effectiveness.

In the first paragraph of INTRODUCTION, we updated the description of the high effectiveness of isokinetic training:

“Recent studies have demonstrated the effectiveness of isokinetic training for knee rehabilitation (16-18), such as enhancing muscle strength, improving stability, increasing postoperative walking speed, and reducing knee pain, in various knee conditions, including anterior cruciate ligament (ACL, 16), posterior cruciate ligament (PCL, 17) and osteoarthritis (OA, 18).”

The medical rationale for the superior effectiveness of isokinetic training lies in **its ability to ensure that muscles generate maximal tension consistently throughout the full range of joint motion during the training process (8-12)**. To provide some experimental support for this explanation, **we conducted an experiment comparing isokinetic training and squatting**:

We measured the muscle activation of the quadriceps and hamstrings during both isokinetic training and squatting using EMG signals. By comparing the variation curves of muscle activation throughout both training processes and quantifying the time during which muscle activation reached 80% of maximal activation, we compared the duration of maximal tension generation between the two training methods.

The quantitative results shown in Fig. SM7 indicate that during isokinetic training, the quadriceps reached 80% of maximal activation (a 20% decrease in peak value, as marked in the figure) for 51% of the repetition while squatting only achieved 21%. Similarly, during isokinetic training, the hamstrings reached 80% muscle activation for 38% of the repetition, whereas squatting achieved only 22%. **In one Repetition, isokinetic training results in a 2.43 (51%/21%) times higher for the quadriceps and a 1.73 (38%/22%) times higher for the hamstrings compared with squatting.**

Fig. S8 | Muscle Activation Comparison between Isokinetic Training and Squatting. (a) Quadriceps muscle activation comparison. (b) Hamstring muscle activation comparison.

We provided a detailed description and discussion of the results in the DISCUSSION. Additionally, we added a section in the **Supplementary Material**, titled “Comparison between Isokinetic Training and Self-Weight Training”.

The added content in Supplementary Material is presented as follows:

Comparison between Isokinetic Training and Self-Weight Training

The medical rationale for the superior effectiveness of isokinetic training lies in its ability to ensure that muscles generate maximal tension consistently throughout the full range of joint motion during the training process (8-12). To validate that isokinetic training leads to longer durations of maximal muscle activation compared to bodyweight exercises (e.g., squatting), an experiment was conducted to compare the duration of maximal muscle activation during isokinetic training and squatting. Muscle activation of the quadriceps (**Fig. S8a**) and hamstrings (**Fig. S8b**) was measured during both isokinetic training and squatting using EMG signals.

Both isokinetic training and squatting exhibit a trend of increasing muscle activity, reaching peak activation, and then decreasing. However, the duration of high muscle activation is significantly longer during isokinetic training compared to squatting. To quantitatively compare the duration of high muscle activation, the time during which the quadriceps and hamstrings experienced muscle activation greater than 80% of their maximum activation was calculated and compared. **Figure S8a** shows that the quadriceps experience 2.43 times longer high muscle activation during isokinetic training (51%) compared to squatting (21%). Similarly, **Figure S8b** demonstrates that the hamstrings experience 1.73 times longer high muscle activation during isokinetic training (38%) compared to squatting (22%). Both the quadriceps and hamstrings demonstrate that isokinetic training resulted in longer durations of high muscle activation compared to squatting, which may contribute to the superior effectiveness of isokinetic training over bodyweight exercises such as squatting.

c) Continuous monitoring during isokinetic training

In addition to playing an important role in rehabilitation, an essential function of the isokinetic training device is **muscle strength evaluation, performing as a dynamometer**. Previous studies have demonstrated that continuous monitoring of progress contributes to improved rehabilitation outcomes (14-15).

The following content is updated in the first paragraph of INTRODUCTION:

“This constant speed characteristic naturally integrates a safety mechanism that reduces resistance if the patient experiences pain or discomfort (13), while also enabling continuous monitoring of rehabilitation progress through objective assessments of muscle strength (14-15).”

During the rehabilitation process, the ability to track real-time progress and training data can significantly enhance patient engagement. By reviewing these records, patients

gain a clearer understanding of their recovery trajectory, which can serve as a motivating factor to maintain adherence to the treatment regimen. Furthermore, the systematic collection and analysis of this data enables healthcare professionals, including physicians and rehabilitation therapists, to obtain a comprehensive understanding of the patient's recovery status. This data-driven approach allows for precise adjustments to the treatment plan, thereby optimizing therapeutic outcomes.

d) Proposed device demonstrates high clinical performance

To validate the clinical performance of the proposed isokinetic training device, we conducted a 6-week rehabilitation program with 10 post-surgical subjects. Improvements in muscle morphology and muscle strength were assessed. **We compared the efficacy of the proposed isokinetic training device with the studies employing existing isokinetic robots**, focusing on both muscle morphology (addressed via MRI) and muscle strength (measured as torque).

A detailed description of these findings is previously provided in the RESULTS and has been moved into DISCUSSION. The content is provided as follows:

Efficacy comparison between our study and commercial isokinetic robots

To assess the rehabilitation outcomes, the quadriceps CSA growth rate (**Fig. 7a**) and lifting torque increase (**Fig. 7b**) of the proposed robots and references utilizing the commercial isokinetic training robots were compared. Notably, the CSA growth rate of the proposed power-free isokinetic training robot 5.9% (**Fig. 7a**) is much higher than the documented 3.7% value by a commercial rehabilitation device (CONTREX MJ, PHYSIOMED, Germany) where the same training program was carried out (33, See **Fig. 7a**). Quadriceps muscle CSA in the untrained leg demonstrated a negligible difference in both the proposed power-free robots and commercial devices compared with the trained leg (**Fig. 7a**). This result may indicate that the impact of unilateral isokinetic training on the contralateral untrained leg is insignificant. Additionally, routine exercises (e.g., walking, jogging, climbing stairs) do not provide as pronounced stimulatory growth as isokinetic training. Similarly, the significant torque increases utilizing the power-free robots 70% is higher than the documented results of 35.9% utilizing the commercial isokinetic training device (33, See **Fig. 7b**). Other experiments conducted using commercial isokinetic devices with the same isokinetic training speed (60 degrees per second) but varying training durations and cycles also did not demonstrate torque increases surpassing the results of our study (33, 37-39, See **Fig. 7b**).

Fig. 7 | Efficacy comparison between our study and commercial isokinetic robots. (a) The CSA growth rate. (b) The torque increases.

e) Phone-based human-machine interface (HMI)

The proposed device utilizes a phone-based HMI, allowing patients to monitor their training status in real-time using only a smartphone. Additionally, all training data can be uploaded to the cloud, potentially enabling remote consultations by experienced physicians. This HMI offers greater convenience, lower costs, and a smaller footprint.

A detailed description of these findings is provided in the DISCUSSION, with the following content:

Human-machine interface

For the convenience of rehabilitation, we developed a home-based isokinetic training robot to offer a real-time human-machine interface. Its operational framework includes several stages. Initially, a custom application was developed to collect training data in real-time via a Bluetooth module. Subsequently, this amassed training data is instantaneously transmitted to a secure cloud service, facilitating seamless and immediate analysis. Following this, a proficient medical practitioner evaluates the patient's adherence to the training regimen, assessing their training performance and providing insightful recommendations. This feedback loop completes as the doctor's assessments and recommendations are relayed back to the patient, fostering a continuous cycle of communication. Noteworthy is the dynamic nature of this engagement, as the received feedback prompts the fine-tuning of the training schedule, adapting it precisely based on the doctor's assessments. This holistic approach underscores the patient-centric design of the robot, harmonizing technological innovation with medical expertise to optimize patient well-being.

Reference:

8. Moffroid, M. et al. A study of isokinetic exercise. *Phys. Ther.* **49**, 735-747 (1969).
9. Verrill, D. et al. Resistive exercise training in cardiac patients. *Sports Med.* **13**, 171-193 (1992).
10. Hislop, H. J. & Perrine, J. The isokinetic concept of exercise. *Phys. Ther.* **47**, 114-117 (1967).
11. Thistle H. G. et al. Isokinetic contraction: a new concept of resistive exercise. *Arch. Phys. Med. Rehabil.* **48**, 279-282 (1967).
12. Cabri J., Isokinetic strength aspects in human joints and muscles. *Appl. Ergon.* **22**, 299-302 (1991).
13. Jee, Y., Usefulness of measuring isokinetic torque and balance ability for exercise rehabilitation. *J. Exerc. Rehabil.* **11**, 65-66 (2015).
14. Goetz, J., et.al, Isokinetic knee muscle strength comparison after enhanced recovery after surgery (ERAS) versus conventional setup in total knee arthroplasty (TKA): a single blinded prospective randomized study. *J. Exp. Orthop.* **10**, 44-56 (2023).
15. Wang, S. et.al, Analysis of Isokinetic Strength Test in Arthroscopic Meniscus Suture to Improve Knee Joint Strength and Function, *Altern. Ther. Health Med.* **29**, 416-424, (2023).
16. Liu, H. et.al, Effect of isokinetic training of thigh muscle group on graft remodeling after anterior cruciate ligament reconstruction. *Chin. J. Reparative Reconstr. Surg.* **33**, 1088-1094 (2019).
17. Kim, H., KOOK, K., Lee, J., Lee, S., The Effects of a Rehabilitation Exercise Program After Posterior Cruciate Ligament (PCL) Reconstruction Between Genders. *Asian J. Kinesiol.* **24**, 39-45 (2022).
18. Rosa, U. et.al, Comparison of the Effectiveness of Isokinetic Exercise Vs Isometric Exercise Performed at Different Angles in Patients with Knee *Osteoarthritis*. *Rheumatol Clin.* **8**, 10-4 (2012).
33. Zhang, X. et al. Impact of whey protein isolate and eccentric training on quadriceps mass and strength in patients with anterior cruciate ligament rupture: A randomized controlled trial. *J. Rehabil. Med.* **52**, 1-8 (2020).
37. Coyle, E. F. et al. Specificity of power improvements through slow and fast isokinetic training. *J. Appl. Physiol. Respir. Environ. Exerc. Physiol.* **51**, 1437-1442 (1981).
- 38 Wang, K. et al. Effect of isokinetic muscle strength training on knee muscle strength, proprioception, and balance ability in athletes with anterior cruciate ligament reconstruction: a randomised control trial. *Front. Physiol.* **14**, (2023).

39. Ewing Jr, J. L. et al. Effects of velocity of isokinetic training on strength, power, and quadriceps muscle fibre characteristics. *Eur. J. Appl. Physiol. Occup. Physiol.* **61**, 159-162 (1990).

(4) **Response to:** “Ethical aspects regarding the clinical trial were correctly presented in the manuscript.”

We sincerely appreciate your recognition.

(5) **Response to:** “The methods section is well described, but some aspects are not completely clear (see Specific Comments). The results need to be reorganized, as several sections are more related to the Methods Section, not to the Results”

We sincerely appreciate your professional suggestion. **The METHOD and RESULTS have been restructured according.**

(a) Dividing the “Demonstrating of isokinetic training” section

In accordance with your suggestion, we have divided the previous section titled “*Demonstration of isokinetic training*” into two distinct parts: one retained as “*Demonstration of isokinetic training*” **now located in RESULTS**, and the other titled “*Procedure of isokinetic training*” which has been **moved to METHODS**.

In lines 158-178, we introduce the isokinetic training with a demonstration figure, **Fig. 3**, consisting of three subfigures: **Fig. 3a**, **Fig. 3b**, and **Fig. 3c**. **Fig. 3a** illustrates the preparation steps before each training trial, while **Fig. 3b** shows experimental pictures depicting the procedure of one repetition. Both **Fig. 3a** and **Fig. 3b** are intended to demonstrate the experimental process, providing context for the experimental data presented in **Fig. 3c**.

In the updated “*Demonstration of isokinetic training*” in RESULTS, the content originally spanning lines 158-172 has been condensed, with a focus solely on presenting the experimental basis for **Fig. 3c**.

The updated content for “Demonstration of isokinetic training” in the RESULTS is as follows:

Demonstration of isokinetic training

To evaluate the performance of our isokinetic training robot, we conducted a study with 10 post-surgical subjects and 10 healthy subjects (See **Supplementary Table S2, Table S3**). Preparation for the isokinetic training involves four steps as outlined in **Fig. 3a** (**Detailed description see METHODS**). The training protocol consists of eight repetitions of lifting and retracting movements in one cycle. Experimental pictures are shown in **Fig. 3b**. **Figure 3c** presents representative isokinetic training data from a post-surgical subject. A constant training velocity of $\omega = 60^\circ/\text{s}$ is maintained during both lifting and retracting processes (33). During the training, subjects are required to exert

maximum muscle strength to lift and retract the calf. Notably, the training device is designed not to assist leg motion but to provide accommodative resistance if the velocity exceeds 60°/s, ensuring the training intensity. Within each repetition, the training velocity exhibits a pattern of quick acceleration, stability at target velocity, and rapid deceleration, in which the constant velocity time occupies about 70% of training (**Fig. 3c, Velocity**). The torque displays a similar pattern with a gradual decline over the training course due to muscle fatigue (**Fig. 3c, Torque**). As the training continues, the regenerated energy accumulates in both the lifting and retracting processes (**Fig. 3c, Regenerated Energy**). A demonstration video is shown in **Supplementary Movie S2**.

The detailed training procedure has been moved to the METHODS and is now titled “**Procedure of isokinetic training**” with the following content:

Procedure of isokinetic training

All demonstration experiments involved four steps as demonstrated in **Fig. 3a**. The first step is positioning the leg to ensure that the main arm of the robot is parallel to the calf, which allows for the precise measurement of the knee joint angle θ (40). The second step is fastening the belt to stabilize the body. This stabilization minimizes the movement of the hip joint, isolating the knee for target training only. The third step is connecting to the smartphone via Bluetooth, establishing the human-machine interface. The fourth step is calibration. The subject lifts and retracts his calf to the maximum and minimum angles, establishing a range of motion benchmark for the training protocol. In all trials, a training speed of 60°/s was used. **Previous studies show that 60°/s is effective in various rehabilitation aspects, contributing to improvements in muscle strength improvement and functional recovery (41-42).**

Fig.3 | Demonstration of isokinetic training. (a) Preparation steps. Step 1: Position the leg and align the knee joint with the rotation axis. Step 2: Fasten the belt to stabilize the body. Step 3: Connect Bluetooth with the smartphone. Step 4: Calibration by testing the maximum and minimum angle of the subject. (b) Experimental pictures showing that a post-surgical subject lifts and retracts the calf during the training. (c) The training data of knee angle, velocity, torque, and regenerated energy over 8 repetitions of lifting-retracting movement.

(b) Reorganizing the placement of the “Energy regeneration during isokinetic training” section

We have moved all content in the section entitled “Energy regeneration during isokinetic training” from RESULTS to METHODS. Accordingly, we have relocated the previous Figure 3 in the main text to the Supplemental Materials as Figure S4. This figure provides a biomedical model and an energy regeneration principle for isokinetic training.

The restructured figure and content are as follows:

Energy regeneration during isokinetic training

The proposed dynamic energy regeneration method enables power-free functionality

and isokinetic training, simultaneously. The knee joint is regarded as a bio-mechanical system comprising two skeleton bones, the femur, and the tibia, as well as two antagonistic muscles, quadriceps femoris (dominating knee lifting) and hamstring muscle (dominating knee retracting), as shown in **Supplementary Fig. S4a**. In this system, after the gravitational compensation, the torque T from human dynamics is approximately modeled as,

$$T = F_s \cdot l_s \approx T_m \cdot \beta \cdot \widetilde{W}_l(\theta) \cdot \widetilde{W}_v(\dot{\theta}) \quad (1)$$

where F_s is a force exerted by the trainee, l_s is the corresponding moment arm. In detail, F_Q and l_Q represent quadriceps femoris contraction, F_H and l_H represent hamstring muscle contraction, T_m is the maximum torque that muscle can exert, β ($0 \leq \beta \leq 1$) is a normalized muscle activation level, $\widetilde{W}_l(\theta)$ is a torque-angle factor and $\widetilde{W}_v(\dot{\theta})$ is a torque-velocity factor. The detailed principle is shown in **Supplementary Fig. S5** and Supplementary Information Text.

If the motor windings were fully shorted, indicating without any control and all low-side MOSFETs are on (**Fig. S4b**), the maximum current I_{max} , due to the torque T ,

$$I_{max} = \frac{T_m \cdot \beta \cdot \widetilde{W}_l(\theta) \cdot \widetilde{W}_v(\dot{\theta})}{k} \quad (2)$$

where k is the motor torque constant.

If the duration of the short motor windings was controlled dynamically (see **Supplementary Fig. S4b**), as duty cycle D , the accommodative resistance torque τ_{res} is

$$\tau_{res} = kI_a = kf(D) \leq kI_{max} \quad (3)$$

where I_a is the phase current, which is associated with D , the maximum I_a is equal to I_{max} .

To realize isokinetic training, a proportional-integral (PI) controller is to adjust D based on errors,

$$D = K_p (\dot{\theta} - \bar{\theta}) + K_I (\dot{\theta} - \bar{\theta}) \quad (4)$$

where K_p is a proportional coefficient, K_I is an integral coefficient and $\bar{\theta}$ is the desired velocity.

Accompanying isokinetic training, dynamic energy regeneration can be obtained, and the regenerative power is

$$P_{reg} = \frac{(\int_0^{t_2} I_{charge} dt)^2 - (\int_0^{t_1} I_{charge} dt)^2}{2C(t_2 - t_1)} \quad (5)$$

where I_{charge} is the charging current. When low-side MOSFETs are off, I_{charge} is equal to I_a , when low-side MOSFETs are on, I_{charge} is 0 (**Fig. S4b**), C is the capacitance and t_1 and t_2 are different timings. The detailed charging circuit principle is shown in **Supplementary Fig. S6** and **Supplementary Fig. S7**.

Fig. S4 | Energy regeneration during isokinetic training. (a) Lifting and retracting torque model. Quadriceps femoris contraction force F_Q dominates the lifting force, with the corresponding moment arm l_Q . Hamstring muscle contraction force F_H dominates the retracting force, with the corresponding moment arm l_H . β is a normalized muscle activation level. G is the gravitational force, and l_G is the corresponding moment arm. **(b)** Energy regeneration circuit. The low-side MOSFETs are switched ON/OFF. When the MOSFETs are OFF, the charging current I_{charge} equals the phase current I_a , and the current flows into the ultracapacitor. When the MOSFETs are ON, although the induced phase current exists, the charging current $I_{charge} = 0$.

(c) Rationale for Selecting 60°/s as the Isokinetic Speed

Clinical studies indicate that a training speed of 60°/s is a commonly used rehabilitation velocity, which is considered effective in promoting muscle strength gains. This speed has been shown to be most effective in enhancing muscle strength during isokinetic training (41-42).

Moreover, to evaluate the effectiveness of the proposed device at a clinical level, we conducted the same clinical trial protocol used in previous studies (isokinetic training speed of 60°/s, 8-10 repetitions per set, training duration of 6 weeks) and compared the outcomes of subjects using our proposed device with those using the commercial device referenced in (33). The use of 60°/s is based on the study referenced in (33), as it allows for controlled variables and provides a meaningful comparison of the outcomes across both devices.

We updated the METHOD by adding the following content:

Procedure of isokinetic training

All demonstration experiments involved four steps as demonstrated in **Fig. 3a**. The first step is positioning the leg to ensure that the main arm of the robot is parallel to the calf, which allows for the precise measurement of the knee joint angle θ (40). The second step is fastening the belt to stabilize the body. This stabilization minimizes the movement of the hip joint, isolating the knee for target training only. The third step is connecting to the smartphone via Bluetooth, establishing the human-machine interface. The fourth step is calibration. The subject lifts and retracts his calf to the maximum and minimum angles, establishing a range of motion benchmark for the training protocol. In all trials, a training speed of 60°/s was used. Previous studies show that 60°/s is effective in various rehabilitation aspects, contributing to improvements in muscle strength improvement and functional recovery (41-42).

Reference:

33. Zhang, X. et al. Impact of whey protein isolate and eccentric training on quadriceps mass and strength in patients with anterior cruciate ligament rupture: A randomized controlled trial. *J. Rehabil. Med.* **52**, 1-8 (2020).
40. Kocsis, L. et al. Biomechanical Models and Measuring Techniques for Ultrasound-Based Measuring System during Gait, *Period. Polytech. Mech. Eng.* **48**, 1-14 (2004).
41. Jan M. H., Gain of muscle torque at low and high speed after isokinetic knee strengthening program in healthy young and older adults. *J. Formos. Med. Assoc.* **97**, 339-344 (1998)
42. Cengizel, Ç., & Pekel, H., Can isokinetic strength training be an alternative to machine-based strength training? *Medicina dello. Sport.* **76**, 495-506 (2023).

- (6) **Response to:** “In the discussion section, the authors clearly described the device features, described several results that were obtained with the robot, but did not adequately discuss their results while contrasting them with the existent literature”

We sincerely appreciate your valuable comments. We conducted two related comparisons of our proposed home-based isokinetic training device with the current hospital-based isokinetic training device.

(a) Comparison of Driver modules

To demonstrate the contrast in the realization of isokinetic control between our proposed home-based isokinetic training device and current commercial hospital-based devices, we have updated the DISCUSSION section by adding a comparison of the driver modules used in both systems.

We added the comparison of **driver modules** in the first and second paragraphs of the DISCUSSION:

“Conventional knee isokinetic training facilities are often characterized limited by their substantial size, weight, and high energy consumption. *At the core of this limitation is the energy-intensive driver module, which is required to generate the controllable resistive torque. Existing driver modules rely on high-power servo motors (19-23), magnetorheological (MR) fluids (24), electrorheological (ER) fluids (25), and electromagnetic powder brakes (26). All of these require continuous and precise electrical control—whether it involves supplying large currents, maintaining high voltages, or generating strong magnetic fields.* These characteristics limit the rehabilitation training exclusively to clinical settings such as hospitals and rehabilitation centers.

The proposed power-free isokinetic robot can provide home-based knee rehabilitation without an external power supply. The power-free working mechanism is realized by the implementation of a controllable motor-winding short strategy. This strategy allows the device to harvest excessive mechanical energy generated by the user during exercise sessions, converting it into electrical energy that runs the system. Meanwhile, the controllable short motor-winding produces a controllable accommodative resistance torque. Using the velocity as closed-loop feedback, isokinetic training is realized.”

(b) Comparison of clinical efficiency

Following your suggestion, **we redefined the structure and have moved the “Efficacy comparison between our study and commercial isokinetic robots” to DISCUSSION.** This part primarily discusses our clinical results in comparison with the clinical outcomes of existing hospital-based isokinetic training devices.

The comparison of clinical efficiency is attached below:

Efficacy comparison between our study and commercial isokinetic robots

To assess the rehabilitation outcomes, the quadriceps CSA growth rate (**Fig. 7a**) and lifting torque increase (**Fig. 7b**) of the proposed robots and references utilizing the commercial isokinetic training robots were compared. Notably, the CSA growth rate of the proposed power-free isokinetic training robot 5.9% (**Fig. 7a**) is much higher than the documented 3.7% value by a commercial rehabilitation device (CONTREX MJ, PHYSIOMED, Germany) where the same training program was carried out (33, See **Fig. 7a**). Quadriceps muscle CSA in the untrained leg demonstrated a negligible difference in both the proposed power-free robots and commercial devices compared with the trained leg (**Fig. 7a**). This result may indicate that the impact of unilateral isokinetic training on the contralateral untrained leg is insignificant. Additionally, routine exercises (e.g., walking, jogging, climbing stairs) do not provide as pronounced stimulatory growth as isokinetic training. Similarly, the significant torque increases utilizing the power-free robots 70% is higher than the documented results of 35.9% utilizing the commercial isokinetic training device (33, See **Fig. 7b**). Other experiments conducted using commercial isokinetic devices with the same isokinetic training speed (60 degrees per second) but varying training durations and cycles also did not demonstrate torque increases surpassing the results of our study (33, 37-39, See **Fig. 7b**).

Fig. 7 | Efficacy comparison between our study and commercial isokinetic robots.

(a) The CSA growth rate. **(b)** The torque increases.

Reference:

19. Biodex System 4 Pro™, Biodex Medical System; <https://www.biodex.com/physical-medicine/products/dynamometers/system-4-pro>.
20. Kineo LegPro, Kineo System Intelligent Load; <https://www.kineosystem.com/range/kineo-leg-pro/>.
21. CON-TREX MJ, PHYSIOMED, <https://physiomed.de/en/produkt/con-trex-mj/>.
22. IsoMed2000, D&R, http://www.isomed2000.de/comp.gen.intro.php?lc=en_cn.
23. Hu B. et.al, Disturbance Rejection Speed Control Based on Linear Extended State Observer for Isokinetic Muscle Strength Training System, *IEEE Trans. Autom. Sci. Eng.* **20**, 1962-1971 (2023).
24. Dong S., Lu K., Sun J., Rudolph K., Adaptive force regulation of muscle strengthening rehabilitation device with magnetorheological fluids. *IEEE Trans. Neural Syst. Rehabil. Eng.* **14**, 55-63 (2006).
25. Nikitzuk, J., Weinberg, B. Canavan, P. K., and Mavroidis C., Active Knee Rehabilitation Orthotic Device With Variable Damping Characteristics Implemented via an Electrorheological Fluid, *IEEE Trans. Mechatron.* **15**, 952-960 (2010).
26. Peñaloza-González, J. A., González-Mejía, S., García-Melo, J. I., Development of a Control Strategy in an Isokinetic Device for Physical Rehabilitation. *Sensors*, **23**, 5827-5846 (2023).
33. Zhang, X. et al. Impact of whey protein isolate and eccentric training on quadriceps mass and strength in patients with anterior cruciate ligament rupture: A randomized controlled trial. *J. Rehabil. Med.* **52**, 1-8 (2020).
34. An, K. N., Linscheid, R. L., Brand, P. W., Correlation of physiological cross-sectional areas of muscle and tendon. *J. Hand. Surg. Br. Eur. Vol.* **16**, 66-67 (1991).
35. Franchi, M V., et al. Muscle thickness correlates to muscle cross-sectional area in the assessment of strength training-induced hypertrophy, *Scand. J. Med. Sci. Sports* **28**, 846-853 (2018).
36. Chen L., et al. How Do Muscle Function and Quality Affect the Progression of KOA? A Narrative Review. *Orthop Surg.* **16**, 802-810 (2024).
37. Coyle, E. F. et al. Specificity of power improvements through slow and fast isokinetic training. *J. Appl. Physiol. Respir. Environ. Exerc. Physiol.* **51**, 1437-1442 (1981).
38. Wang, K. et al. Effect of isokinetic muscle strength training on knee muscle strength, proprioception, and balance ability in athletes with anterior cruciate ligament reconstruction: a randomised control trial. *Front. Physiol.* **14**, (2023).
39. Ewing Jr, J. L. et al. Effects of velocity of isokinetic training on strength, power,

and quadriceps muscle fibre characteristics. *Eur. J. Appl. Physiol. Occup. Physiol.* **61**, 159-162 (1990).

Specific Comments:

Comment 1:

The words “highly integrated” in the title do not allow the reader to understand with what exactly the rehabilitation robot is integrated with.

Response 1:

Thank you for your comment. We have updated the title by removing the term “**Highly Integrated**” to avoid any misunderstanding.

Previous Title: “Highly Integrated and Power-Free Knee Rehabilitation Robot for Home-Based Isokinetic Training”

Updated Title: “Power-Free Knee Rehabilitation Robot for Home-Based Isokinetic Training”

Please allow me to explain the meaning of why we were using “highly integrated”, including 3 aspects:

(1) Highly Integrated Controller

The current control methods require an external energy source to supply the necessary current. However, our proposed isokinetic training device operates without external power, meaning it is power-free. By employing the motor winding short (MWS) control method, we have replaced the large control modules typically found in hospital-based devices with a self-designed, highly integrated control board (**Fig. 1a**). This modification significantly reduces both the size and cost of the device.

The updated fourth paragraph of the INTRODUCTION now includes the following description of the proposed **home-based** isokinetic training device features:

“Notably, this strategy can provide electromagnetic damping torque of tens of N·m with energy consumption lower than several watts (28-29). For instance, a transtibial prosthesis utilizing the MWS method achieved precise control with a small energy consumption of 1.5 W (28).”

In contrast, we give an example of a **hospital-based** isokinetic training device, and updated the content in the second paragraph of INTRODUCTION:

“For instance, Biodex system works under 230 V power, consumes 3450 W, weighs over 600 kg, and costs over \$50,000 (19).”

(2) Highly Integrated Energy Module

The isokinetic device proposed in this study utilizes the motor winding short (MWS) method to provide controlled resistance while simultaneously harvesting training energy to achieve **power-free isokinetic training**. In contrast to other **high-power-consuming** driving methods (19-26), which rely on energy modules and power supply

units, the proposed power-free isokinetic training device does not require any external power source. The energy harvesting and control components are highly integrated into a thin, compact control board (see **Figure 1**).

(3) Highly integrated Human machine interface (HMI)

Most existing human-machine interfaces (HMI) are desktop-based, requiring large control cabinets that occupy significant space, incur high costs, and lack portability. In contrast, the proposed device utilizes a phone-based HMI, enabling patients to monitor their training status via a smartphone. Moreover, all training data can be uploaded to the cloud, facilitating potential remote consultations with experienced physicians. This HMI offers enhanced convenience, reduced costs, and a more compact design.

Through the “highly integrated” of these three aspects, our proposed isokinetic training device successfully enables home-based training.

Fig. 1 | Proposed power-free knee rehabilitation robot, realization and comparison. (a) Existing hospital-based rehabilitation devices. These devices are bulky and need additional power supplies. (b) Proposed home-based isokinetic training robot. The

proposed isokinetic training robot is power-free and highly integrated. **(c)** Explosion Assembly. During the training, the rotation center of the knee joint and the motor are concentric. **(d)** Isokinetic realization and energy regeneration. The isokinetic training robot can generate energy while training. A desired constant velocity is predetermined. The velocity is monitored via angle sensors and the motor output resistance is modulated to ensure the actual velocity closely approximates the predetermined desired velocity. The controlled resistance is adjusted and the negative mechanical work during training is harvested. The harvested energy is subsequently channeled to power the control circuit, sensors, and other usage. **(e)** Comparison. A comparison between commercial isokinetic training robots and our proposed isokinetic training robot. The consumed power and weight of our robot are both dramatically reduced.

Comment 2:

It would be interesting to see some results in the abstract so that the reader does not have to just believe in what you are saying that happened, but actually can see what happened.

Response 2:

Thanks for your comment, we updated the ABSTRACT to include the clinical validation results, and encompass improvements in muscle growth (cross-sectional area) and muscle strength (torque) for both the quadriceps and hamstrings. The updated content is as follows:

Previous Abstract:

Robot-assisted isokinetic training has been widely adopted for knee rehabilitation. However, existing rehabilitation facilities are often heavy, bulky, expensive, and extremely energy-consuming, which limits the rehabilitation opportunities only at designated hospitals. In this work, we introduce a highly integrated and lightweight (52 kg) knee rehabilitation robot that can provide home-based isokinetic training without external power. By integrating a motor, torque/angle sensors, control circuit, and energy regeneration circuit into a single driver module, our robot can provide power-free isokinetic training by recycling mechanical work from the trainee. The system's user-friendly interface is managed via a smartphone, facilitating ease of use and allowing for the transmission of real-time training data for remote training assistance. Ten post-surgical subjects were involved in an interventional randomized trial (ChiCTR2300076715) and the cross-sectional area of trained legs (experimental group) was significantly higher than that of untrained legs (control group). Our analysis reveals that muscle growth and strength improvements achieved with our robots outperform existing commercial rehabilitation devices. These findings indicate that our robot presents a viable option for home-based knee rehabilitation, significantly enhancing the accessibility of effective treatment.

Updated Abstract:

“Robot-assisted isokinetic training has been widely adopted for knee rehabilitation. However, existing rehabilitation facilities are often heavy, bulky, expensive, and extremely energy-consuming, which limits the rehabilitation opportunities only at designated hospitals. In this work, we introduce a highly integrated and lightweight (52 kg) knee rehabilitation robot that can provide home-based isokinetic training without external power. By integrating a motor, torque/angle sensors, control circuit, and energy regeneration circuit into a single driver module, our robot can provide power-free isokinetic training by recycling mechanical work from the trainee. The system’s user-friendly interface is managed via a smartphone, facilitating ease of use and allowing for the transmission of real-time training data for remote training assistance. Ten post-surgical subjects were involved in an interventional randomized trial (ChiCTR2300076715) and the cross-sectional area of trained legs was significantly higher than that of untrained legs. Our analysis reveals that muscle growth (quadriceps: 5.93%, hamstrings: 10.27%) and strength improvements (quadriceps: 70%, hamstrings: 84%) achieved with our robots outperform existing commercial rehabilitation devices. These findings indicate that our robot presents a viable option for home-based knee rehabilitation, significantly enhancing the accessibility of effective treatment.”

Comment 3:

Line 51. I suggest that you use more recent studies to support what you are stating here, as reference 8 is among the earliest studies in the isokinetic area.

Related Content:

This training methodology ensures that muscles consistently generate maximal tension throughout the entire range of joint motion (8-12), giving rise to an optimal rehabilitation outcome that has been widely substantiated in recent research (1, 8).

Response 3:

Thank you for your comment. To demonstrate the effectiveness of isokinetic training in rehabilitation, we have **updated more recent studies (16-18)** to support our opinion that isokinetic training offers significant benefits for the rehabilitation of various knee injuries. These benefits include enhancing muscle strength, improving stability, increasing postoperative walking speed, and reducing knee pain.

We have updated the first paragraph of the INTRODUCTION with the following content:

“Recent studies have demonstrated the effectiveness of isokinetic training for knee rehabilitation (16-18), such as enhancing muscle strength, improving stability, increasing postoperative walking speed, and reducing knee pain, in various knee

conditions, including anterior cruciate ligament (ACL, 16), posterior cruciate ligament (PCL, 17) and osteoarthritis (OA, 18).”

Reference:

16. Liu, H. et.al, Effect of isokinetic training of thigh muscle group on graft remodeling after anterior cruciate ligament reconstruction. *Chin. J. Reparative Reconstr. Surg.* **33**, 1088-1094 (2019).

17. Kim, H., KOOK, K., Lee, J., Lee, S., The Effects of a Rehabilitation Exercise Program After Posterior Cruciate Ligament (PCL) Reconstruction Between Genders. *Asian J. Kinesiol.* **24**, 39-45 (2022).

18. Rosa, U. et.al, Comparison of the Effectiveness of Isokinetic Exercise Vs Isometric Exercise Performed at Different Angles in Patients with Knee *Osteoarthritis*. *Reumatol Clin.* **8**, 10-4 (2012).

Comment 4:

Line 59. Please avoid the use of “etc” and list the most important external hardware

Response 4:

Thanks for your comment, we updated the content in Line 59 by deleting “etc” and listing the most important external hardware.

Previous content:

However, the employment of a high-power servo motor and external electricity usually involves several external hardware (e.g., control cabin, power transformer, etc), making these facilities bulky, heavy, and energy-consuming.

Updated content:

However, the employment of a high-power servo motor and external electricity usually involves several external hardware (e.g., energy supply module, power transformer, **and control cabin**), making these facilities bulky, heavy, and energy-consuming.

Comment 5:

Why would an isokinetic device be better than closed kinetic chain exercises involving body weight (e.g., squatting, single-leg squat) or than exercises with free weights for the home-based knee rehabilitation program? There is no or little cost and we may get similar structural and functional gains post-training. In other words, you need to convince the reader of why your device is better.

Response 5:

For patients with knee injuries, rehabilitation without a home-based isokinetic training device is limited to exercises that can be done without the need for equipment,

e.g., squats. The following compares isokinetic training with squatting from the following perspectives:

(1) Isokinetic training provides safer training

During isokinetic training, the trainee sits on the isokinetic training device and actively exerts force, with the machine providing accommodating resistance throughout the process. This setup offers an **inherent safety mechanism**, as the resistance disengages when the patient experiences pain or discomfort (13).

In contrast, bodyweight squats present **potential safety risks**, particularly for patients with knee injuries. Improper posture during squatting can lead to secondary knee injuries, or excessive strain on other joints, thereby increasing the likelihood of further damage. Additionally, squatting requires a high degree of body stability and control, which patients with knee injuries often lack, making them more susceptible to improper movements and elevating the risk of additional injury.

In the first paragraph of INTRODUCTION, we updated the inherent safety nature of isokinetic training:

“Particularly, isokinetic training denotes dynamic muscular contractions wherein movement speed is a constant, ensuring that muscles exert maximal tension throughout the entire range of joint motion (8-12). This constant speed characteristic naturally integrates a safety mechanism that reduces resistance if the patient experiences pain or discomfort (13), while also enabling continuous monitoring of rehabilitation progress through objective assessments of muscle strength (14-15).”

(2) Isokinetic training achieves better effectiveness

Many clinical studies demonstrate that isokinetic training yields superior effectiveness.

In the first paragraph of INTRODUCTION, we updated the description of the high effectiveness of isokinetic training:

“Recent studies have demonstrated the effectiveness of isokinetic training for knee rehabilitation (16-18), such as enhancing muscle strength, improving stability, increasing postoperative walking speed, and reducing knee pain, in various knee conditions, including anterior cruciate ligament (ACL, 16), posterior cruciate ligament (PCL, 17) and osteoarthritis (OA, 18).”

The medical rationale for the superior effectiveness of isokinetic training lies in **its ability to ensure that muscles generate maximal tension consistently throughout the full range of joint motion during the training process (8-12)**. To provide some experimental support for this explanation, **we conducted an experiment comparing isokinetic training and squatting:**

We measured the muscle activation of the quadriceps and hamstrings during both

isokinetic training and squatting using EMG signals. By comparing the variation curves of muscle activation throughout both training processes and quantifying the time during which muscle activation reached 80% of maximal activation, we compared the duration of maximal tension generation between the two training methods.

The quantitative results shown in **Fig. S8** indicate that during isokinetic training, the quadriceps reached 80% of maximal activation (a 20% decrease in peak value, as marked in the figure) for 51% of the repetition while squatting only achieved 21%. Similarly, during isokinetic training, the hamstrings reached 80% muscle activation for 38% of the repetition, whereas squatting achieved only 22%. **In one Repetition, isokinetic training results in a 2.43 (51%/21%) times higher for the quadriceps and a 1.73 (38%/22%) times higher for the hamstrings compared with squatting.**

Fig. S8 | Muscle Activation Comparison between Isokinetic Training and Squatting. (a) Quadriceps muscle activation comparison. (b) Hamstring muscle activation comparison.

We added a section in the **Supplementary Material**, titled “**Comparison between Isokinetic Training and Self-Weight Training**” with the following content:

The added content in Supplementary Material is presented as follows:

Comparison between Isokinetic Training and Self-Weight Training

The medical rationale for the superior effectiveness of isokinetic training lies in its ability to ensure that muscles generate maximal tension consistently throughout the full

range of joint motion during the training process (8-12). To validate that isokinetic training leads to longer durations of maximal muscle activation compared to bodyweight exercises (e.g., squatting), an experiment was conducted to compare the duration of maximal muscle activation during isokinetic training and squatting. Muscle activation of the quadriceps (**Fig. S8a**) and hamstrings (**Fig. S8b**) was measured during both isokinetic training and squatting using EMG signals.

Both isokinetic training and squatting exhibit a trend of increasing muscle activity, reaching peak activation, and then decreasing. However, the duration of high muscle activation is significantly longer during isokinetic training compared to squatting. To quantitatively compare the duration of high muscle activation, the time during which the quadriceps and hamstrings experienced muscle activation greater than 80% of their maximum activation was calculated and compared. **Figure S8a** shows that the quadriceps experience 2.43 times longer high muscle activation during isokinetic training (51%) compared to squatting (21%). Similarly, **Figure S8b** demonstrates that the hamstrings experience 1.73 times longer high muscle activation during isokinetic training (38%) compared to squatting (22%). Both the quadriceps and hamstrings demonstrate that isokinetic training resulted in longer durations of high muscle activation compared to squatting, which may contribute to the superior effectiveness of isokinetic training over bodyweight exercises such as squatting.

(3) Continuous monitoring during isokinetic training

In addition to playing an important role in rehabilitation, an essential function of the isokinetic training device is **muscle strength evaluation, performing as a dynamometer**. Previous studies have demonstrated that continuous monitoring of progress contributes to improved rehabilitation outcomes (14-15).

The following content is updated in the first paragraph of INTRODUCTION:

“This constant speed characteristic naturally integrates a safety mechanism that reduces resistance if the patient experiences pain or discomfort (13), while also enabling continuous monitoring of rehabilitation progress through objective assessments of muscle strength (14-15).”

During the rehabilitation process, the ability to track real-time progress and training data can significantly enhance patient engagement. By reviewing these records, patients gain a clearer understanding of their recovery trajectory, which can serve as a motivating factor to maintain adherence to the treatment regimen. Furthermore, the systematic collection and analysis of this data enables healthcare professionals, including physicians and rehabilitation therapists, to obtain a comprehensive understanding of the patient's recovery status. This data-driven approach allows for precise adjustments to the treatment plan, thereby optimizing therapeutic outcomes.

(4) Phone-based human-machine interface (HMI)

The proposed device utilizes a phone-based HMI, allowing patients to monitor their training status in real time using only a smartphone. Additionally, all training data can be uploaded to the cloud, potentially enabling remote consultations by experienced physicians. This HMI offers greater convenience, lower costs, and a smaller footprint.

A detailed description of these findings is provided in the DISCUSSION, with the following content:

Human-machine interface

For the convenience of rehabilitation, we developed a home-based isokinetic training robot to offer a real-time human-machine interface. Its operational framework includes several stages. Initially, a custom application was developed to collect training data in real-time via a Bluetooth module. Subsequently, this amassed training data is instantaneously transmitted to a secure cloud service, facilitating seamless and immediate analysis. Following this, a proficient medical practitioner evaluates the patient's adherence to the training regimen, assessing their training performance and providing insightful recommendations. This feedback loop completes as the doctor's assessments and recommendations are relayed back to the patient, fostering a continuous cycle of communication. Noteworthy is the dynamic nature of this engagement, as the received feedback prompts the fine-tuning of the training schedule, adapting it precisely based on the doctor's assessments. This holistic approach underscores the patient-centric design of the robot, harmonizing technological innovation with medical expertise to optimize patient well-being.

Reference:

8. Moffroid, M. et al. A study of isokinetic exercise. *Phys. Ther.* **49**, 735-747 (1969).
9. Verrill, D. et al. Resistive exercise training in cardiac patients. *Sports Med.* **13**, 171-193 (1992).
10. Hislop, H. J. & Perrine, J. The isokinetic concept of exercise. *Phys. Ther.* **47**, 114-117 (1967).
11. Thistle H. G. et al. Isokinetic contraction: a new concept of resistive exercise. *Arch. Phys. Med. Rehabil.* **48**, 279-282 (1967).
12. Cabri J., Isokinetic strength aspects in human joints and muscles. *Appl. Ergon.* **22**, 299-302 (1991).
13. Jee, Y., Usefulness of measuring isokinetic torque and balance ability for exercise rehabilitation. *J. Exerc. Rehabil.* **11**, 65-66 (2015).
14. Goetz, J., et.al, Isokinetic knee muscle strength comparison after enhanced recovery after surgery (ERAS) versus conventional setup in total knee arthroplasty (TKA): a single blinded prospective randomized study. *J. Exp. Orthop.* **10**, 44-56 (2023).
15. Wang, S. et.al, Analysis of Isokinetic Strength Test in Arthroscopic Meniscus Suture to Improve Knee Joint Strength and Function, *Altern. Ther. Health Med.* **29**, 416-

424, (2023).

16. Liu, H. et.al, Effect of isokinetic training of thigh muscle group on graft remodeling after anterior cruciate ligament reconstruction. *Chin. J. Reparative Reconstr. Surg.* **33**, 1088-1094 (2019).

17. Kim, H., KOOK, K., Lee, J., Lee, S., The Effects of a Rehabilitation Exercise Program After Posterior Cruciate Ligament (PCL) Reconstruction Between Genders. *Asian J. Kinesiol.* **24**, 39-45 (2022).

18. Rosa, U. et.al, Comparison of the Effectiveness of Isokinetic Exercise Vs Isometric Exercise Performed at Different Angles in Patients with Knee *Osteoarthritis*. *Reumatol Clin.* **8**, 10-4 (2012).

Comment 6:

In addition, it would be important for the reader to get an idea at the introduction about previous literature on the use of isokinetic devices for knee rehabilitation and the limitation of these devices for home-based rehabilitation. How many studies have been conducted to rehabilitate knee patients using isokinetic devices? Is there any study that used isokinetic devices in home-based knee rehabilitation? What is the state of the art regarding isokinetic knee rehabilitation, and which are the main possible advances your new device may bring?

Response 6:

Thank you for your professional comments regarding INTRODUCTION, we updated the manuscript according to your comments, with the following changes:

(1) Response to: “In addition, it would be important for the reader to get an idea at the introduction about previous literature on the use of isokinetic devices for knee rehabilitation and the limitation of these devices for home-based rehabilitation.”

a) Previous literature on the use of isokinetic devices for knee rehabilitation

Currently, isokinetic devices serve two primary purposes. The first is their use in long-term, structured isokinetic training programs designed to enhance muscle strength, thereby supporting the rehabilitation of various knee injuries, such as anterior cruciate ligament (ACL, 16) injuries, posterior cruciate ligament (PCL, 17) injuries, and osteoarthritis (OA, 18). The second purpose involves their application as dynamometers to quantitatively measure isokinetic muscle strength (14-15). This functionality enables precise assessments of rehabilitation progress, facilitates comparisons between different rehabilitation methods, and evaluates changes in a patient’s condition before and after an injury.

To support their use as rehabilitation devices, we updated the INTRODUCTION with the following content:

“Recent studies have demonstrated the effectiveness of isokinetic training for knee rehabilitation (16-18), such as enhancing muscle strength, improving stability, increasing postoperative walking speed, and reducing knee pain, in various knee conditions, including anterior cruciate ligament (ACL, 16), posterior cruciate ligament (PCL, 17) and osteoarthritis (OA, 18).”

To highlight their use as muscle strength measuring instruments, we updated the INTRODUCTION with the following:

“This constant speed characteristic naturally integrates a safety mechanism that reduces resistance if the patient experiences pain or discomfort (13), while also enabling continuous monitoring of rehabilitation progress through objective assessments of muscle strength (14-15).”

b) The limitation of these devices for home-based rehabilitation

To demonstrate the limitations of existing devices, we have updated a brief introduction to current hospital-based isokinetic training devices.

In the second paragraph of the INTRODUCTION, we added:

“Isokinetic training is usually administrated by a specialized rehabilitation facility that applies resistive forces for controlled movement of the injured knee. Existing commercial facilities, such as Biodex (19), Kineo (20), CONTEX MJ (21), IsoMed2000 (22), and research prototypes represent the state-of-the-art in isokinetic training technology (23-26) (Supplementary Table 1). These systems employ various operating principles in their driver modules, including high-power servo motors (19-23), magnetorheological (MR) fluids (24), electrorheological (ER) fluids (25), and electromagnetic powder brake (26) (Fig. 1a).”

Table S1 | Comparison between existing hospital-based and our home-based rehabilitation robot

	Research	Example/Picture	Power	Weight	Size	Human-machine Interface	Price
Commercial Product (Hospital-based)	Biodex ^[19]		Immovable (230V, 3450W)	612 kg	2.92 (m ²)	22" flat panel touchscreen	\$57,765
	Kineo ^[20]		Immovable (230V, 1600W)	345 kg	2.79 (m ²)	2×15.6" touch-screen displays	\$49,500
	CONTEX MJ ^[21]		Immovable (230V, 2300W)	350 kg	1.48 (m ²)	Display	Est. \$50000
	IsoMed2000 ^[22]		Immovable (380V, 8000W)	650 kg	2.6 (m ²)	Display	Est. \$50000
This Study (Home-based)			Movable (Power-free)	52 kg	0.82 (m ²)	6.67" Smart Phone	<\$1,000

In conclusion, the limitations of current hospital-based isokinetic training devices in enabling home-based rehabilitation are their bulk, weight, high energy consumption, and high cost. A key contributing factor is that all existing isokinetic devices **require a high-power driver module and external hardware for electricity supply**. However, if an isokinetic training device could be developed to operate without an external power source and harvest the training energy from human dynamics to power itself, it could provide a potential solution to the current limitation in this area.

We introduced “*the limitation of these devices for home-based rehabilitation*” in the second paragraph of the INTRODUCTION, the updated content is as follows:

“By modulating the electrical current, the driver produces an adjustable damping force that precisely counteracts knee motion. However, the employment of a high-power driver module and external electricity usually involves several external hardware (e.g., energy supply module, power transformer, and control cabin), making these facilities bulky, heavy, energy-consuming, and expensive. For instance, Biodex system works under 230 V power, consumes 3450 W, weighs over 600 kg, and costs over \$50,000 (19). As a result, access to isokinetic rehabilitation is largely limited to well-equipped hospitals or rehabilitation centers. There has been a growing yet unfulfilled need for compact isokinetic training systems that can be deployed for home-based rehabilitation at low cost, especially in resource-limited areas.”

(2) Response to: “How many studies have been conducted to rehabilitate knee patients using isokinetic devices? Is there any study that used isokinetic devices in home-based knee rehabilitation?”

a) How many studies have been conducted to rehabilitate knee patients using isokinetic devices?

We searched the keywords “*isokinetic knee rehabilitation*” on the *Web of Science*, finding 1972 studies.

Fig. R2. Screenshot of the Web of Science search results for the keywords

“isokinetic knee rehabilitation”.

While the majority of studies focus on clinical trials, rehabilitation, and orthopedics, only **195 (10%)** of the studies address the engineering aspect.

Fig. R3. “Refine by Research Areas” for the search keywords “isokinetic knee home” in the Web of Science.

b) Is there any study that used isokinetic devices in home-based knee rehabilitation?

To the best of our effort, we have not identified any studies that utilize isokinetic training devices in home-based rehabilitation. After searching for the keywords “isokinetic knee rehabilitation home” on the *Web of Science*, we found only 71 relevant studies. Upon reviewing these studies, we found that none specifically focused on the use of isokinetic training devices in a home-based rehabilitation setting.

Fig. R4. Screenshot of the Web of Science search results for the keywords “isokinetic knee rehabilitation home”.

(3) Response to: “What is the state of the art regarding isokinetic knee rehabilitation, and which are the main possible advances your new device may bring?”

a) What is the state of the art regarding isokinetic knee rehabilitation

Studies on isokinetic training primarily focus on the use of hospital-based isokinetic devices to explore their clinical applications, highlighting the effectiveness of these devices in the diagnosis, assessment, and rehabilitation of various knee injuries (16-18).

The section in the INTRODUCTION detailing the effectiveness of isokinetic training has been updated as follows:

“Particularly, isokinetic training denotes dynamic muscular contractions wherein movement speed is a constant, ensuring that muscles exert maximal tension throughout the entire range of joint motion (8-12). This constant speed characteristic naturally integrates a safety mechanism that reduces resistance if the patient experiences pain or discomfort (13), while also enabling continuous monitoring of rehabilitation progress through objective assessments of muscle strength (14-15). Recent studies have demonstrated the effectiveness of isokinetic training for knee rehabilitation (16-18), such as enhancing muscle strength, improving stability, increasing postoperative walking speed, and reducing knee pain, in various knee conditions, including anterior cruciate ligament (ACL, 16), posterior cruciate ligament (PCL, 17) and osteoarthritis (OA, 18).”

b) Which are the main possible advances your new device may bring?

Currently, isokinetic knee training is predominantly limited to hospital-based rehabilitation settings (19-22). Numerous studies have relied on commercial, hospital-based isokinetic devices for the diagnosis, assessment, and rehabilitation of various knee injuries (16-18). However, the home-based rehabilitation is a growing yet unfulfilled area.

We listed the possible advances our new device may bring in the second paragraph of INTRODUCTION:

“As a result, access to isokinetic rehabilitation is largely limited to well-equipped hospitals or rehabilitation centers. There has been a growing yet unfulfilled need for compact isokinetic training systems that can be deployed for home-based rehabilitation at low cost, especially in resource-limited areas.”

A thorough search on the Web of Science using the keywords “isokinetic knee rehabilitation home” yielded 71 results, **but none specifically focused on the use of isokinetic training devices in home-based settings.**

Fig. R5. Screenshot of the Web of Science search results for the keywords “isokinetic knee rehabilitation home”.

Given the safety, effectiveness, and continuous monitoring capabilities of isokinetic training, there is a clear demand for home-based isokinetic knee rehabilitation. **The primary objective of our device is to fill this gap, facilitating the transition from hospital-based, bulky, heavy, energy-consuming, and expensive isokinetic knee rehabilitation systems to a more accessible home-based solution.** A comparison between our device and current hospital-based devices is presented in **Table S1**.

Table S1 | Comparison between existing hospital-based and our home-based rehabilitation robot

	Research	Example/Picture	Power	Weight	Size	Human-machine Interface	Price
Commercial Product (Hospital-based)	Biodex ^[19]		Immovable (230V, 3450W)	612 kg	2.92 (m ²)	22" flat panel touchscreen	\$57,765
	Kineo ^[20]		Immovable (230V, 1600W)	345 kg	2.79 (m ²)	2×15.6" touch-screen displays	\$49,500
	CONTREX MJ ^[21]		Immovable (230V, 2300W)	350 kg	1.48 (m ²)	Display	Est. \$50000
	IsoMed2000 ^[22]		Immovable (380V, 8000W)	650 kg	2.6 (m ²)	Display	Est. \$50000
This Study (Home-based)			Movable (Power-free)	52 kg	0.82 (m ²)	6.67" Smart Phone	<\$1,000

Implementing home-based training offers several significant benefits:

First, for the patients, it **eliminates the time and costs** associated with traveling to the hospital for rehabilitation (such as travel time, waiting in queues, and the need for family accompaniment). Moreover, it removes the inconvenience of hospital appointment scheduling, allowing patients to undergo rehabilitation conveniently at home at any time.

Second, for hospitals, this device proposes a potential **solution to the shortage of**

rehabilitation resources. The number of patients who can receive rehabilitation in hospitals is limited, yet the demand is substantial, leading to a scarcity of healthcare resources. If this device can offer patients another rehabilitation option—small in size, highly effective, low-cost, and suitable for home-based use—it can significantly alleviate the resource constraints in hospitals. Additionally, it could reduce the workload of rehabilitation therapists.

Lastly, our proposed isokinetic training device utilizes a phone-based Human-Machine Interface (HMI). All training data is uploaded to the cloud, enabling physicians to perform remote consultations. This ensures the effectiveness of the training and allows for timely adjustments to the rehabilitation plan by the physician.

Reference:

8. Moffroid, M. et al. A study of isokinetic exercise. *Phys. Ther.* **49**, 735-747 (1969).
9. Verrill, D. et al. Resistive exercise training in cardiac patients. *Sports Med.* **13**, 171-193 (1992).
10. Hislop, H. J. & Perrine, J. The isokinetic concept of exercise. *Phys. Ther.* **47**, 114-117 (1967).
11. Thistle H. G. et al. Isokinetic contraction: a new concept of resistive exercise. *Arch. Phys. Med. Rehabil.* **48**, 279-282 (1967).
12. Cabri J., Isokinetic strength aspects in human joints and muscles. *Appl. Ergon.* **22**, 299-302 (1991).
13. Jee, Y., Usefulness of measuring isokinetic torque and balance ability for exercise rehabilitation. *J. Exerc. Rehabil.* **11**, 65-66 (2015).
14. Goetz, J., et.al, Isokinetic knee muscle strength comparison after enhanced recovery after surgery (ERAS) versus conventional setup in total knee arthroplasty (TKA): a single blinded prospective randomized study. *J. Exp. Orthop.* **10**, 44-56 (2023).
15. Wang, S. et.al, Analysis of Isokinetic Strength Test in Arthroscopic Meniscus Suture to Improve Knee Joint Strength and Function, *Altern. Ther. Health Med.* **29**, 416-424, (2023).
16. Liu, H. et.al, Effect of isokinetic training of thigh muscle group on graft remodeling after anterior cruciate ligament reconstruction. *Chin. J. Reparative Reconstr. Surg.* **33**, 1088-1094 (2019).
17. Kim, H., KOOK, K., Lee, J., Lee, S., The Effects of a Rehabilitation Exercise Program After Posterior Cruciate Ligament (PCL) Reconstruction Between Genders.

Asian J. Kinesiol. **24**, 39-45 (2022).

18. Rosa, U. et.al, Comparison of the Effectiveness of Isokinetic Exercise Vs Isometric Exercise Performed at Different Angles in Patients with Knee *Osteoarthritis*. *Reumatol Clin.* **8**, 10-4 (2012).

Comment 7:

Line 95. Please better define what you mean by the words “Highly integrated” in the subtitle. Are the other isokinetic devices not highly integrated?

Response 7:

Thanks for your comment, we updated the subtitle:

Previous subtitle

“Highly integrated isokinetic training robot and its working principle”

Updated subtitle

“Power-free isokinetic training robot and its working principle”

Our proposed isokinetic training device is more highly integrated compared to current hospital-based isokinetic training devices, and this can be explained in three aspects:

(1) Highly Integrated Controller

The current control methods require an external energy source to supply the necessary current. However, our proposed isokinetic training device operates without external power, meaning it is power-free. By employing the motor winding short (MWS) control method, we have replaced the large control modules typically found in hospital-based devices with a self-designed, highly integrated control board (**Fig. 1a**). This modification significantly reduces both the size and cost of the device.

The updated fourth paragraph of the INTRODUCTION now includes the following description of the proposed **home-based** isokinetic training device features:

“Notably, this strategy can provide electromagnetic damping torque of tens of N·m with energy consumption lower than several watts (28-29). For instance, a transtibial prosthesis utilizing the MWS method achieved precise control with a small energy consumption of 1.5 W (28).”

In contrast, we give an example of a **hospital-based** isokinetic training device, and updated the content in the second paragraph of INTRODUCTION:

“For instance, Biodex system works under 230 V power, consumes 3450 W, weighs over 600 kg, and costs over \$50,000 (19).”

(2) Highly Integrated Energy Module

The isokinetic device proposed in this study harvests training energy to achieve

power-free isokinetic training. In contrast to other driving methods (19-26), which rely on energy-consuming modules and external power supplies, our proposed isokinetic training device does not require any external power source. The highly integrated energy circuit is embedded in a thin, compact control board (see Figure 1). This highly integrated design allows us to achieve power-free isokinetic training.

(3) Highly integrated Human machine interface (HMI)

Most existing HMIs are desktop-based, requiring separate control cabinets that occupy considerable space, incur high costs, and lack mobility. In contrast, the proposed device employs a phone-based HMI, allowing patients to monitor their training status using only a smartphone. Additionally, all training data can be uploaded to the cloud, enabling potential remote consultations by experienced physicians. This HMI offers greater convenience, lower costs, and a smaller footprint.

Through the “highly integrated” of these three aspects, our proposed isokinetic training device successfully enables home-based training.

Fig. 1 | Proposed power-free knee rehabilitation robot, realization and comparison.

(a) Existing hospital-based rehabilitation devices. These devices are bulky and need additional power supplies. (b) Proposed home-based isokinetic training robot. The proposed isokinetic training robot is power-free and highly integrated. (c) Explosion Assembly. During the training, the rotation center of the knee joint and the motor are concentric. (d) Isokinetic realization and energy regeneration. The isokinetic training robot can generate energy while training. A desired constant velocity is predetermined. The velocity is monitored via angle sensors and the motor output resistance is modulated to ensure the actual velocity closely approximates the predetermined desired velocity. The controlled resistance is adjusted and the negative mechanical work during training is harvested. The harvested energy is subsequently channeled to power the control circuit, sensors, and other usage. (e) Comparison. A comparison between commercial isokinetic training robots and our proposed isokinetic training robot. The consumed power and weight of our robot are both dramatically reduced.

Reference:

19. Biodex System 4 Pro™, Biodex Medical System; <https://www.biodex.com/physical-medicine/products/dynamometers/system-4-pro>.
28. Yuan, K., Wang, Q. & Wang, L. Energy-Efficient Braking Torque Control of Robotic Transfemoral Prosthesis. *IEEE Trans. Mechatron.* **22**, 149-160 (2017).
29. Wang, Q. et al. Walk the Walk: A Lightweight Active Transfemoral Prosthesis, *IEEE Robot. Autom. Mag.* **22**, 80-89 (2015).

Comment 8:

Lines 125-155. This section on “Energy regeneration during isokinetic training” appears to me out of place, as it does not show any result, but explains how the system regenerates energy during the isokinetic training, and **therefore should be moved to the Methods Section.**

Response 8:

Thanks for your comment! **We have moved all content in the section entitled “Energy regeneration during isokinetic training” from RESULTS to METHODS.** Accordingly, we have relocated the previous **Figure 3** in the main text to the **Supplemental Materials** as **Figure S4**. This figure provides a biomedical model and an energy regeneration principle for isokinetic training. The restructured figure and content are as follows:

The restructured figure and content are as follows:

Energy regeneration during isokinetic training

The proposed dynamic energy regeneration method enables power-free functionality and isokinetic training, simultaneously. The knee joint is regarded as a bio-mechanical system comprising two skeleton bones, the femur and the tibia, as well as two

antagonistic muscles, quadriceps femoris (dominating knee lifting) and hamstring muscle (dominating knee retracting), as shown in **Supplementary Fig. S4a**. In this system, after the gravitational compensation, the torque T from human dynamics is approximately modeled as,

$$T = F_s \cdot l_s \approx T_m \cdot \beta \cdot \widetilde{W}_l(\theta) \cdot \widetilde{W}_v(\dot{\theta}) \quad (1)$$

where F_s is a force exerted by the trainee, l_s is the corresponding moment arm. In detail, F_Q and l_Q represent quadriceps femoris contraction, F_H and l_H represent hamstring muscle contraction, T_m is the maximum torque that muscle can exert, β ($0 \leq \beta \leq 1$) is a normalized muscle activation level, $\widetilde{W}_l(\theta)$ is a torque-angle factor and $\widetilde{W}_v(\dot{\theta})$ is a torque-velocity factor. The detailed principle is shown in **Supplementary Fig. S5** and Supplementary Information Text.

If the motor windings were fully shorted, indicating without any control and all low-side MOSFETs are on (**Fig. S4b**), the maximum current I_{max} , due to the torque T ,

$$I_{max} = \frac{T_m \cdot \beta \cdot \widetilde{W}_l(\theta) \cdot \widetilde{W}_v(\dot{\theta})}{k} \quad (2)$$

where k is the motor torque constant.

If the duration of the short motor windings was controlled dynamically (see **Fig. S4b**), as duty cycle D , the accommodative resistance torque τ_{res} is

$$\tau_{res} = kI_a = kf(D) \leq kI_{max} \quad (3)$$

where I_a is the phase current, which is associated with D , the maximum I_a is equal to I_{max} .

To realize isokinetic training, a proportional-integral (PI) controller is to adjust D based on errors,

$$D = K_p(\dot{\theta} - \bar{\theta}) + K_I(\dot{\theta} - \bar{\theta}) \quad (4)$$

where K_p is a proportional coefficient, K_I is an integral coefficient and $\bar{\theta}$ is the desired velocity.

Accompanying isokinetic training, dynamic energy regeneration can be obtained, and the regenerative power is

$$P_{reg} = \frac{(\int_0^{t_2} I_{charge} dt)^2 - (\int_0^{t_1} I_{charge} dt)^2}{2C(t_2 - t_1)} \quad (5)$$

where I_{charge} is the charging current. When low-side MOSFETs are off, I_{charge} is equal to I_a , when low-side MOSFETs are on, I_{charge} is 0 (**Fig. S4b**), C is the capacitance and t_1 and t_2 are different timings. The detailed charging circuit principle is shown in **Supplementary Fig. S5** and **Supplementary Fig. S6**.

Fig. S4 | Energy regeneration during isokinetic training. (a) Lifting and retracting torque model. Quadriceps femoris contraction force F_Q dominates the lifting force, with the corresponding moment arm l_Q . Hamstring muscle contraction force F_H dominates the retracting force, with the corresponding moment arm l_H . β is a normalized muscle activation level. G is the gravitational force, and l_G is the corresponding moment arm. **(b)** Energy regeneration circuit. The low-side MOSFETs are switched ON/OFF. When the MOSFETs are OFF, the charging current I_{charge} equals the phase current I_a , and the current flows into the ultracapacitor. When the MOSFETs are ON, although the induced phase current exists, the charging current $I_{charge} = 0$.

Comment 9:

Lines 157-178. I would separate this section into two different sections: (1) In lines 158-172 you are describing the methodology used to position the subject in the device and the procedures for executing the isokinetic knee extension-flexion exercises. Therefore, this belongs to the Methods section. (2) In lines 172-178 you show results that illustrate the functioning of the device and how the training is evaluated and how energy is regenerated. Therefore, this part belongs to the Results section.

Response 9:

Thanks for your professional comments. In accordance with your suggestion, we have divided the previous section titled “*Demonstration of isokinetic training*” into two distinct parts: **one retained as “*Demonstration of isokinetic training*” now located in**

RESULTS, and the other titled “*Procedure of isokinetic training*” which has been moved to METHODS.

In lines 158-178, we introduce the isokinetic training with a demonstration figure, **Fig. 3**, consisting of three subfigures: **Fig. 3a**, **Fig. 3b**, and **Fig. 3c**. **Fig. 3a** illustrates the preparation steps before each training trial, while **Fig. 3b** shows experimental pictures depicting the procedure of one repetition. Both **Fig. 3a** and **Fig. 3b** are intended to demonstrate the experimental process, providing context for the experimental data presented in **Fig. 3c**.

In the updated “*Demonstration of isokinetic training*” in RESULTS, the content originally spanning lines 158-172 has been condensed, with a focus solely on presenting the experimental basis for **Fig. 3c**.

The updated content for “*Demonstration of isokinetic training*” in the RESULTS is as follows:

Demonstration of isokinetic training

To evaluate the performance of our isokinetic training robot, we conducted a study with 10 post-surgical subjects and 10 healthy subjects (See **Supplementary Table S2, Table S3**). Preparation for the isokinetic training involves four steps as outlined in **Fig. 3a** (Detailed description see METHODS). The training protocol consists of eight repetitions of lifting and retracting movements in one cycle. Experimental pictures are shown in **Fig. 3b**. **Figure 3c** presents representative isokinetic training data from a post-surgical subject. A constant training velocity of $\omega = 60^\circ/s$ is maintained during both lifting and retracting processes (33). During the training, subjects are required to exert maximum muscle strength to lift and retract the calf. Notably, the training device is designed not to assist leg motion but to provide accommodative resistance if the velocity exceeds $60^\circ/s$, ensuring the training intensity. Within each repetition, the training velocity exhibits a pattern of quick acceleration, stability at target velocity, and rapid deceleration, in which the constant velocity time occupies about 70% of training (**Fig. 3c, Velocity**). The torque displays a similar pattern with a gradual decline over the training course due to muscle fatigue (**Fig. 3c, Torque**). As the training continues, the regenerated energy accumulates in both the lifting and retracting processes (**Fig. 3c, Regenerated Energy**). A demonstration video is shown in **Supplementary Movie S2**.

The detailed training procedure has been moved to the METHODS and is now titled “Procedure of isokinetic training” with the following content:

Procedure of isokinetic training

All demonstration experiments involved four steps as demonstrated in **Fig. 3a**. The first step is positioning the leg to ensure that the main arm of the robot is parallel to the calf, which allows for the precise measurement of the knee joint angle θ (40). The second step is fastening the belt to stabilize the body. This stabilization minimizes the movement of the hip joint, isolating the knee for target training only. The third step is connecting to the smartphone via Bluetooth, establishing the human-machine interface. The fourth step is calibration. The subject lifts and retracts his calf to the maximum and

minimum angles, establishing a range of motion benchmark for the training protocol. In all trials, a training speed of 60°/s was used. Previous studies show that 60°/s is effective in various rehabilitation aspects, contributing to improvements in muscle strength improvement and functional recovery (41-42).

Fig.3 | Demonstration of isokinetic training. (a) Preparation steps. Step 1: Position the leg and align the knee joint with the rotation axis. Step 2: Fasten the belt to stabilize the body. Step 3: Connect Bluetooth with the smartphone. Step 4: Calibration by testing the maximum and minimum angle of the subject. (b) Experimental pictures showing that a post-surgical subject lifts and retracts the calf during the training. (c) The training data of knee angle, velocity, torque, and regenerated energy over 8 repetitions of lifting-retracting movement.

Comment 10:

Lines 168-169. Why did you choose the single angular velocity of 60°/s? In addition, what do you mean by “optimizing the training outcomes”? It would be interesting if your device had the possibility of establishing different angular velocities allowing setting these different angular velocities for each antagonist muscle group. According to the torque-velocity relationship of concentric contractions, the lower the velocity the higher is the muscle force generating capacity. In this way you would be able to

establish different external loads and different mechanical work for each muscle group, thereby individualizing your training outcomes.

Response 10:

(1) **Response to:** “Why did you choose the single angular velocity of 60°/s?”

a) Clinical evidence supporting 60°/s training speed

Clinical trials indicate that a training speed of 60°/s is a commonly used rehabilitation velocity, which is considered effective for promoting muscle strength gains. **This speed has been shown to be most effective in enhancing muscle strength during isokinetic training (41-42).**

We updated the METHOD by adding the following content:

“In all trials, a training speed of 60°/s was used. Previous studies show that 60°/s is effective in various rehabilitation aspects, contributing to improvements in muscle strength improvement and functional recovery (41-42).”

b) Consistency with Previous Clinical Trials for Evaluation

To evaluate the effectiveness of the proposed device at a clinical level, we conducted **the same clinical trial protocol** used in previous studies (isokinetic training **speed of 60°/s**, 8-10 repetitions per set, training duration of 6 weeks) and compared the outcomes of subjects using our proposed device with those using the commercial device referenced in (33). The use of 60°/s is based on the study referenced in (33), as it allows for controlled variables and provides a meaningful comparison of the outcomes across both devices.

We stated that “the same training program was carried out” to achieve consistency in METHODS:

“Notably, the CSA growth rate of the proposed power-free isokinetic training robot 5.9% (**Fig. 7a**) is much higher than the documented 3.7% value by a commercial rehabilitation device (CONTREX MJ, PHYSIOMED, Germany) **where the same training program was carried out (33, See Fig. 7a).**”

(2) **Response to:** “In addition, what do you mean by ‘optimizing the training outcomes’.”

In this context, “optimizing the training outcomes” refers to achieving potentially better training effects at an isokinetic speed of 60°/s compared to other commonly used speeds (41-42). To avoid any potential misunderstanding by readers, we have **removed the phrase “optimizing the training outcomes”**.

Previous content:

A constant training velocity of $\omega = 60^\circ/\text{s}$ is maintained during both lifting and retracting

processes to optimize the training outcomes (33).

Updated content:

A constant training velocity of $\omega = 60^\circ/\text{s}$ is maintained during both lifting and retracting processes (33).

(3) **Response to:** “It would be interesting if your device had the possibility of establishing different angular velocities allowing setting these different angular velocities for each antagonist muscle group.”

The proposed home-based isokinetic training device is capable of supporting training at various angular velocities. We conducted additional experiments at **speeds of $90^\circ/\text{s}$ and $120^\circ/\text{s}$** , with training demonstrations provided below. The accompanying speed and angle curves demonstrate that the device can successfully achieve and regulate different angular velocities, further highlighting its versatility in accommodating diverse training protocols.

Fig. R6. Representation of Isokinetic Training Performed at Varying Speeds

Reference:

33. Zhang, X. et al. Impact of whey protein isolate and eccentric training on quadriceps mass and strength in patients with anterior cruciate ligament rupture: A randomized controlled trial. *J. Rehabil. Med.* **52**, 1-8 (2020).

41. Jan M. H., Gain of muscle torque at low and high speed after isokinetic knee strengthening program in healthy young and older adults. *J. Formos. Med. Assoc.* **97**, 339-344 (1998)

42. Cengizel, Ç., & Pekel, H., Can isokinetic strength training be an alternative to machine-based strength training? *Medicina dello. Sport.* **76**, 495-506 (2023).

Comment 11:

Lines 169-172. How do you know if the patient is exerting maximal knee extensor torque during the exercise if the device only provides the resistance to control the angular velocity of the patient’s limb? I imagine that the patient would be able to exert a knee extension faster than $60^\circ/\text{s}$, but not necessarily at maximal effort. In this case the system will limit the angular velocity but will not be able to determine if maximal effort

was performed. In addition, determining the angular velocity of the knee extensor exercise into a specific velocity does not give you training intensity. To get the **training intensity** at a constant angular velocity you will need to determine the work performed over time or the area under each torque-velocity curve of the whole protocol, and changing the protocol time will allow you to change the training intensity.

Response 11:

We sincerely appreciate your professional comments.

- (1) **Response to:** “How do you know if the patient is exerting maximal knee extensor torque during the exercise if the device only provides the resistance to control the angular velocity of the patient’s limb? I imagine that the patient would be able to exert a knee extension faster than 60°/s, but not necessarily at maximal effort. In this case the system will limit the angular velocity but will not be able to determine if maximal effort was performed. In addition, determining the angular velocity of the knee extensor exercise into a specific velocity does not give you training intensity.”

Firstly, while it is true that the device can achieve the target speed of 60°/s even if a patient does not exert maximal effort during isokinetic training, **however, the resulting peak torque would vary**. This variation can be observed in real-time through the system. Using a smartphone-based human-machine interface (HMI), the training torque is continuously monitored, and real-time feedback is provided to patients, enabling them to view their training data. This feedback acts as positive reinforcement, encouraging patients to consistently exert maximal effort.

Moreover, in all our clinical trials, **patients were explicitly instructed to exert maximum strength during each repetition, with consistent encouragement provided throughout the training sessions**. Additionally, patients typically possess strong intrinsic motivation for rehabilitation, which fosters high adherence to training requirements and a commitment to maintaining maximal effort.

Finally, to illustrate the variation in torque and training intensity, we have included data in **Figure 6 to demonstrate torque variation**. Furthermore, we have added **Figure S4 to depict changes in work**. These data offer quantifiable evidence of the progressive increase in both training torque and intensity throughout the rehabilitation process.

Fig. 6 | Evaluation of muscle strength after a 6-week rehabilitation program. (a) The maximum torque of 10 subjects with 6 weeks. Positive values represent lifting torque by quadriceps and negative values denote retracting torque by hamstrings. **(b)** The maximum torque of 10 post-surgical subjects in one repetition in the first session. **(c)** The maximum torque of 10 post-surgical subjects in the last session. **(d)** Comparison of averaged maximum torque in the first and last session. The left and right bars indicate torques in the first and last training sessions, respectively, with red bars representing lifting and blue bars for retracting. The percentage values indicate the increase from the first to the last session. * refers to p-value < 0.05 by one-tail t-test.

(2) **Response to:** “To get the training intensity at a constant angular velocity you will

need to determine the work performed over time or the area under each torque-velocity curve of the whole protocol, and changing the protocol time will allow you to change the training intensity.”

In response to your valuable comment, **we have added the mean work values in one repetition for each of the 10 post-surgical subjects across all 12 training sessions**, incorporating both lifting and retracting work. The detailed results of this work are presented in **Fig. S9**, which is also provided below.

Fig. S9 | Evaluation of training intensity changes after a 6-week rehabilitation program. (a)-(b) Mean lifting and retracting work per repetition for 10 subjects over 6 weeks. (a) Subject 1-5. (b) Subject 6-10. (c)-(d) Progression of isokinetic training work from the first to the last week. (c) Quadriceps work. (d) Hamstrings work.

Comment 12:

Lines 197-198. Please provide some references giving support for this sentence regarding CSAs being widely adopted as the principal metric for assessing muscle growth and recovery.

Response 12:

Thanks for your comments, we updated several references to support the statement “CSAs being widely adopted as the principal metric for assessing muscle growth and recovery”.

Muscle cross-sectional area (CSA) is an important indicator in rehabilitation assessments (34) and is widely used in sports science as well as in the post-surgery rehabilitation of knee surgeries (35). Studies have shown a significant positive correlation between CSA and muscle volume, mass, and strength (36), indicating CSAs can be adopted as the principal metric for assessing muscle growth and recovery.

References:

34. An, K. N., Linscheid, R. L., Brand, P. W., Correlation of physiological cross-sectional areas of muscle and tendon. *J. Hand. Surg. Br. Eur. Vol.* **16**, 66-67 (1991).
35. Franchi, M V., et al. Muscle thickness correlates to muscle cross-sectional area in the assessment of strength training-induced hypertrophy, *Scand. J. Med. Sci. Sports* **28**, 846-853 (2018).
36. Chen L., et al. How Do Muscle Function and Quality Affect the Progression of KOA? A Narrative Review. *Orthop Surg.* **16**, 802-810 (2024).

Comment 13:

Lines 217-232. As the knee extensors and the hamstrings have different muscle architecture and different (opposite) functions regarding the anterior/posterior tibial translation, and as the patients had different knee ligament tears, how do you define the best load for each muscle group from each patient? For example, for an ACL rupture patient, you may want to strengthen more the hamstrings to avoid tibial anterior translation, whereas for a PCL rupture the hamstrings may not be the best choice. And you have patients in your sample with different clinical problems regarding the knee ligaments that probably need different loads for each knee muscle group. Therefore, you need to be careful with the interpretation of your knee flexor and extensor increments, which should be evaluated critically in your discussion.

Response 13:

Thanks for your insightful comment. The reason we did not customize different training intensities for various muscle groups based on specific knee clinical problems in our previous studies is as follows:

In clinical rehabilitation, the protocols during the early post-surgical period (typically within the first three months) differ based on the specific clinical problems, as the ruptured ligaments are generalized have not yet fully healed. For example, in the case of an anterior cruciate ligament (ACL) rupture, the rehabilitation strategy focuses on preventing tibial anterior translation, which requires encouraging the downward motion of the calf and prioritizing hamstring strengthening. Conversely, for posterior cruciate ligament (PCL) rupture, the objective is to elevate the calf, thereby necessitating greater emphasis on quadriceps training. Therefore, for post-surgical patients with different knee conditions, it is crucial to adjust the training loads for each muscle group during the early recovery phase, as you have suggested.

However, all of our post-surgical subjects were **at least 3 months post-surgery, a timeframe generally considered sufficient for ligament healing**, as discussed and recommended by professional physicians. MRI scans revealed varying degrees of atrophy in both the quadriceps and hamstrings among the post-surgical subjects. **Consequently, our training protocol was specifically designed to target both muscle groups simultaneously, with the goal of effectively rehabilitating both the quadriceps and hamstrings. Individualized differences were not prioritized as a primary consideration in this approach.**

We acknowledge the importance of personalized training programs to more specifically address the needs of patients based on their individual knee clinical problems. **As such, we have included this consideration in the *Limitations* section in DISCUSSION**, highlighting the need for more targeted load adjustments and individualized rehabilitation plans for different knee clinical issues.

The following content has been added to the Limitation:

“Fourth, the study employed a uniform training protocol for all post-surgical subjects. Customizing training loads based on individual clinical conditions could potentially enhance rehabilitation outcomes and will be explored in future research.”

Comment 14:

Lines 233-248. This part of the results seems to me more appropriate for the Discussion Section (not the Results Section), as here you are comparing and discussing your results with the results from previous literature on the theme.

Response 14:

Thanks for your comment. **We redefined the structure and have moved the “Efficacy comparison between our study and commercial isokinetic robots” to DISCUSSION.** This part primarily discusses about our clinical results in comparison with the clinical outcomes of existing hospital-based isokinetic training device.

The related content and figure in the manuscript are attached below:

Efficacy comparison between our study and commercial isokinetic robots

To assess the rehabilitation outcomes, the quadriceps CSA growth rate (**Fig. 7a**) and lifting torque increase (**Fig. 7b**) of the proposed robots and references utilizing the commercial isokinetic training robots were compared. Notably, the CSA growth rate of the proposed power-free isokinetic training robot 5.9% (**Fig. 7a**) is much higher than the documented 3.7% value by a commercial rehabilitation device (CONTREX MJ, PHYSIOMED, Germany) where the same training program was carried out (33, See **Fig. 7a**). Quadriceps muscle CSA in the untrained leg demonstrated a negligible difference in both the proposed power-free robots and commercial devices compared with the trained leg (**Fig. 7a**). This result may indicate that the impact of unilateral isokinetic training on the contralateral untrained leg is insignificant. Additionally, routine exercises (e.g., walking, jogging, climbing stairs) do not provide as pronounced stimulatory growth as isokinetic training. Similarly, the significant torque increases utilizing the power-free robots 70% is higher than the documented results of 35.9% utilizing the commercial isokinetic training device (33, See **Fig. 7b**). Other experiments conducted using commercial isokinetic devices with the same isokinetic training speed (60 degrees per second) but varying training durations and cycles also did not demonstrate torque increases surpassing the results of our study (33, 37-39, See **Fig. 7b**).

Fig. 7 | Efficacy comparison between our study and commercial isokinetic robots.

(a) The CSA growth rate. **(b)** The torque increases.

Reference:

33. Zhang, X. et al. Impact of whey protein isolate and eccentric training on quadriceps mass and strength in patients with anterior cruciate ligament rupture: A randomized controlled trial. *J. Rehabil. Med.* **52**, 1-8 (2020).

37. Coyle, E. F. et al. Specificity of power improvements through slow and fast isokinetic training. *J. Appl. Physiol. Respir. Environ. Exerc. Physiol.* **51**, 1437-1442 (1981).

38 Wang, K. et al. Effect of isokinetic muscle strength training on knee muscle strength, proprioception, and balance ability in athletes with anterior cruciate ligament reconstruction: a randomised control trial. *Front. Physiol.* **14**, (2023).

39. Ewing Jr, J. L. et al. Effects of velocity of isokinetic training on strength, power, and quadriceps muscle fibre characteristics. *Eur. J. Appl. Physiol. Occup. Physiol.* **61**, 159-162 (1990).

Comment 15:

Usually in the Discussion Section, in addition to reminding the reader about the goals of the study and showing a summary of the main findings, authors should contrast the obtained results with those from previous studies from the existent literature. Your discussion does not have a single article from previous literature in the area of existent devices and there is no contrast of your results with the results obtained with these isokinetic devices in knee rehabilitation. Similarly, it does not show any mention to previous home-based isokinetic training in clinical rehabilitation of patients with knee joint problems and/or ligament reconstruction. Part of this discussion was observed in different sections of the manuscript, and therefore I suggest that you do restructure the manuscript so that methodological aspects are presented in the Methods Section, results are presented in the Results Section, and discussion and contrasts with the literature are presented in the Discussion Section.

Response 15:

Thanks for your comment. According to your comment, we have added additional content in DISCUSSION and restructured the manuscript by appropriately adjusting the content across the RESULTS, DISCUSSION, and METHODS sections.

(1) Response to: “Your discussion does not have a single article from previous literature in the area of existent devices and there is no contrast of your results with the results obtained with these isokinetic devices in knee rehabilitation.”

a) Comparison of realization of isokinetic control

To demonstrate the contrast in the realization of isokinetic control between our proposed home-based isokinetic training device and current commercial hospital-based devices, we have updated the DISCUSSION section by adding a comparison of the driver modules used in both systems.

The following content has been added to the first paragraph of the DISCUSSION:

“Conventional knee isokinetic training facilities are often characterized limited by their substantial size, weight, and high energy consumption. At the core of this limitation is the energy-intensive driver module, which is required to generate the controllable resistive torque. Existing driver modules rely on high-power servo motors (19-23), magnetorheological (MR) fluids (24), electrorheological (ER) fluids (25), and electromagnetic powder brakes (26). All of these require continuous and precise electrical control—whether it involves supplying large currents, maintaining high voltages, or generating strong magnetic fields. These characteristics limit the rehabilitation training exclusively to clinical settings such as hospitals and rehabilitation centers.”

b) Comparison of clinical efficiency

Besides the comparison of isokinetic device characteristics, we also compared the rehabilitation outcomes for patients using our proposed home-based isokinetic training device with those using commercial isokinetic training robots in clinical settings. **This part was previously placed in RESULTS and has now been moved to DISCUSSION, following your comment.**

The content regarding clinical efficiency comparison is as follows:

Efficacy comparison between our study and commercial isokinetic robots

To assess the rehabilitation outcomes, the quadriceps CSA growth rate (**Fig. 7a**) and lifting torque increase (**Fig. 7b**) of the proposed robots and references utilizing the commercial isokinetic training robots were compared. Notably, the CSA growth rate of the proposed power-free isokinetic training robot 5.9% (**Fig. 7a**) is much higher than the documented 3.7% value by a commercial rehabilitation device (CONTREX MJ, PHYSIOMED, Germany) where the same training program was carried out (33, See **Fig. 7a**). Quadriceps muscle CSA in the untrained leg demonstrated a negligible difference in both the proposed power-free robots and commercial devices compared with the trained leg (**Fig. 7a**). This result may indicate that the impact of unilateral isokinetic training on the contralateral untrained leg is insignificant. Additionally, routine exercises (e.g., walking, jogging, climbing stairs) do not provide as pronounced stimulatory growth as isokinetic training. Similarly, the significant torque increases utilizing the power-free robots 70% is higher than the documented results of 35.9% utilizing the commercial isokinetic training device (33, See **Fig. 7b**). Other experiments conducted using commercial isokinetic devices with the same isokinetic training speed (60 degrees per second) but varying training durations and cycles also did not demonstrate torque increases surpassing the results of our study (33, 37-39, See **Fig. 7b**).

Fig. 7 | Efficacy comparison between our study and commercial isokinetic robots. (a) The CSA growth rate. (b) The torque increases.

(2) Response to: “Similarly, it does not show any mention to previous home-based isokinetic training in clinical rehabilitation of patients with knee joint problems and/or ligament reconstruction.”

We acknowledge that our manuscript does not include any references to home-based isokinetic training devices for the clinical rehabilitation of patients with knee joint problems and/or ligament reconstruction. This omission is due to the fact that, despite our best efforts, **we were unable to identify any studies utilizing isokinetic training devices in home-based rehabilitation.** A search in the *Web of Science* database using the keywords “isokinetic knee rehabilitation home” yielded only 71 relevant studies. Upon a thorough review of these studies, none specifically addressed the use of isokinetic training devices in a home-based rehabilitation context.

Fig. R7. Screenshot of the Web of Science search results for the keywords “isokinetic knee rehabilitation home”.

Comment 16:

Lines 276-288. I suggest that you add, in your limitations paragraph, all the other limitations, which I presented/discussed above and below, here in this final paragraph of the discussion.

Response 16:

Thanks for your valuable comments. We **have added two more limitations** to the Limitation section.

The full content of the Limitation section is as follows, with the newly added parts highlighted in blue:

Limitations

Firstly, our proposed power-free isokinetic training robot is primarily suitable for patients in the middle to late stages of rehabilitation. In other words, the device is not intended for patients in the early postoperative, bedridden stages, and they need to possess basic walking abilities. Secondly, our current design is specifically targeted at post-knee surgery rehabilitation training. Knee rehabilitation primarily focuses on training the quadriceps muscle and hamstrings muscle, which are major muscle groups in the thigh. As for isokinetic training of other joints, such as the ankle, elbow, and shoulder joints, whether the proposed isokinetic training robot can approve power-free has not yet been experimentally validated. **Third, the present system supports only concentric isokinetic contractions. Future work will aim to incorporate additional training modes, such as eccentric contractions, to broaden its rehabilitative capabilities.** Fourth, the study employed a uniform training protocol for all post-surgical subjects. **Customizing training loads based on individual clinical conditions could potentially enhance rehabilitation outcomes and will be explored in future research.** Finally, as we

are currently in the experimental validation phase, we can only assess the manufacturing costs (See **Supplementary Table S10**). Due to uncertainties in marketing costs, including factors such as marketization, it is challenging to compare the costs of our device with the sale prices of commercial isokinetic training devices.

Comment 17:

The technical aspects of the robot are well described. However, it is not clear in the methods section why a single angular velocity was determined for the isokinetic training robot to be used in the rehabilitation program. Is there a technical limitation, for example, in the energy regeneration system which needs that a single angular velocity is established for the system to work properly?

Response 17:

(1) **Response to:** “However, it is not clear in the methods section why a single angular velocity was determined for the isokinetic training robot to be used in the rehabilitation program?”

a) Clinical evidence supporting 60°/s training speed

Clinical trials indicate that a training speed of 60°/s is a commonly used rehabilitation velocity, which is considered effective for promoting muscle strength gains. **This speed has been shown to be most effective in enhancing muscle strength during isokinetic training (41-42).**

We updated the METHOD by adding the following content:

“In all trials, a training speed of 60°/s was used. Previous studies show that 60°/s is effective in various rehabilitation aspects, contributing to improvements in muscle strength improvement and functional recovery (41-42).”

b) Consistency with Previous Clinical Trials for Evaluation

To evaluate the effectiveness of the proposed device at a clinical level, we conducted the same clinical trial protocol used in previous studies (isokinetic training speed of 60°/s, 8-10 repetitions per set, training duration of 6 weeks) and compared the outcomes of subjects using our proposed device with those using the commercial device referenced in (33). The use of 60°/s is based on the study referenced in (33), as it allows for controlled variables and provides a meaningful comparison of the outcomes across both devices.

We stated that “the same training program was carried out” to achieve consistency in METHODS:

“Notably, the CSA growth rate of the proposed power-free isokinetic training robot 5.9% (**Fig. 7a**) is much higher than the documented 3.7% value by a commercial rehabilitation device (CONTREX MJ, PHYSIOMED, Germany) **where the same**

training program was carried out (33, See Fig. 7a).”

(2) **Response to:** “Is there a technical limitation, for example, in the energy regeneration system which needs that a single angular velocity is established for the system to work properly”

Our system is capable of achieving isokinetic training across a range of speeds. We conducted additional experiments at **speeds of 90°/s and 120°/s**, with training demonstrations provided below.

Fig. R8. Representation of Isokinetic Training Performed at Varying Speeds

Reference:

33. Zhang, X. et al. Impact of whey protein isolate and eccentric training on quadriceps mass and strength in patients with anterior cruciate ligament rupture: A randomized controlled trial. *J. Rehabil. Med.* **52**, 1-8 (2020).

41. Jan M. H., Gain of muscle torque at low and high speed after isokinetic knee strengthening program in healthy young and older adults. *J. Formos. Med. Assoc.* **97**, 339-344 (1998)

42. Cengizel, Ç., & Pekel, H., Can isokinetic strength training be an alternative to machine-based strength training? *Medicina dello. Sport.* **76**, 495-506 (2023).

Comment 18:

A second important aspect that needs to be indicated is that the robot only allows knee flexor-extensor concentric isokinetic contractions, which is a limitation of the device when compared to the commercially available isokinetic machines that were presented in the manuscript, as eccentric and isometric contractions can be performed in these isokinetic machines and are important contraction modalities that are clinically important for rehabilitation programs.

Response 18:

Thank you for your comments. We have indeed focused primarily on knee flexor-extensor concentric isokinetic contractions in our current design, without incorporating eccentric contractions. As you suggested, in future research, we plan to enhance the

functionality of the isokinetic training device, such as the ability to perform eccentric contractions.

The following content has been added to the Limitation:

“Third, the present system supports only concentric isokinetic contractions. Future work will aim to incorporate additional training modes, such as eccentric contractions, to broaden its rehabilitative capabilities.”

Comment 19:

Why 10 patients? Have you performed a sample size calculation? If not, this needs to be added as a limitation in the discussion.

Response 19:

We have performed a sample size calculation, with the following steps,

$$N = 2 * \left[\left(t_{\frac{\alpha}{2}} + t_{\beta} \right) * \frac{S}{\delta} \right]^2$$

Given that $\alpha = 0.05$ and $\beta = 0.1$, the critical values $t_{\frac{\alpha}{2}}$ and t_{β} are calculated as:

$$t_{\frac{\alpha}{2}} = 1.96, \quad t_{\beta} = 1.28$$

From the reference 33, $S = 16.1$, $\delta = 32.3$. After calculation, we obtained $N > 5.21$. Therefore, the minimum sample size $N_{min} = 6$. To strengthen the validity and rigor of the experiment, we conducted the study with 10 post-surgical subjects, to the best of our ability.

Reference:

33. Zhang, X. et al. Impact of whey protein isolate and eccentric training on quadriceps mass and strength in patients with anterior cruciate ligament rupture: A randomized controlled trial. *J. Rehabil. Med.* **52**, 1-8 (2020).

Comment 20:

Fig. 3a. Please correct the word “angle” in the figure.

Response 20:

We were really sorry for the oversight in our manuscript. Thank you for pointing it out. We have corrected the “angle” in the manuscript.

Figure 3 has been moved to Fig. S4.

Fig. S4 | Energy regeneration during isokinetic training. (a) Lifting and retracting torque model. Quadriceps femoris contraction force F_Q dominates the lifting force, with the corresponding moment arm l_Q . Hamstring muscle contraction force F_H dominates the retracting force, with the corresponding moment arm l_H . β is a normalized muscle activation level. G is the gravitational force, and l_G is the corresponding moment arm. **(b)** Energy regeneration circuit. The low-side MOSFETs are switched ON/OFF. When the MOSFETs are OFF, the charging current I_{charge} equals the phase current I_a , and the current flows into the ultracapacitor. When the MOSFETs are ON, although the induced phase current exists, the charging current $I_{charge} = 0$.

Reviewer #3:**General Comment**

The experiment design and data analysis are generally appropriate. I have the following comments:

Our Response

We deeply appreciate your recognition of our work.

Specific Comment:

Comment 1:

Please summarize the experimental data more systematically, with mean and standard deviation for trained and untrained legs, before and after the training.

Response 1:

Thank you for your professional comment. To systematically summarize the experimental data and provide detailed information on the mean and standard deviation of the trained and untrained legs before and after training, we have organized the data into tables and included them in the supplementary material (**Tables S4–7, Tables S9–10**)

The contents of each table are as follows:

Table S4: Regenerated power and consumed power of healthy subjects and post-surgical subjects. Corresponding to **Fig. 4**.

Table S4 Power-free validation. Regeneration power and consumer power of healthy subjects (n=10) and post-surgical subjects (n=10).											
		subject 1	subject 2	subject 3	subject 4	subject 5	subject 6	subject 7	subject 8	subject 9	subject 10
Healthy Subjects	Regeneration power (W)	1.62±0.18	1.97±0.50	2.21±0.14	2.25±0.29	2.35±0.22	2.42±0.03	2.62±0.21	2.85±0.25	3.69±0.16	4.66±0.28
	Consumed power (W)	2.32±0.03	2.35±0.04	2.36±0.04	2.35±0.03	2.33±0.04	2.35±0.03	2.32±0.04	2.34±0.04	2.32±0.02	2.33±0.04
	Ratio (%)	130%	155%	173%	177%	188%	191%	211%	227%	299%	373%
Post-surgical Subjects	Regeneration power (W)	1.76±0.66	2.67±1.03	2.29±0.96	2.61±0.87	2.38±1.02	2.72±0.65	1.81±0.44	2.20±0.61	1.50±0.50	2.52±0.50
	Consumed power (W)	1.17±0.03	1.19±0.02	1.21±0.01	1.20±0.02	1.27±0.03	1.25±0.01	1.18±0.02	1.24±0.01	1.16±0.03	1.20±0.02
	Ratio (%)	150%	224%	190%	217%	187%	218%	154%	178%	130%	210%

Table S5: Tight Cross-sectional area of both trained leg and untrained leg for post-surgical subjects. Corresponding to **Fig. 5**.

Table S5 | Tight Cross-sectional Area. Tight cross-sectional area of all post-surgical subjects (n=10) before and after 6 weeks training.
BT: before training, AT: after training, Qua: quadriceps, Ham: hamstrings.

		subject 1	subject 2	subject 3	subject 4	subject 5	subject 6	subject 7	subject 8	subject 9	subject 10	mean±std
Trained leg	Qua BT (cm ²)	59.02	72.43	42.81	64.23	44.62	36.87	51.62	47.78	40.58	57.22	51.72±11.35
	Qua AT (cm ²)	61.4	74.53	43.11	65.81	46.96	40.45	56.63	51.67	43.88	61.96	54.64±11.26
	Ham BT (cm ²)	26.46	38.45	21.43	31.53	21.85	24.23	27.04	18.46	19.44	23.06	25.20±6.06
	Ham AT (cm ²)	30.48	39.23	24.49	32.63	23.71	28.07	30.54	20.03	22.76	24.17	27.61±5.73
	Qua Increment (cm ²)	2.38	2.10	0.30	1.58	2.34	3.58	5.01	3.89	3.30	4.74	2.92±1.46
	Qua Increment (%)	4.03%	2.90%	0.70%	2.46%	5.24%	9.71%	9.71%	8.14%	8.13%	8.28%	5.93%±3.27%
	Ham Increment (cm ²)	4.02	0.78	3.06	1.1	1.86	3.84	3.5	1.57	3.32	1.11	2.42±1.25
Ham Increment (%)	15.19%	2.03%	14.28%	3.49%	8.51%	15.85%	12.94%	8.50%	17.08%	4.81%	10.27%±5.53%	
Untrained leg	Qua BT (cm ²)	81.76	87.44	59.04	79.67	51.53	48	75.43	50.55	62.09	77.73	67.32±14.68
	Qua AT (cm ²)	81.95	85.99	59.14	79.96	52.07	48.54	76.51	51.38	62.71	76.71	67.50±14.23
	Ham BT (cm ²)	24.08	38.39	23.16	29.65	22.67	25.16	27.75	22.31	21.24	25.37	25.98±5.06
	Ham AT (cm ²)	25.75	35.14	24.6	30.07	22.85	26.48	28.3	22.84	22.64	24.27	26.29±3.96
	Qua Increment (cm ²)	0.19	-1.45	0.1	0.29	0.54	0.54	1.08	0.83	0.62	-1.02	0.17±0.80
	Qua Increment (%)	0.23%	-1.66%	0.17%	0.36%	1.05%	1.13%	1.43%	1.64%	1.00%	-1.31%	0.40%±1.11%
	Ham Increment (cm ²)	1.67	-3.25	1.44	0.42	0.18	1.32	0.55	0.53	1.4	-1.1	0.32±1.50
Ham Increment (%)	6.94%	-8.47%	6.22%	1.42%	0.79%	5.25%	1.98%	2.38%	6.59%	-4.34%	1.88%±5.00%	

Table S6: Maximum lifting torque of trained leg for all post-surgical subjects. Corresponding to all **red lines and dots in Fig. 6.**

Table S6 Maximum Lifting Torque. Maximum lifting torque of trained leg for all post-surgical subjects (mean±std, Nm).										
Week	subject 1	subject 2	subject 3	subject 4	subject 5	subject 6	subject 7	subject 8	subject 9	subject 10
1	26.43±1.66	46.39±0.65	24.30±3.71	37.09±1.39	20.74±1.06	32.46±0.54	34.60±2.66	29.17±1.47	14.37±3.32	40.17±1.16
	25.69±1.31	60.16±3.67	29.15±1.44	39.50±1.03	30.67±0.99	41.67±1.81	43.04±0.49	29.34±1.85	19.82±1.76	33.18±0.69
2	27.83±2.45	66.51±3.98	30.72±1.53	48.61±1.98	35.60±1.08	44.90±1.44	49.55±1.09	31.90±0.43	26.81±2.63	46.19±1.35
	29.41±2.13	69.21±1.82	29.49±2.05	59.38±0.54	30.80±1.56	45.24±0.31	46.30±1.40	33.69±0.12	27.12±1.93	59.35±1.18
3	29.23±1.73	73.49±0.86	36.81±1.35	51.61±1.39	34.55±1.29	46.51±1.95	48.48±0.72	31.75±2.89	25.86±1.07	60.14±1.74
	28.76±2.43	70.53±2.31	34.25±1.40	56.32±2.37	40.67±1.33	52.24±1.04	51.90±2.19	34.68±1.48	22.64±1.27	62.15±1.68
4	27.49±2.77	76.00±1.32	37.93±0.78	56.65±1.64	31.94±3.52	50.65±4.60	55.46±1.69	35.88±1.19	31.30±2.95	63.50±2.20
	30.07±4.61	80.26±1.54	36.65±3.22	58.89±1.01	41.86±2.11	53.02±4.03	46.66±2.83	34.67±2.84	28.93±4.43	71.43±1.59
5	30.04±1.00	81.27±1.31	38.94±0.87	62.24±1.18	39.06±4.13	57.54±1.86	55.29±2.34	34.76±1.11	31.70±4.00	67.40±0.62
	32.72±1.64	79.48±1.84	41.11±3.06	70.28±1.19	46.33±1.44	61.70±0.85	53.70±0.87	33.89±1.98	29.16±3.94	74.25±1.07
6	31.71±1.14	85.18±2.14	38.79±1.34	62.97±0.87	36.43±1.86	57.02±0.97	58.46±0.97	38.23±0.84	33.19±2.11	78.94±1.89
	32.31±1.95	79.97±0.69	38.92±3.04	67.96±3.69	43.29±1.67	59.66±1.07	58.80±0.83	38.93±0.78	34.43±2.82	64.94±3.85

Table S7: Maximum retracting torque of trained leg for all post-surgical subjects. Corresponding to all **blue lines and dots in Fig. 6.**

Table S7 | Maximum Retracting Torque. Maximum retracting torque of trained leg for all post-surgical subjects (mean±std, Nm). A negative value indicates torque generated in the leg retracting direction, which is the opposite of the leg lifting direction.

Week	subject 1	subject 2	subject 3	subject 4	subject 5	subject 6	subject 7	subject 8	subject 9	subject 10
1	-27.27±1.13	-39.50±1.47	-16.11±0.74	-23.99±0.80	-10.09±1.01	-17.90±1.39	-20.12±0.36	-12.65±2.18	-9.50±0.65	-20.61±0.58
	-28.22±1.88	-39.55±3.04	-13.87±1.07	-29.52±0.97	-12.35±1.13	-25.97±1.78	-23.79±0.83	-19.29±2.29	-11.80±1.05	-21.29±1.29
2	-35.74±1.45	-40.03±2.51	-23.47±1.36	-35.31±0.66	-25.53±1.58	-29.96±1.51	-26.95±1.47	-20.85±0.89	-18.16±2.33	-27.93±1.56
	-35.71±1.56	-40.29±2.30	-26.99±2.08	-37.40±1.18	-24.51±2.33	-27.42±0.78	-26.28±1.47	-21.23±0.99	-21.93±2.27	-26.79±1.77
3	-38.57±1.38	-46.88±1.15	-32.74±1.58	-39.77±1.85	-27.50±1.99	-26.46±2.39	-29.72±0.78	-23.25±1.73	-23.47±2.26	-30.87±0.73
	-40.45±1.30	-42.67±1.07	-34.42±2.32	-44.19±2.60	-34.13±1.11	-30.52±1.82	-35.92±0.79	-25.42±1.99	-27.83±0.58	-34.35±1.41
4	-39.36±2.88	-47.29±1.12	-33.63±0.51	-44.48±0.64	-31.77±3.31	-34.59±3.46	-32.91±0.60	-27.91±0.72	-28.45±0.71	-33.50±1.23
	-38.21±2.54	-45.88±1.91	-33.99±2.79	-56.27±0.87	-30.85±1.26	-27.67±3.11	-39.40±0.49	-31.79±0.79	-26.18±1.08	-32.38±1.28
5	-41.33±1.72	-46.74±0.57	-41.77±1.59	-45.28±2.51	-27.66±1.74	-36.06±1.64	-42.99±1.49	-40.09±1.76	-48.63±1.20	-62.28±2.03
	-43.52±1.20	-48.42±1.52	-43.99±1.91	-48.91±1.46	-36.07±2.12	-42.47±2.30	-43.22±0.62	-37.30±2.35	-38.27±3.92	-62.53±1.55
6	-41.62±1.43	-47.43±1.75	-39.71±1.77	-56.13±0.88	-37.24±3.10	-40.70±0.61	-44.03±0.99	-36.92±3.13	-47.03±1.40	-60.92±1.48
	-43.75±1.95	-48.59±0.76	-34.55±2.89	-55.61±2.12	-32.59±4.50	-42.47±0.59	-45.11±0.66	-41.82±1.94	-45.80±0.98	-61.00±3.28

Table S9: Total lifting work of trained leg for all post-surgical subjects. Corresponding to all red lines and dots in Fig. S9.

Table S9 | Lifting Work. Mean lifting work of trained leg for all post-surgical subjects in one repetition (mean±std, Nm).

Week	subject 1	subject 2	subject 3	subject 4	subject 5	subject 6	subject 7	subject 8	subject 9	subject 10
1	16.41±4.41	40.92±5.14	14.67±3.74	24.22±6.43	16.90±3.35	20.01±4.91	24.74±3.78	17.90±4.11	10.25±2.63	25.05±2.33
	16.92±3.80	53.36±5.69	17.15±3.67	28.46±9.12	24.68±5.77	29.32±6.62	30.89±6.80	20.65±3.48	13.66±3.38	20.83±4.67
2	19.93±2.01	59.00±9.12	19.16±4.23	37.28±6.60	29.55±7.55	29.77±8.52	34.24±5.75	21.96±4.09	19.31±4.20	31.72±5.15
	21.10±4.65	61.40±6.82	20.38±8.45	48.29±8.29	25.69±6.98	33.48±4.34	34.21±5.67	26.17±5.66	19.16±2.29	40.59±2.95
3	21.55±4.54	65.19±4.87	28.48±6.78	41.32±10.12	30.64±6.27	37.04±8.11	38.00±4.52	27.87±8.86	19.90±4.05	41.64±7.00
	20.56±2.34	62.56±10.29	24.77±6.97	46.91±4.41	36.08±6.58	34.56±5.88	42.64±7.42	29.28±4.15	19.62±3.45	43.73±5.18
4	19.84±5.45	67.42±5.94	26.05±8.03	48.29±10.26	28.33±5.16	36.99±9.14	45.96±5.62	28.62±5.77	25.24±8.67	50.66±9.11
	21.16±5.33	71.20±10.26	24.91±7.12	50.37±3.36	37.13±7.53	37.38±3.72	39.12±9.86	28.73±5.97	23.57±6.60	55.08±7.40
5	24.39±5.15	72.09±4.65	30.02±7.59	53.76±9.29	34.65±9.72	45.01±6.33	49.04±8.88	28.85±5.53	27.34±6.28	56.14±3.49
	28.17±3.62	70.50±10.37	29.56±4.78	57.44±6.77	41.10±5.80	47.75±10.19	47.63±4.65	29.37±3.46	24.70±4.80	65.07±12.92
6	28.13±6.63	75.56±9.96	34.41±7.70	55.10±10.08	32.31±8.73	50.58±6.84	51.85±11.33	33.91±8.85	28.34±9.60	66.86±8.53
	28.66±7.97	70.94±10.40	34.52±4.86	60.28±8.47	38.40±6.60	52.92±10.08	52.16±9.84	34.54±4.40	30.54±3.97	57.61±8.34

Table S10: Total retracting work of trained leg for all post-surgical subjects. Corresponding to all **blue lines and dots in Fig. S9**.

Table S10 Retracting Work. Mean retracting work of trained leg for all post-surgical subjects in one repetition (mean±std, Nm).										
Week	subject 1	subject 2	subject 3	subject 4	subject 5	subject 6	subject 7	subject 8	subject 9	subject 10
1	16.93±4.78	34.84±6.88	9.73±2.66	15.66±3.41	8.22±2.69	11.04±4.98	14.38±4.44	19.82±4.66	11.99±2.29	25.85±3.13
	18.59±3.53	35.09±5.49	8.16±2.25	21.27±4.22	9.94±3.61	18.27±3.26	17.07±5.46	22.81±2.52	18.82±4.43	29.95±5.17
2	25.60±4.08	35.51±7.72	14.64±4.65	27.08±8.53	21.19±8.31	19.86±7.59	18.92±8.54	23.84±6.18	19.17±8.91	31.08±2.19
	25.63±5.07	35.74±4.39	18.65±4.82	30.42±7.98	20.45±5.42	20.29±7.90	25.40±8.24	30.79±8.67	25.33±8.57	35.12±5.56
3	28.44±8.04	41.58±4.86	25.34±3.13	31.84±5.04	24.39±3.22	21.07±3.88	31.68±8.33	33.38±5.90	23.37±4.68	36.22±8.36
	28.92±3.78	37.85±4.83	24.90±5.62	36.81±5.85	30.28±4.95	20.19±6.82	30.74±3.81	31.07±4.04	29.24±6.53	41.86±6.21
4	28.42±7.48	41.95±8.24	23.10±3.85	37.92±9.27	28.18±3.64	25.26±5.26	30.50±7.64	32.97±9.16	31.66±4.86	46.53±3.76
	26.89±4.76	40.70±4.49	23.11±3.19	48.13±10.59	27.37±6.89	19.51±5.26	36.27±6.16	31.05±7.41	33.04±9.55	48.74±10.93
5	33.55±8.06	41.46±10.26	32.21±4.17	39.11±4.70	24.54±6.60	28.21±3.86	38.13±6.92	33.27±9.88	41.93±9.19	51.87±3.88
	37.47±8.76	42.95±5.18	31.63±9.11	39.98±9.32	32.00±5.46	32.86±4.14	38.34±3.64	32.32±9.07	32.41±6.23	54.80±10.10
6	36.92±3.74	42.07±9.63	35.23±5.47	49.11±7.87	33.03±3.55	36.11±6.64	39.06±3.97	32.75±4.53	40.16±10.06	51.60±6.08
	38.80±7.37	43.10±3.58	30.64±7.75	49.33±9.74	28.91±4.44	37.67±4.74	40.01±7.61	37.10±4.00	40.62±9.68	54.11±4.00

Comment 2:

Since there is a control group (untrained leg), the hypothesis to be tested should be the improvement/change in the trained leg is larger than that of the untrained leg.

Response 2:

Thank you for your professional comment. We need to better clarify the hypothesis test of the trained leg and the untrained leg.

Fig. 5 has been reorganized to better present the significant differences in a more visually intuitive and effective manner. A hypothesis test analysis was conducted to **compare the improvements in the trained leg with those in the untrained leg**, demonstrating that the changes in the trained leg were significantly greater. The results are presented in Fig. 5f and Fig. 5g.

Fig. 5 | Evaluation of muscle morphology after a 6-week rehabilitation program. (a) MRI images of quadriceps and hamstrings in both trained and untrained legs before and after training. (b)-(e) Quantitative cross-sectional area (CSA) before and after training. (b) Quadriceps of trained leg. (c) Hamstrings of trained leg (d) Quadriceps of

untrained leg. (e) Hamstrings of untrained leg. (f) Increased CSA of quadriceps/hamstrings in the trained/untrained leg. (g) Growth rate of quadriceps/hamstrings in the trained/untrained leg. ACL: Anterior Cruciate Ligament, MCL: Medial Collateral Ligament, PD: Patellar Dislocation. * refers to p-value < 0.05 by one-tail t-test.

Detailed information of hypothesis test:

We conducted the hypothesis test on quadriceps area increment between the trained and untrained legs. The result showed a p-value of $p = 3.03 \times 10^{-4} \ll 0.05$. This result indicates a significant difference in **quadriceps CSA increment between the trained and untrained legs**. Quadriceps percentage increment between the trained and untrained legs give the similar result $p = 2.33 \times 10^{-4} \ll 0.05$.

Similarly, we conducted the same hypothesis test for the hamstring increment between the trained and untrained legs. The p-value was $p = 6.88 \times 10^{-5} \ll 0.05$. This result indicates a significant difference in **hamstring CSA increment between the trained and untrained legs**. Hamstring percentage increment between the trained and untrained legs give the similar result $p = 4.40 \times 10^{-6} \ll 0.05$.

Comment 3:

Comparing Figs 6b and 6f, the CSA of quadriceps of the trained leg appears to be consistently smaller than the untrained leg. Please explain.

Response 3:

Thank you for your thorough review. In response to your question, the quadriceps CSAs of the trained leg were consistently smaller than those of the untrained leg (see Table. S5). **The trained leg refers to the injured leg that underwent surgery, while the untrained leg refers to the uninjured leg.**

After knee surgery, patients often avoid the necessary movement of the injured knee due to temporary dysfunction, pain, swelling, and limited flexion and extension. This lack of movement stimulation can lead to a gradual decrease in muscle volume, strength, and function, potentially resulting in partial **muscle atrophy**. CSAs are a primary indicator of muscle growth and recovery (34-36), with larger CSAs corresponding to a greater muscle mass in the leg. **Muscle atrophy leads to a reduction in muscle mass and volume, which consequently results in a decrease in CSAs.**

Table S5 | Tight Cross-sectional Area. Tight cross-sectional area of all post-surgical subjects (n=10) before and after 6 weeks training.
BT: before training, AT: after training, Qua: quadriceps, Ham: hamstrings.

		subject 1	subject 2	subject 3	subject 4	subject 5	subject 6	subject 7	subject 8	subject 9	subject 10	mean±std
Trained leg	Qua BT (cm ²)	59.02	72.43	42.81	64.23	44.62	36.87	51.62	47.78	40.58	57.22	51.72±11.35
	Qua AT (cm ²)	61.4	74.53	43.11	65.81	46.96	40.45	56.63	51.67	43.88	61.96	54.64±11.26
	Ham BT (cm ²)	26.46	38.45	21.43	31.53	21.85	24.23	27.04	18.46	19.44	23.06	25.20±6.06
	Ham AT (cm ²)	30.48	39.23	24.49	32.63	23.71	28.07	30.54	20.03	22.76	24.17	27.61±5.73
	Qua Increment (cm ²)	2.38	2.10	0.30	1.58	2.34	3.58	5.01	3.89	3.30	4.74	2.92±1.46
	Qua Increment (%)	4.03%	2.90%	0.70%	2.46%	5.24%	9.71%	9.71%	8.14%	8.13%	8.28%	5.93%±3.27%
	Ham Increment (cm ²)	4.02	0.78	3.06	1.1	1.86	3.84	3.5	1.57	3.32	1.11	2.42±1.25
	Ham Increment (%)	15.19%	2.03%	14.28%	3.49%	8.51%	15.85%	12.94%	8.50%	17.08%	4.81%	10.27%±5.53%
Untrained leg	Qua BT (cm ²)	81.76	87.44	59.04	79.67	51.53	48	75.43	50.55	62.09	77.73	67.32±14.68
	Qua AT (cm ²)	81.95	85.99	59.14	79.96	52.07	48.54	76.51	51.38	62.71	76.71	67.50±14.23
	Ham BT (cm ²)	24.08	38.39	23.16	29.65	22.67	25.16	27.75	22.31	21.24	25.37	25.98±5.06
	Ham AT (cm ²)	25.75	35.14	24.6	30.07	22.85	26.48	28.3	22.84	22.64	24.27	26.29±3.96
	Qua Increment (cm ²)	0.19	-1.45	0.1	0.29	0.54	0.54	1.08	0.83	0.62	-1.02	0.17±0.80
	Qua Increment (%)	0.23%	-1.66%	0.17%	0.36%	1.05%	1.13%	1.43%	1.64%	1.00%	-1.31%	0.40%±1.11%
	Ham Increment (cm ²)	1.67	-3.25	1.44	0.42	0.18	1.32	0.55	0.53	1.4	-1.1	0.32±1.50
	Ham Increment (%)	6.94%	-8.47%	6.22%	1.42%	0.79%	5.25%	1.98%	2.38%	6.59%	-4.34%	1.88%±5.00%

Reference:

34. An, K. N., Linscheid, R. L., Brand, P. W., Correlation of physiological cross-sectional areas of muscle and tendon. *J. Hand. Surg. Br. Eur. Vol. 16*, 66-67 (1991).
35. Franchi, M V., et al. Muscle thickness correlates to muscle cross-sectional area in the assessment of strength training-induced hypertrophy, *Scand. J. Med. Sci. Sports 28*, 846-853 (2018).
36. Chen L., et al. How Do Muscle Function and Quality Affect the Progression of KOA? A Narrative Review. *Orthop Surg. 16*, 802-810 (2024).

Comment 4:

On line 345, it should be "We consider p-values of less than 0.05 to be statistically significant". (Remarks on code availability)

Response 4:

We were sorry for the oversight in our manuscript. Thank you for pointing it out. We have corrected all instances of "t-value" to "p-value" in the manuscript.

Before Correction:

Statistical methods

All statistical analyses were conducted on data collected from all subjects and performed with the SPSS 25.0 statistical package (IBM, Armonk, NY, USA), and Microsoft Excel. Using one-tailed t-test to analyze the significant difference of data. We consider t-values of less than 0.05 to be statistically significant. All data were analyzed independently, and shown as mean \pm SD.

After Correction:

Statistical methods

All statistical analyses were conducted on data collected from all subjects and performed with the SPSS 25.0 statistical package (IBM, Armonk, NY, USA), and Microsoft Excel. Using one-tailed t-test to analyze the significant difference of data. We consider **p-values** of less than 0.05 to be statistically significant. All data were analyzed independently, and shown as mean \pm SD.

**Responses to Comments on "Nature Communications-#NCOMMS-
24-18580C"**

Dear Reviewers:

We would like to express our sincere gratitude for accepting our manuscript entitled "*Power-Free Knee Rehabilitation Robot for Home-Based Isokinetic Training*" in principle. The constructive comments and suggestions have been fully addressed point-by-point in this letter. For the convenience of the reviewers, all comments from the reviewers are highlighted in purple, and the corresponding modifications in the revised manuscript are marked in blue. We believe that the revisions have enhanced the quality and clarity of our manuscript, making it more suitable for publication in *Nature Communications*.

Best,

Liu Wang, Yanggang Feng

Reviewer #1:

General Comment:

The objectives of this study were

- (1) to present an integrated isokinetic knee rehabilitation robot for home-based isokinetic training,
- (2) test the device with 10 post-surgical subjects and 10 healthy individuals, and
- (3) carry out a 6-week interventional clinical trial with the 10 patients after knee surgeries and validate recovered knee functions after training using the robot.

The manuscript's theme is adequate for publication in Nature Communications, is clinically relevant to the area of knee home-based rehabilitation and has novelty for the specific area of assistive technologies for knee rehabilitation. According to the authors, the main findings of the study were that the muscle growth and strength improvements achieved with the robot outperformed existing commercial rehabilitation devices, thereby indicating that the robot presents a viable option for home-based knee rehabilitation, significantly enhancing the accessibility of effective treatment. The reviewed literature is now adequate and updated. The authors correctly conducted the reader through the possible limitations of existent isokinetic devices and a possible way to resolve these problems through a new device. The authors now present at the introduction recent results from previous studies regarding the use of isokinetic devices in knee rehabilitation, what is missing in the area and how this new device will achieve similar/better results when used clinically than the exercises that can be done without the need of any equipment. Ethical aspects regarding the clinical trial were correctly presented in the manuscript. The methods section is well described, and now all aspects that were raised by the reviewer are completely clear. The results are now well organized, and several sections related to the methods were correctly placed in the Methods Section. In the discussion section, the authors clearly described the device features, described several results that were obtained with the robot, and now adequately discuss their results while contrasting them with the existent literature. Congratulations to the authors on the excellent work done in the revised manuscript and on the new technology that they presented very well and that may indeed help in the rehabilitation of patients with knee problems. The manuscript now demonstrates the rationale for the use of this new technology in clinical practice and **can therefore be accepted for publication.**

Our response:

We deeply appreciate your positive evaluation of our manuscript. This paper has greatly benefited from your professional suggestions. Thank you very much for your recognition.

Minor Comment:

Comment 1:

I would add some of the information that you provided in the Rebuttal Letter for the Reviewers in the manuscript.

For example, I would add this sentence at the introduction or at the discussion sections.
- We were unable to find any studies that utilized isokinetic training devices in home-based settings.

Response 1:

Thank you for your insightful comment.

In the second paragraph of INTRODUCTION, we added the meaningful sentence you suggested:

“We were unable to find any studies that utilized isokinetic training devices in home-based settings. There has been a growing yet unfulfilled need for compact isokinetic training systems that can be deployed for home-based rehabilitation.”

Comment 2:

I would use the same units for the angular velocities throughout the text, and perhaps $60^\circ \cdot s^{-1}$ (with the $^{-1}$ superscript) may be better.

Response 2:

Thank you for your professional comment. We have updated all the units in the angular velocities throughout the text by $^\circ \cdot s^{-1}$ in the related content in the manuscript, with the correction as follows.

Revised 1: In the subsection “Demonstration of isokinetic training” in RESULTS, we revised:

“A constant training velocity of $\omega = 60^\circ \cdot s^{-1}$ is maintained during both lifting and retracting processes (33).”

“Notably, the training device is designed not to assist leg motion but to provide accommodative resistance if the velocity exceeds $60^\circ \cdot s^{-1}$, ensuring the training intensity.”

Revised 2: In the subsection “Efficacy comparison between our study and commercial isokinetic robots” in DISCUSSION, we revised:

Other experiments conducted using commercial isokinetic devices with the same isokinetic training speed ($60^\circ \cdot s^{-1}$) but varying training durations and cycles also did not demonstrate torque increases surpassing the results of our study (33, 37-39, See **Fig. 7b**).

Revised 3: In the subsection “Isokinetic program for 10 post-surgical subjects” in METHODS, we revised:

The post-surgical subjects participated in sessions twice a week for six weeks, with

each session conducted at a velocity of $60^\circ \cdot s^{-1}$.

Revised 4: In the subsection “Procedure of isokinetic training” in METHODS, we revised:

In all trials, a training speed of $60^\circ \cdot s^{-1}$ was used. Previous studies show that $60^\circ \cdot s^{-1}$ is effective in various rehabilitation aspects, contributing to improvements in muscle strength improvement and functional recovery (41-42).

Comment 3:

Also, please add the range of angular velocities that the isokinetic training device can work in the Methods section and add information regarding if different angular velocities can be determined at the same exercise for the two antagonist muscle groups: knee extensors and flexors (e.g., can we set up the angular velocities of $60^\circ \cdot s^{-1}$ for the knee extensors and $90^\circ \cdot s^{-1}$ for the knee flexors?). You showed in your rebuttal letter results from three angular velocities, and how you set up the angular velocities for the two movements (knee extension vs flexion) needs to be better described.

Response 3:

Thank you for your valuable comment.

In response, we have updated the “Procedure of isokinetic training” in METHODS by adding the range of angular velocities at which the isokinetic training device operates.

In the revised “Procedure of isokinetic training” in METHODS, we added:

“Additionally, the proposed device is capable of accommodating different speeds for isokinetic training, such as $90^\circ \cdot s^{-1}$ and $120^\circ \cdot s^{-1}$.”

Our study primarily focuses on training with the same isokinetic speed for both the knee extensors and knee flexors, using angular velocities such as $60^\circ \cdot s^{-1}$ or $90^\circ \cdot s^{-1}$ for both muscle groups. The use of different angular velocities for the antagonist muscle groups will be explored in future work.

To address this limitation, we have added the related content in “Limitations”:

“Fifth, future studies will explore the use of varying angular velocities for antagonist muscle groups during the same exercise, such as applying angular velocities of $60^\circ \cdot s^{-1}$ for the knee extensors and $90^\circ \cdot s^{-1}$ for the knee flexors.”

Reviewer #3:

General Comment:

Thanks for the thorough revision. I don't have further comments.

Our response:

We deeply appreciate your positive evaluation. This paper has greatly benefited from your professional suggestions. Thank you very much for your recognition.